# Sample-efficient Learning of Concepts with Theoretical Guarantees: from Data to Concepts without Interventions

**Hidde Fokkema**
Korteweg-de Vries Institute for Mathematics
University of Amsterdam
`h.j.fokkema@uva.nl`

**Tim van Erven**[*]
Korteweg-de Vries Institute for Mathematics
University of Amsterdam
`tim@timvanerven.nl`

**Sara Magliacane**[*]
Informatics Institute
University of Amsterdam
`s.magliacane@uva.nl`

## Abstract

Machine learning is a vital part of many real-world systems, but concerns remain about the lack of interpretability, explainability and robustness of black-box AI systems. Concept Bottleneck Models (CBM) address some of these challenges by learning interpretable *concepts* from high-dimensional data, e.g. images, which are used to predict labels. An important issue in CBMs are spurious correlations between concepts, which effectively lead to learning "wrong" concepts. Current mitigating strategies have strong assumptions, e.g., they assume that the concepts are statistically independent of each other, or require substantial interaction in terms of both interventions and labels provided by annotators. In this paper, we describe a framework that provides theoretical guarantees on the correctness of the learned concepts and on the number of required labels, without requiring any interventions. Our framework leverages causal representation learning (CRL) methods to learn latent causal variables from high-dimensional observations in a unsupervised way, and then learns to align these variables with interpretable concepts with few concept labels. We propose a linear and a non-parametric estimator for this mapping, providing a finite-sample high probability result in the linear case and an asymptotic consistency result for the non-parametric estimator. We evaluate our framework in synthetic and image benchmarks, showing that the learned concepts have less impurities and are often more accurate than other CBMs, even in settings with strong correlations between concepts.

## 1 Introduction

Machine learning is a vital part of many real-world systems, but concerns remain about the lack of interpretability, robustness and safety of current systems [9]. These issues might be exacerbated by the lack of guarantees in explaining the behavior of AI systems in terms of interpretable, high-level *concepts*. The field of interpretable machine learning and explainable AI has developed many techniques to interpret models and explain their predictions [44], either by extracting known concepts from the internals of black-box models [25, 15, 16, 36], or by building the explicit use of concepts into the internals of these systems, e.g. as in *concept bottleneck models* (CBM) [28, 21, 39, 57].

---

[*]Equal contribution

39th Conference on Neural Information Processing Systems (NeurIPS 2025).

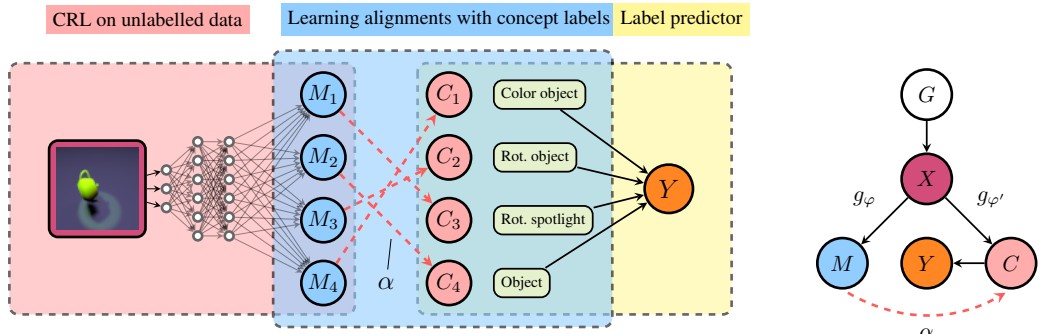

Figure 1: **Left**: An overview of our framework: we learn the alignment function $\alpha$ that maps causal representations $M_i$ learned on cheap unlabelled data by a causal representation learning (CRL) encoder, to interpretable concepts $C_j$ using only few concept labels. As in standard CBMs, these concepts are used in a downstream task like regression or classification of $Y$. **Right**: Data generating process, where $G$ are the latent causal variables, $X$ is an observation, $M$ are the representations learned by a model $g_\psi$, $C$ are the interpretable concepts and $Y$ is the final label.

An advantage of CBMs is that they can in many cases provide similar accuracy in terms of prediction compared to black-box methods, while also ensuring interpretability by construction [28, 21, 57], as opposed to post-hoc methods [8]. Users may also interact with the models by verifying and, if necessary, correcting the detected concepts at test time [28]. However, existing methods suffer from two important limitations. First, *spurious correlations* in the training data may lead to learned concepts that do not capture the intended semantics of the concept [15, 41], or encode more information than intended [38, 58], even with concept-level supervision. Recent works propose mitigation strategies, but implicitly assume that the concepts are statistically independent [39, 50] or require interventions on the data generating process with labels [40]. Secondly, in addition to labels required for the learning task at hand, CBMs require *many expensive concept labels* that may not always be available. Attempts at working around this issue have so far focused on obtaining concept labels from additional sources like GPT-3 [45] or using multi-model information [54].

In this paper, we propose a framework that provides theoretical guarantees on the correctness of the learned concepts and on the number of required labels without requiring any interventions. We assume that the concepts that we want to learn correspond to latent variables in a causal system and allow arbitrary dependences between them. As illustrated in Fig. 1 (left), we propose a two-step pipeline: (i) leverage out-of-the-box causal representation learning (CRL) methods to learn provably disentangled representations of concepts $M$ even in case of spurious correlations; (ii) learn an alignment map $\alpha$ between these representations and the interpretable concepts $C$ with theoretical guarantees and as few concept labels as possible. In our setting, the map $\alpha$ consists of a permutation and element-wise transformations. Unlike standard CBMs, our pipeline only uses concept labels in the second stage to disambiguate the relation between the representations learned by CRL and the concepts. This requires much fewer concept labels, addressing the second limitation for CBMs.

We introduce two principled estimators to learn $\alpha$ based on variants of the group lasso [56] and a weighted matching procedure. The first estimator is based on a linear model with an optional feature map, and comes with a high probability finite-sample result on its correctness (Thm. 4.2). This result provides explicit guidance for tuning the regularization hyperparameter, and shows how the required number of concept labels scales with problem parameters like the number of concepts and the dimension of the feature map. Our second estimator is based on a kernelized procedure, allowing for more flexibility, but comes only with asymptotic guarantees (Thm. 5.2).

We evaluate our framework in synthetic and image benchmarks with both real-valued and binary concepts. The experiments show that the learned concepts have less impurities, measured in terms of OIS and NIS [58], and are often more accurate than other CBMs. The experimental results additionally indicate that our estimator performs well beyond the assumptions of our theorems. For example, our estimators are able to find the correct permutation, even in settings with strong correlations between the concepts.

## 2 Related Work

Extracting high-level concepts from the inner workings of machine learning models has gained traction over the last years [3, 14, 42]. These concepts can be used to create interpretable explanations. In concept bottleneck models, the concepts are hard-coded into the structure of the model [28, 39, 21, 57]. This approach has been quite successful, with multiple variants, e.g. probabilistic CBMs [26] or energy-based CBMs [53]. One drawback of CBMs is that the learned concepts may not correctly capture the intended semantics [41] and that their representations can encode impurities, e.g. more information than intended, [38, 58]. Havasi et al. [17] address this problem by allowing a more expressive autoregressive concept predictor that can take advantage of the correlations between concepts. On the other hand, the issue is exacerbated when there are spurious correlations among concepts in the training data. In this case, Marconato et al. [39] encourage disentangled representations of the concepts by adding a regularization term in the concept predictor. Similarly, Sheth and Kahou [50] introduce a concept orthogonal loss that encourages separation between concept representations. Both approaches implicitly assume the concepts to be statistically independent, which means that they can fail to learn the correct concepts in settings in which there are dependences.

We instead propose to leverage causal representation learning (CRL) [49] as the first step of our pipeline. CRL aims at recovering potentially dependent latent variables from observations. Identifying these latent variables is often only possible up to a permutation and element-wise transformation. Many CRL methods exist with different assumptions on the available data, e.g. the availability of interventional, counterfactual or temporal data, or parametric assumptions about the underlying system and observations, e.g., [20, 24, 51, 31, 33, 2, 30, 55] and many others. Most methods focus on the infinite sample setting, with the exception of [1], which provides sample complexity results for the CRL task with interventions. Our framework is agnostic to the CRL method used and focuses on the supervised learning task of aligning concepts efficiently to the causal representations.

Recent work relates identifiable representation learning, including causal representation learning, with concept-based models, but in an unsupervised concept discovery setting, as opposed to our supervised setting. Leemann et al. [32] propose an identifiable concept discovery method that uses classic methods like PCA and ICA for independent concepts, but also introduces novel disentanglement methods for dependent concepts, based on disjoint mechanisms or independent mechanisms. Rajendran et al. [46] argue that the identifiability of CRL methods often requires too many assumptions, e.g. access to interventional data, and hence relaxes the identifiability of concepts by defining them as affine subspaces of the latent space of the causal variables. Given a set of concept conditional datasets, they prove they can recover these concepts up to linear transformations.

The most closely related work is by Marconato et al. [40], who describe theoretical framework that shows how that disentangled representations can be used in CBMs to avoid concept leakage. Their theoretical framework suggests that to align learned embeddings and concepts one might require interventions on the data generating process and labels. Instead, we do not require interventions. Moreover, we provide a theoretical analysis of the alignment function, proving guarantees about error bounds on the learned concepts or the required number of labels.

## 3 Framework and Main Definitions

Our setting takes inspiration from causal representation learning (CRL) [49] and the connections between CRL and CBMs described by Marconato et al. [39, 40]. We assume that the interpretable concepts are related to causal variables in an unknown data generating process.

As illustrated in Fig. 1 (right), we assume a causal system with latent causal variables $G = (G_1, \ldots, G_d) \in \mathcal{G} \subseteq \mathbb{R}^d$, which can potentially have causal relations between them. The *observation* is denoted by $X \in \mathcal{X} \subseteq \mathbb{R}^D$ and is generated by an unobserved invertible mixing function $f \colon \mathcal{G} \to \mathcal{X}$ as $X = f(G)$. The goal of CRL is to recover the causal variables by learning a function $g_\psi \colon \mathcal{X} \to \mathbb{R}^d$ that approximates $f^{-1}$. This requires specific assumptions and we cannot identify the ground-truth variables exactly, but only up to an equivalence class. A common notion of identifiability is up to a permutation, scaling and translation [19, 24, 31], which means that the learned representation satisfies $g_\psi(X) = P\Lambda G + b$ for a permutation matrix $P$, an invertible diagonal matrix $\Lambda$ and a

vector $b$. A more general notion is *identifiability up to permutation and element-wise transformations* in which, instead of a diagonal matrix, we consider a diffeomorphism, which we formalize as:

**Definition 3.1.** Let $\pi\colon \{1,\dots,d\} \to \{1,\dots,d\}$ be a permutation of the variable indices. Let $P \in \mathbb{R}^{d\times d}$ be the permutation matrix associated with $\pi$, meaning that $P_{i\pi(i)} = 1$ and 0 otherwise, and $T : \mathbb{R}^d \to \mathbb{R}^d$ a map. A representation $Z$ identifies the ground-truth causal variables up to a permutation and element-wise transformation if:

$$PT(Z) = \left[T_{\pi(1)}(Z_{\pi(1)}), \dots, T_{\pi(d)}(Z_{\pi(d)})\right]^\top = [G_1, \dots G_d]^\top = G$$

There are many more notions of identifiability in the literature, e.g., in some cases the causal variables can be multidimensional [33] or identified up to a *block*, i.e. a group of causal variables [2, 51, 55, 30]. In the main paper we focus mostly on the single-dimensional case in which we identify each individual variable, but we provide extensions to the multidimensional case in App. A.

We denote the learned causal variables by $M = (M_1, \dots, M_d) \in \mathbb{R}^d$. For simplicity of exposition, in the main paper we will assume that the interpretable concepts $C = (C_1, \dots, C_d)^\top \in \mathbb{R}^d$ correspond to the ground-truth causal variables $G_1, \dots, G_d$ up to permutation and element-wise transformations; in other words, $g_{\psi'}$ shown in Fig. 1 (right) also identifies ground-truth causal variables up to the same identifiability class. In App. A and Sec. 5 we extend this to allow each of the concepts to be a transformation of a group of causal variables. Our goal is to learn the alignment map $\alpha$ that transforms the learned representations $M$ to the concepts $C$ efficiently and accurately.

Note that in our theoretical results we assume that both concepts and causal variables are continuous-valued. Our motivation for this choice is technical: most current CRL methods assume that the underlying causal variables are continuous-valued, and only very few methods identify discrete-valued causal variables under very specific assumptions, e.g., in cases where different causal variables always affect different parts of an observation [29]. Since in concept-based models it is common to work with binary concepts, we also test these cases in our experiments and show that our estimators still provide good results, but we leave the extension of the theoretical results to discrete causal variables and concepts for future work.

For our theoretical analysis, we assume that we are given a function $g_\psi$ from a CRL method that correctly identifies the ground-truth causal variables up to some identifiability class and that the interpretable concepts correctly identify the causal variables up to the same class.

**Assumption 3.2.** Let $M = g_\psi(X)$ be the causal representations learned by a pretrained CRL method and let $C = g_{\psi'}(X)$ be the interpretable concepts annotated from the observation $X$. We assume that $M$ and $C$ identify $G$ up to the same identifiability class. This implies that

$$PT(M) = \left[T_{\pi(1)}(M_{\pi(1)}), \dots, T_{\pi(d)}(M_{\pi(d)})\right]^\top = [C_1, \dots C_d]^\top = C. \tag{1}$$

While this assumption is needed for our theoretical analysis, we will show that empirically our framework works even when the CRL methods do not fully identify the causal variables. Finding the alignment function $\alpha$ in Fig. 1 reduces to learning the permutation $\pi$ and a separate regression per concept $C_i$ to learn the transformation from learned causal representation $M_{\pi(i)}$ to interpretable concept $C_i$. The main difficulty here is identifying $\pi$ from observational data, i.e. without performing interventions, and with few samples. In the following we introduce two estimators for this setting, one assuming the element-wise transformation is linear, e.g. as is the case in some CRL methods like [20, 24], for which we will be able to provide finite sample results based on a tunable parameters, and a second, non-parametric method based on kernel methods that allows for arbitrary invertible element-wise transformations, and can hence be applied to most CRL methods. After recovering the interpretable concepts, we can use them as in CBMs as inputs to a label predictor for $Y$.

## 4 Linear Regression Alignment Learning with the Group Lasso

In this section, we describe a linear regression approach based on the Group Lasso to learn the permutation $\pi$ and transformation $T$ in (1). We will prove that this method simultaneously provides accurate regression estimates for $T$ and identifies $\pi$ correctly with high probability. To simplify the exposition, we focus here on the case of scalar variables. Proofs are in App. A, which also contains discussion of the assumptions, a pseudo-code description in Alg. 1 and a generalization to block variables. The proof combines techniques from high-dimensional statistics [10, 35].

**Method.** Linear regression can describe non-linear relations by transforming covariates using a feature map $\varphi : \mathbb{R} \to \mathbb{R}^p$. In this section, we assume that $T_i$ can be expressed as a linear function of $\varphi(M_{\pi(i)})$. The choice of $\varphi$ therefore gives precise control to trade off interpretability with expressive power for $T_i$. For instance, in the simplest and most easily interpretable case, $\varphi$ can be the identity function, so that $p = 1$ and $C_i$ and $M_{\pi(i)}$ are related by scaling. In more challenging settings, richer functional relations may be needed, e.g. splines or random Fourier features. We apply the same feature map to all machine variables in $M$, for which we write $\varphi(M) = [\varphi(M_1)^\top, \ldots, \varphi(M_d)^\top]^\top$. Then each $C_i$ is modeled as a linear function of the transformed variables:

$$C_i = \varphi(M)\beta_i^\star + \varepsilon_i, \tag{2}$$

where $\beta_i^\star \in \mathbb{R}^{pd}$ is an unknown parameter vector, and $\varepsilon_i \sim \mathcal{N}(0, \sigma^2)$ is Gaussian noise. By assumption, $C_i$ only depends on $M_{\pi(i)}$ and not on any of the other variables, so $\beta_i^\star$ is sparse: only the coefficients for $\varphi(M_{\pi(i)})$ are non-zero. To express this formally, let $G_j = \{(j-1)p, \ldots, jp\}$ be the indices that belong to variable $M_j$ and, for any $\beta \in \mathbb{R}^{pd}$, define $\beta^j = (\beta_k \mid k \in G_j)$ to be the corresponding coefficients. Then $(\beta_i^\star)^j$ is non-zero only for $j = \pi(i)$.

We assume we are given a data set $\mathcal{D} = \{(C^{(\ell)}, M^{(\ell)})\}_{\ell=1}^n$ that contains $n$ independent samples of corresponding pairs $C^{(\ell)} = (C_1^{(\ell)}, \ldots, C_d^{(\ell)})$ and $M^{(\ell)} = (M_1^{(\ell)}, \ldots, M_d^{(\ell)})$. We stack the $C^{(\ell)}$ into a matrix $\mathbf{C} \in \mathbb{R}^{n \times d}$ and the feature vectors $\varphi(M^{(\ell)})$ into $\Phi \in \mathbb{R}^{n \times pd}$. This leads to the relation

$$\mathbf{C}_i = \Phi\beta_i^\star + \varepsilon_i,$$

where $\mathbf{C}_i$ is the $i$-th column of $\mathbf{C}$ and the noise vector $\varepsilon_i$ consists of $n$ independently drawn $\mathcal{N}(0, \sigma^2)$ variables. To estimate $\beta_i^\star$, we use the Group Lasso with parameter $\lambda > 0$:

$$\widehat{\beta}_i = \arg\min_{\beta \in \mathbb{R}^{dp}} \frac{1}{n}\|\mathbf{C}_i - \Phi\beta\|^2 + \lambda\sqrt{p}\|\beta\|_{2,1}, \qquad \|\beta\|_{2,1} = \sum_{j=1}^d \|\beta^j\|. \tag{3}$$

The $(2,1)$-mix norm $\|\beta\|_{2,1}$ in (3) encourages group-wise sparsity. It applies the Euclidean norm $\|\beta^j\|$ to each group $j$ separately, and sums the results over groups, as defined below. We also define the $(2,\infty)$-mix norm as $\|\beta\|_{2,\infty} = \max_{j=1,\ldots,d} \|\beta^j\|$

**Theoretical Analysis.** We denote the full covariance matrix by $\widehat{\Sigma} = \frac{1}{n}\Phi^\top\Phi$. For the group of $p$ columns of $\Phi$ that correspond to $\varphi(M_j)$ we write $\Phi_j = \Phi_{G_j}$. Also let $\widehat{\Sigma}_{jj'} = \frac{1}{n}\Phi_j^\top\Phi_j'$ denote the covariance matrix between groups $j$ and $j'$, and abbreviate $\widehat{\Sigma}_j = \widehat{\Sigma}_{jj}$. Then, w.l.o.g., we can assume that the data within each group have been centered and decorrelated:

$$\frac{1}{n}\mathbb{1}^\top\Phi_j = 0 \quad \text{and} \qquad \widehat{\Sigma}_j = I \qquad \text{for all } j = 1, \ldots, d. \tag{4}$$

This can be achieved by pre-processing: subtract the empirical mean of $\Phi_j$ and multiply it from the right by the inverse square root of the empirical covariance matrix. Preprocessing is allowed in our theoretical results, because they apply to the fixed design setting, so probabilities refer to the randomness in $\mathbf{C}$ conditional on already having observed $\Phi$. If $\varphi(M_j)$ and $\varphi(M_{j'})$ are completely correlated, then $\beta_i^\star$ is not uniquely identifiable, no matter how much data we have. To rule out this possibility, we make the following assumption, which limits the amount of correlation. This is a standard assumption when analyzing the Group Lasso [35, 34].

**Assumption 4.1.** There exists $a > 1$ s.t. for all $j \neq j'$,

$$\max_{t \in \{1,\ldots,p\}} |(\widehat{\Sigma}_{jj'})_{tt}| \leq \frac{1}{14a}, \quad \max_{t,t' \in \{1,\ldots,p\}} |(\widehat{\Sigma}_{jj'})_{tt'}| \leq \frac{1}{14ap}.$$

**Theorem 4.2.** *Suppose the data have been pre-processed to satisfy* (4) *and let Assump.* (4.1) *hold. Take any $\delta \in (0,1)$ and set $\lambda \geq 4\lambda_0$, where*

$$\lambda_0 = \frac{2\sigma}{\sqrt{n}}\sqrt{1 + \sqrt{\frac{8\log(d/\delta))}{p}} + \frac{8\log(d/\delta)}{p}},$$

*and set $c = \left(1 + \frac{24}{7(a-1)}\right)$. Then, any solution $\widehat{\beta}_i$ of the Group Lasso objective* (3) *satisfies*

$$\|\widehat{\beta}_i - \beta_i^\star\|_{2,\infty} \leq c\lambda\sqrt{p} \tag{5}$$

with probability at least $1 - \frac{\delta}{d}$. If, in addition, $\|(\beta_i^\star)^{\pi(i)}\| > 2c\lambda\sqrt{p}$, then (5) implies that $\widehat{J}_i = \arg\max_{j=1,\ldots,d}\|\widehat{\beta}_i^j\|$ estimates $\pi(i)$ correctly.

Theorem 4.2 gives us an explicit relation between the parameters $n, p, d, \delta$ of the learning task, the tuning of the hyperparameter $\lambda$, and the estimation errors for $\beta_i^\star$ and $\pi(i)$. For example, if we set $\delta = \frac{1}{n}$, $\lambda = 4\lambda_0$ and let $n \to \infty$, then $\lambda \to 0$ and $\widehat{J}_i$ estimates the correct index $\pi(i)$ with probability tending to 1. So, regardless of the true parameter magnitude $\|(\beta_i^\star)^{\pi(i)}\|$, the estimator is consistent given a sufficient amount of data. Another way to express this is to ask about *sample complexity*: which sample size $n$ do we need to reach accuracy $E > 0$? Setting $\lambda = 4\lambda_0$ and solving for $n$ large enough that $c\lambda\sqrt{p} \le E$, we see that

$$n \ge \frac{64c^2\sigma^2\big(p + \sqrt{8p\log(d/\delta)} + 8\log(d/\delta)\big)}{E^2}$$

is sufficient. For estimating the permutation $\pi(i)$ correctly, the required accuracy is $E \le \|(\beta_i^\star)^{\pi(i)}\|/2$, so the larger the true parameters, the easier this task becomes.

The estimation is performed separately for each concept $C_i$, and, if $\widehat{J}_i$ is correct for all $i$ simultaneously, we can construct a valid estimate of the permutation by $\widetilde{\pi}(i) = \widehat{J}_i$. However, this estimate is not robust to estimation errors and may even produce functions $\widetilde{\pi}$ that are not permutations if some $\widehat{J}_i$ are incorrect. The actual estimator of the permutation, $\widehat{\pi}$, therefore optimizes a weighted matching problem, which leads to the same estimate as $\widetilde{\pi}$ if the $\widehat{J}_i$ together produce a valid permutation, but forces $\widehat{\pi}$ to be a valid permutation even if they do not:

$$\widehat{\pi} = \arg\max_{\pi \in \Pi} \sum_{i=1}^d \|\widehat{\beta}_i^{\pi(i)}\|. \tag{6}$$

Here, $\Pi$ is the set of all permutations. This assignment can be solved without cycling through all permutations, with cubic runtime in the dimension $d$. By a union bound over $i$, it follows from Theorem 4.2 that $\widehat{\pi}$ estimates the true permutation $\pi$ correctly with high probability:

**Corollary 4.3.** *Assume the same setting as Theorem 4.2 such that for each $i = 1, \ldots, d$, $\|(\beta_i^\star)^{\pi(i)}\| > 2c\lambda\sqrt{p}$ and consider the estimator $\widehat{\pi}$ as defined in (6). Then $\widehat{\pi} = \pi$ with probability at least $1 - \delta$.*

## 5 Kernelized Alignment Learning

The previous section describes how to learn functions with finite-dimensional representations. We now extend the estimator to use general functions from a *reproducing kernel Hilbert space* (RKHS) [18]. This may be interpreted as a (typically infinite-dimensional) feature map $\varphi$ that maps to the RKHS. However, using a representer theorem, all computations can be performed on finite-dimensional representations. We summarize the method in Alg. 2 in Appendix B.

**Method.** Define again $M = (M_1, \ldots, M_d)$, where we now allow each machine variable $M_j$ to take values in an abstract space $\mathcal{Z}_j$. Let $\mathcal{Z} = \mathcal{Z}_1 \times \ldots \mathcal{Z}_d$. We then generalize (2) to

$$C_i = \beta_i^\star(M) + \varepsilon_i,$$

where $\beta_i^\star \in \mathcal{H}$ is a function from $\mathcal{Z}$ to $\mathbb{R}$, and $\varepsilon_i \sim \mathcal{N}(0, \sigma^2)$. The space of possible functions $\mathcal{H}$ will be an RKHS containing functions of the form $\beta(M) = \sum_{j=1}^d \beta^j(M_j)$, where each $\beta^j$ is an element of an RKHS $\mathcal{H}_j$ that captures the effect of variable $M_j$ on $C_i$. The assumption that $C_i$ depends only on $M_{\pi(i)}$ means that $\beta_i^\star(M) = (\beta_i^\star)^{\pi(i)}(M_{\pi(i)})$. Each $\mathcal{H}_j$ can be freely chosen, and is typically specified indirectly by the choice of a positive definite *kernel* $\kappa_j : \mathcal{Z}_j \times \mathcal{Z}_j \to \mathbb{R}$ [18]. This kernel defines a measure of similarity between inputs: $\kappa_j(M_j, M_j') = \langle \varphi_j(M_j), \varphi_j(M_j') \rangle_{\mathcal{H}_j}$, where $\varphi_j : \mathcal{Z}_j \to \mathcal{H}_j$ is the corresponding feature map. See p. 34 for examples.

Given data $\mathcal{D} = \{(C^{(\ell)}, M^{(\ell)})\}_{\ell=1}^n$, let $\mathbf{M} \in \mathcal{Z}^n$ denote the matrix with the machine variables $(M^{(\ell)})^\top$ stacked as rows. If we further define $\beta(\mathbf{M}) = [\beta(M^{(1)}), \ldots, \beta(M^{(\ell)})]^\top$, then the Group Lasso objective (3) generalizes to

$$\widehat{\beta}_i = \arg\min_{\beta \in \mathcal{H}} \frac{1}{n}\|\mathbf{C}_i - \beta(\mathbf{M})\|^2 + \lambda \sum_{j=1}^d \|\beta^j\|_{\mathcal{H}_j}, \tag{7}$$

where $\|\beta^j\|_{\mathcal{H}_j}$ is the norm associated with $\mathcal{H}_j$. To optimize the objective in (7), we need a finite dimensional objective to give to a Group Lasso solver. We provide a version of the Representer Theorem showing that the solution of (7) lives in a subspace of $\mathcal{H}$ that can be described by finite-dimensional parameters $\hat{c}_i^1, \ldots, \hat{c}_i^j \in \mathbb{R}^n$:

**Theorem 5.1.** *Let $\varphi_1, \ldots, \varphi_d$ be the feature maps associated with $\mathcal{H}_1, \ldots, \mathcal{H}_d$. Then there exist $\hat{c}_i^1, \ldots, \hat{c}_i^d \in \mathbb{R}^n$ such that the optimization problem in (7) has solution $\widehat{\beta}_i$ with each $\widehat{\beta}_i^j$ of the form*

$$\widehat{\beta}_i^j = \sum_{\ell=1}^{n} \varphi_j(M_j^{(\ell)})(\hat{c}_i^j)_\ell.$$

Substitution of this form into (7) gives that $\hat{c}_i^1, \ldots, \hat{c}_i^d$ will be the minimizers of the following finite-dimensional optimization problem:

$$\min_{c^1, \ldots, c^d \in \mathbb{R}^n} \frac{1}{n} \|\mathbf{C}_i - \sum_{j=1}^{d} K_j c^j\|^2 + \lambda \sum_{j=1}^{d} \|c^j\|_{K_j}, \tag{8}$$

where $K_j \in \mathbb{R}^{n \times n}$ with $(K_j)_{\ell k} = \kappa_j(M_j^{(\ell)}, M_j^{(k)})$, and $\|c^j\|_{K_j} = \sqrt{c^{j\top} K_j c^j}$.
This procedure is performed for $i = 1, \ldots d$. The permutation is estimated as in the linear case:

$$\widehat{\pi} = \arg\max_{\pi \in \Pi} \sum_{i=1}^{d} \|\hat{c}_i^{\pi(i)}\|_{K_{\pi(i)}}.$$

**Theoretical Analysis.** Using a result by Bach [4], we prove that our estimator for $\pi$ is consistent under suitable conditions, discussed in App. B. This holds for random design, so for the joint randomness of $\mathcal{D}$.

**Theorem 5.2.** *Assume (A–D) in App. B.2. Then, for any sequence of regularization parameters $\lambda_n$ such that $\lambda_n \to 0$ and $\sqrt{n}\lambda_n \to +\infty$ when $n \to \infty$, the estimated permutation $\widehat{\pi}$ converges in probability to $\pi$.*

**Implementation.** To use a standard Group Lasso solver we need to reparametrize the optimization problem in (8), because of the scaled norms $\|\cdot\|_{K_j}$. We can do this with a Cholesky decomposition:

**Lemma 5.3.** *For each $j = 1, \ldots, d$ let $L_j$ be the Cholesky decomposition of the Gramm matrix $K_j$, then $\hat{c}_i^j = (L_j^\top)^{-1}\hat{\gamma}_i^j$ if the parameters $\hat{\gamma}_i^1, \ldots, \hat{\gamma}_i^d$ are minimizers of*

$$\min_{\gamma^1, \ldots, \gamma^d \in \mathbb{R}^n} \frac{1}{n} \|\mathbf{C}_i - \sum_{j=1}^{d} L_j \gamma^j\|^2 + \lambda \|\gamma^j\|_{2,1}. \tag{9}$$

## 6 Experiments

We perform two types of experiments on four different datasets to evaluate our estimators. In the first type of experiment, we evaluate how well the estimator can learn the permutations and recover the original concepts, which can be either continuous or binary. In the second type of experiment we assess the usefulness of our estimator to perform a downstream classification task, where the target label is binary. For this task, concepts are binarized by setting the concept to $0$, if the value is lower than the midpoint of the range of that concept, and $1$ otherwise. The label is randomly generated by selecting a sub-selection of concepts and checking if some of those columns are $1$. This creates a random classification task for each seed. While the theoretical guarantees for our estimator do not translate directly to a setting where the concepts are binary, we can use the logistic Group Lasso [43] and we see empirically that this variant also performs well.

The first dataset, called "Toy dataset", is synthetic. Here the concepts are generated using either a linear combination of features (which we call the wellspecified case) or diffeomorphisms (which we call the misspecified case) of the representations. The concepts are then permuted. We evaluate on datasets common in CRL: Action and Temporal Sparsity [31] and Temporal Causal3Dident [33]. We train several CRL methods as a first step of our pipeline: DMS-VAE [31], CITRIS-VAE [33], iVAE [24] and TCVAE [11]. In the downstream task experiments, we compare to CBM [28], CEM [57] and HardCBM [17]. Details are in App. D.

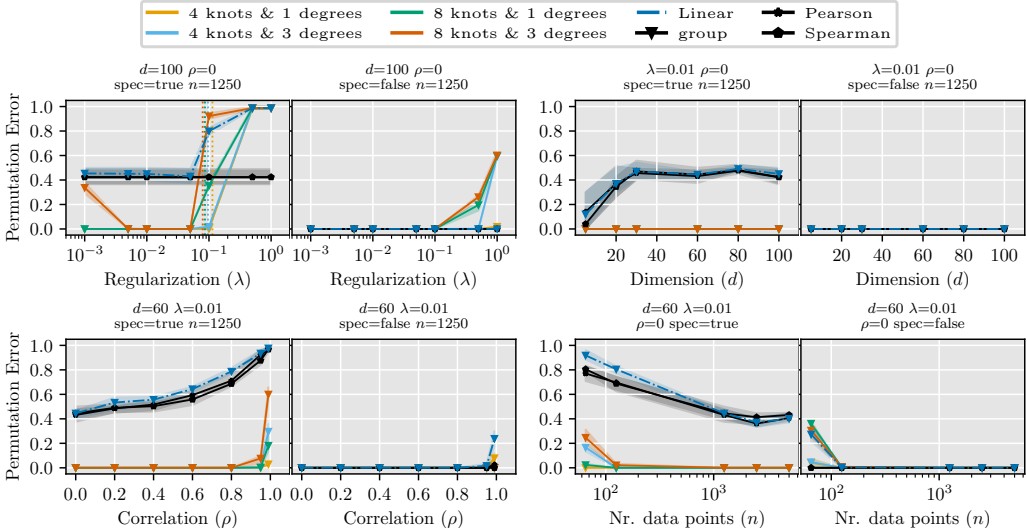

Figure 2: Permutation error rate for spline features. From top left to bottom right we vary: (i) the regularization parameter, (ii) the number of dimensions, (iii) the correlation of the variables and (iv) the number of labels. The first plot of each pair shows the wellspecified and the second the misspecified case. We average over 10 seeds and shade the 25-75th percentile.

**Performance Metrics.** To assess our estimator in terms of concept reconstruction capabilities we report the mean error in the learned permutation of the variables: MPE $= \frac{1}{d} \sum_{i=1}^{d} \mathbb{1}\{\widetilde{\pi}(i) \neq \pi(i)\}$. In the experiments where we test its capabilities for downstream tasks such as classification, we report the accuracy on the final label and the Oracle Impurity Score (OIS) [58]. The OIS measures how much information of each concept is contained in other concepts. For concept interpretability, it is desirable that the OIS should be as small as possible. We report an extended discussion of our results in Appendix E, with additional metrics, e.g. the NIS [57], and computation times.

**Toy Dataset.** The synthetic data experiments consist of 4 settings, each using a different set of features to perform the regression: linear, splines, random Fourier features (RFF) and kernels. The $M$ variables are distributed according to $\mathcal{N}(0, (1-\rho)I_{d \times d} + \rho\mathbb{1})$, where $\mathbb{1}$ denotes a matrix filled with only 1's. The $\rho \in (0, 1)$ parameter controls the amount of correlation between the marginal variables. We sample $n$ data points, on which we perform a $80/20$ train/test data split. In the wellspecified case, we generate the $M$ variables using the features from that setting. For each dimension $j = 1, \ldots, d$, we draw a random weight vector $\beta_j^\star \in \mathbb{R}^p$, such that $\|\beta_j^\star\| \in [16\lambda_0, 32\lambda_0]$ uniformly. A permutation $\pi \colon \{1, \ldots, d\} \to \{1, \ldots, d\}$ is sampled uniformly from all possible permutations. Finally, with independent $\varepsilon_i \sim N(0, \sigma^2)$ noise variables, we get $C_i = \varphi(M_{\pi(i)})^\top \beta_{\pi(i)}^\star + \varepsilon_i$. In the misspecified case, the $M$ variables are still sampled the same as before, but the $C$ variables are generated by sampling a diffeomorphism for each dimension. These outcomes are then permuted using a random permutation again. We cover a large range of values of $\rho, d, n$ and $\lambda$ as shown in App. D.1.

**Action/Temporal Sparsity Datasets.** We use the two synthetic datasets from [31] that represent the action and temporal sparsity settings in a time series. These settings have 10 causal variables $z_1, \ldots, z_{10} \in [-2, 2]$ with a causal structure. The mixing function is an invertible neural network with Gaussian random weights, after which the columns in the linear layers are orthogonalized to ensure injectivity. To recover the ground-truth permutation, we follow Lachapelle et al. [31] and use the test set to calculate a permutation based on Pearson correlations.

**Temporal Causal3DIdent.** We evaluate our methods on an image benchmark, Temporal-Causal3DIdent [33]. The dataset consists of images of 3D objects, rendered in different positions, rotation and lighting. The causal variables are the position $(x, y, z) \in [-2, 2]^2$, the object rotation with two dimensions $[\alpha, \beta] \in [0, 2\pi)^2$, the hue, the background and spotlight in $[0, 2\pi)$. The object

Table 1: Label Accuracy and the OIS-metric on our downstream task. The $(n)$ indicates the number of train and test points used in each column. We averaged the results over 10 seeds and report the mean and standard deviation. The best result for each $n$ is written in **bold**.

| Model | Method | Label Acc. ↑ $(n)$ | | | | OIS ↓ $(n)$ | | | |
|---|---|---|---|---|---|---|---|---|---|
| | | 20 | 100 | 1000 | 10000 | 20 | 100 | 1000 | 10000 |
| **Action Sparsity Dataset** | | | | | | | | | |
| DMS-VAE | Linear | **0.77 ± 0.03** | 0.81 ± 0.01 | 0.83 ± 0.01 | 0.84 ± 0.01 | **0.63 ± 0.01** | 0.40 ± 0.00 | 0.15 ± 0.00 | 0.12 ± 0.00 |
| | Spline | 0.72 ± 0.03 | **0.83 ± 0.02** | **0.85 ± 0.01** | 0.86 ± 0.01 | 0.63 ± 0.01 | 0.40 ± 0.00 | 0.15 ± 0.00 | 0.12 ± 0.00 |
| | RFF | 0.77 ± 0.02 | 0.82 ± 0.02 | 0.84 ± 0.01 | 0.85 ± 0.01 | 0.64 ± 0.01 | **0.39 ± 0.00** | 0.14 ± 0.00 | 0.12 ± 0.00 |
| iVAE | Linear | 0.73 ± 0.03 | 0.79 ± 0.02 | 0.81 ± 0.01 | 0.83 ± 0.01 | 0.63 ± 0.01 | 0.40 ± 0.00 | 0.29 ± 0.00 | 0.26 ± 0.00 |
| | Spline | 0.69 ± 0.02 | 0.76 ± 0.02 | 0.81 ± 0.01 | 0.83 ± 0.01 | 0.63 ± 0.01 | 0.40 ± 0.00 | 0.28 ± 0.00 | 0.26 ± 0.00 |
| | RFF | 0.59 ± 0.04 | 0.70 ± 0.02 | 0.78 ± 0.01 | 0.81 ± 0.01 | 0.64 ± 0.01 | 0.39 ± 0.00 | 0.28 ± 0.00 | 0.26 ± 0.00 |
| CBM [28] | | 0.60 ± 0.03 | 0.60 ± 0.02 | 0.73 ± 0.01 | 0.88 ± 0.01 | 0.92 ± 0.01 | 0.43 ± 0.01 | **0.11 ± 0.00** | 0.07 ± 0.00 |
| CEM [57] | | 0.66 ± 0.02 | 0.69 ± 0.03 | 0.82 ± 0.01 | 0.88 ± 0.01 | 0.90 ± 0.03 | 0.47 ± 0.01 | 0.40 ± 0.01 | 0.57 ± 0.01 |
| HardCBM [17] | | 0.56 ± 0.03 | 0.61 ± 0.01 | 0.68 ± 0.01 | **0.89 ± 0.00** | 0.92 ± 0.02 | 0.46 ± 0.01 | 0.12 ± 0.00 | **0.06 ± 0.00** |
| **Temporal Causal3DIdent Dataset** | | | | | | | | | |
| CITRISVAE | Linear | 0.74 ± 0.06 | 0.75 ± 0.06 | 0.80 ± 0.04 | 0.81 ± 0.04 | 0.69 ± 0.02 | 0.44 ± 0.01 | 0.19 ± 0.00 | 0.16 ± 0.00 |
| | Spline | 0.74 ± 0.06 | **0.80 ± 0.04** | **0.82 ± 0.03** | **0.83 ± 0.03** | 0.69 ± 0.02 | 0.44 ± 0.01 | **0.13 ± 0.00** | **0.09 ± 0.00** |
| | RFF | 0.70 ± 0.06 | 0.76 ± 0.05 | 0.79 ± 0.04 | 0.81 ± 0.04 | **0.65 ± 0.01** | 0.43 ± 0.00 | 0.16 ± 0.01 | 0.13 ± 0.01 |
| iVAE | Linear | **0.74 ± 0.06** | 0.76 ± 0.05 | 0.78 ± 0.05 | 0.80 ± 0.04 | 0.69 ± 0.02 | 0.44 ± 0.01 | 0.17 ± 0.00 | 0.14 ± 0.00 |
| | Spline | 0.73 ± 0.06 | 0.78 ± 0.04 | 0.79 ± 0.04 | 0.81 ± 0.04 | 0.69 ± 0.02 | 0.43 ± 0.04 | 0.17 ± 0.00 | 0.13 ± 0.00 |
| | RFF | 0.69 ± 0.06 | 0.75 ± 0.05 | 0.78 ± 0.04 | 0.80 ± 0.04 | 0.65 ± 0.01 | **0.42 ± 0.02** | 0.16 ± 0.00 | 0.13 ± 0.00 |
| CBM [28] | | 0.57 ± 0.02 | 0.53 ± 0.03 | 0.68 ± 0.04 | 0.78 ± 0.05 | 1.00 ± 0.03 | 0.48 ± 0.02 | 0.25 ± 0.00 | 0.18 ± 0.00 |
| CEM [57] | | 0.64 ± 0.04 | 0.67 ± 0.03 | 0.72 ± 0.05 | 0.78 ± 0.05 | 0.97 ± 0.05 | 0.51 ± 0.02 | 0.36 ± 0.01 | 0.49 ± 0.01 |
| HardCBM [17] | | 0.57 ± 0.05 | 0.53 ± 0.02 | 0.63 ± 0.03 | 0.77 ± 0.05 | 0.96 ± 0.03 | 0.47 ± 0.02 | 0.24 ± 0.00 | 0.16 ± 0.00 |

shape is a categorical variable. We use a pretrained CITRIS-VAE encoder, which outputs a 32 dimensional latent space and a grouping of which dimensions relate to which causal variables. Although CITRIS-VAE provides the correct permutation of the groups, we ignore it and perform a random permutation on the variables. We train an MLP for each of the 32 dimensions that predicts all causal variables. Based on the $R^2$ scores of these regressions, we learn the group assignments.

**Results.** We show a set of representative results in Fig. 2 for the Toy Dataset, where we compare as baselines to the permutations one can learn with Pearson or Spearman correlation, and a linear sum assignment. Our estimator is able to reconstruct the correct permutation perfectly with only a small number of features. It performs well for a broad range of regularization parameters. In the wellspecified case we calculate the $\lambda_0$ parameter and see that the estimator performs well around this value. The dimension dependence is almost negligible, which was predicted by the dimension appearing only in the log factor in Thm. 4.2. The estimator works well with few data points and even better than theory predicts, as it has a low error even with high correlation, as shown in Fig. 2. Spearman correlation also performs well in some simple settings, which suggests that the relation is indeed monotonic in these settings. On the other hand, there are no theoretical guarantees for this baseline and it cannot be used for multi-dimensional representations, for which our estimator still provides a principled approach. We report results with similar trends for RFF and the kernelized approach in App. E.1. To test the robustness of our estimation to the case in which CRL methods do not provide a complete, but only partial disentanglement, we also show an ablation where we provide mixtures of pairs of causal variables as input to our algorithm, showing that it still recovers the correct permutation for each pair. Similar trends apply also to the image dataset, as reported in App. E.2.

A selection of results for the downstream classification task for the action sparsity dataset and temporal Causal3DIdent dataset are in Tab. 1. Our estimator consistently scores high in terms of Label Acc. and OIS, while requiring less labels. For image data, our estimator beats the baseline for every number of datapoints. In App. E.2 we report the complete results, which show that the baselines perform well with regards to the mean concept accuracy in the Action/Temporal sparsity datasets, but our estimator still performs better with the smallest number of labels and with less computation power to calculate. Our estimators consistently achieve a perfect NIS [58] score, which measures the impurities distributed across the concept representations.

Finally, in Fig. 3 we show the execution times of our estimator compared to several baseline methods. We report the times in both the continuous experiments and binary classification experiments conducted with the Temporal Causal3DIdent dataset. The baseline in the continuous setting is given

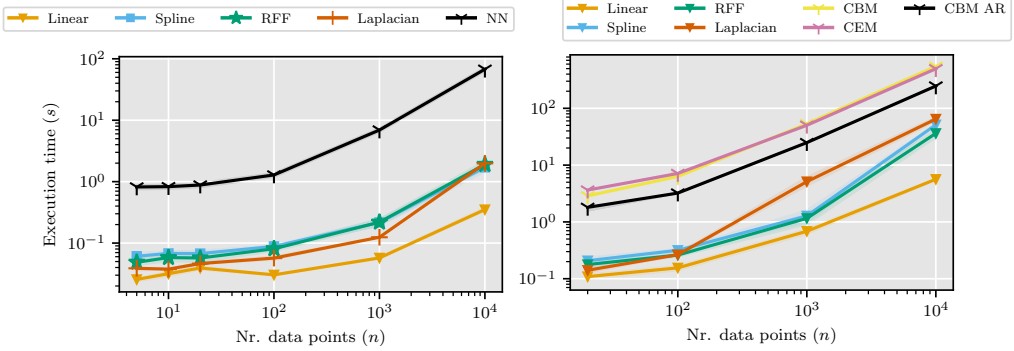

Figure 3: Execution times for our estimators and several baseline models on the Temporal Causal3DIdent dataset. **Left:** The causal variables are continuous-valued. The baseline is given by training several neural networks and the matching is based on the $R^2$-scores. We report the time needed to estimate the matching and train the neural networks. **Right:** The causal variables are binarized and a downstream classification task is added. We report the time needed to estimate the matching and learn the classification task or the time needed to train the concept-based models.

by training a neural network for each causal variable. Each neural network learns how to predict the causal variable from the learned encodings, and the $R^2$-scores are used to construct a matching. The time that is reported is the time that is needed to estimate the matching and the training, if needed. In the binary classification experiments, the baselines are given by the CBM, CEM, and HardCBM models. The time that is reported is the time needed to either estimate the matching and learn the classification or to train one of the concept-based models. We see that all versions of our estimator require significantly less computation time than the baselines.

## 7 Conclusions and Discussion

We propose a framework that provides theoretical guarantees on learning of concepts by leveraging causal representation learning (CRL). We provide two estimators that are able to learn the alignment between the learned representations and the concepts: a linear estimator with finite sample guarantees and a non-parametric kernelized estimator with asymptotic guarantees. We test our methods on CRL benchmarks and show they perform even better than the theory predicted.

While our work proposes a promising research direction, several limitations remain. For example, we assume that human concepts coincide with the causal variables recovered by the CRL methods, which limits the framework's applicability. While our framework can also consider settings in which only blocks of causal variables are identifiable, as opposed to each individual variable, then the alignment can be only learned between these blocks and the corresponding blocks of concepts. To generalize our framework to different levels of coarse-grained human concepts, it would be interesting to incorporate ideas from *causal abstraction* [47, 13, 7] to extend this analysis to cases in which human concepts are abstractions of the underlying causal system. Moreover, while in the current framework we focus on CRL methods in order to get theoretical guarantees on the identifiability of latent variables from high-dimensional observations, we do not use any causal semantics, so in principle our framework could be applied also to other types of methods that provide similar theoretical guarantees.

A theoretical limitation of our analysis is the assumption of low correlation between the learned representations. In our experiments we find that our methods still work with significantly stronger correlations than assumed in our theoretical results, so we believe there may be room here to strengthen the theory. This might also allow us to relax our assumption that CRL methods achieve perfect disentanglement of the causal variables, which is not realistic in many domains. Finally, although we consider discrete concepts in our evaluation, we do not provide a theoretical analysis for them, because current CRL methods cannot identify discrete variables, or can only do so in specific settings [29]. A possible way to extend our theoretical results to this setting would be to consider previous results for the Group Lasso for logistic regression [43].

## Acknowledgments and Disclosure of Funding

We thank SURFsara for the support in using the Snellius Compute Cluster and Robert Jan Schlimbach for the technical support in parallelizing most of the experiments. Van Erven was supported by the Netherlands Organization for Scientific Research (NWO) under grant numbers VI.Vidi.192.095 and https://doi.org/10.61686/OWREN85146.

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

# A    Proofs for Linear Regression Alignment Learning

In this section we provide proof details and the algorithm for Section 4. As discussed there, we extend Theorem 4.2 to a more general result Theorem A.5 that allows for blocks of causal variables corresponding to a single concept. To state the general result we will redefine our model and introduce additional notation. The variables $C_i \in \mathbb{R}$ and $M_j \in \mathbb{R}^{k_j}$ now live in potentially different spaces. Let $C = (C_1, \ldots, C_d)^\top \in \mathbb{R}^d$ and $M = (M_1, \ldots, M_d)^\top \in \mathbb{R}^k$, where $k = \sum_{j=1}^d k_j \geq d$. Let $K_j = \{\sum_{a=1}^{j-1} k_a + b \mid b \in \{1, \ldots, k_j\}\}$ denote the subset of indices in $M$ that correspond to the block $M_j$. The permutation that we want to recover is $\pi$. Each dimension in the $M$ variable can be transformed through a separate feature map $\varphi_t \colon \mathbb{R} \to \mathbb{R}^{p_t}$ that can be different for each $t = 1, \ldots, k$. We will denote the total feature vector by

$$\varphi(M) = \begin{bmatrix} \varphi_1(M_1) \\ \vdots \\ \varphi_k(M_k) \end{bmatrix}.$$

The grouped features will be denoted by $\varphi(M)^j = (\varphi_t(M_t) \mid t \in K_j)$. Define the average feature set size as $\overline{p} = \frac{1}{k} \sum_{t=1}^k p_t$. The model is described by

$$C_i = \varphi(M)\beta_i^\star + \varepsilon_i, \quad \beta_i^\star \in \mathbb{R}^{k\overline{p}}, \varepsilon_i \sim \mathcal{N}(0, \sigma^2).$$

For the actual regression task, we can define data matrices again. The matrix $\mathbf{C}$ will be defined as in the main text and $\Phi$ now becomes an $n \times k\overline{p}$ matrix, in which all feature vectors $\varphi(M^{(\ell)})$ are stacked. With $\varepsilon_i$ denoting $n$ independently draw $\mathcal{N}(0, \sigma^2)$ variables, this results in the relation

$$\mathbf{C}_i = \Phi\beta_i^\star + \boldsymbol{\varepsilon_i}.$$

The $\beta_i^\star$ again has a sparse structure, because only the parameters corresponding to $\varphi(M)^j$ should be non-zero. Let the indices of these parameters be denoted by $G_j$. Alternatively, this $G_j$ is defined through $\varphi(M)_{G_j} = \varphi(M)^j$. Thus, in this setting we again have $d$ groups all denoted by $G_j$. To ease notation we set again $\beta_i^j = ((\beta_i)_t \mid t \in G_j)$. The definitions of the norm $\| \cdot \|_{2,\infty}$ and covariance matrices, $\widehat{\Sigma}_{jj'} = \frac{1}{n}\varphi(M)_{G_j}^\top \varphi(M)_{G_{j'}} = \frac{1}{n}\Phi_j^\top \Phi_{j'}$, are altered in accordance with these groups. As the groups can now be of different size, we have to change the definition of the $\| \cdot \|_{2,1}$-norm to take the different group sizes into account,

$$\|\beta\|_{2,1} = \sum_{j=1}^d \|\beta^j\|\sqrt{p^j},$$

where $p^j = \sum_{t \in K_j} p_t$. The loss function that we want to optimize to estimate $\beta_i^\star$ has the same form as before

$$\widehat{\beta}_i = \underset{\beta \in \mathbb{R}^{k\overline{p}}}{\arg\min} \frac{1}{n}\|\mathbf{C}_i - \Phi\beta\|^2 + \lambda\|\beta\|_{2,1} = \underset{\beta \in \mathbb{R}^{k\overline{p}}}{\arg\min} \frac{1}{n}\|\mathbf{C}_i - \Phi\beta\|^2 + \lambda\sum_{j=1}^d \|\beta^j\|\sqrt{p^j}. \quad (10)$$

---

**Algorithm 1** Estimating the permutation using linear regression with Group Lasso regularization

---

1: Input: regularization parameter $\lambda > 0$
2: Data: $\{(C^{(\ell)}, M^{(\ell)})\}_{\ell=1}^n$
3: **for** $i = 1, \ldots, d$ **do**
4: $\quad \widehat{\beta}_i \leftarrow \underset{\beta \in \mathbb{R}^{dp}}{\arg\min} \|\mathbf{C}_i - \Phi\beta\|^2 + \lambda\sqrt{p}\|\beta\|_{2,1}$
5: **end for**
6: $\widehat{\pi} \leftarrow \underset{\pi \in \Pi}{\arg\max} \sum_{i=1}^d \|\widehat{\beta}_i^{\pi(i)}\|$

---

The optimality conditions for any solution, $\widehat{\beta}_i$, for this convex optimization problem are given by

$$\frac{1}{n}(\Phi^\top(\mathbf{C}_i - \Phi\widehat{\beta}_i)^j = \frac{\lambda\sqrt{p^j}}{2}\frac{\widehat{\beta^j}}{\|\widehat{\beta^j}\|} \qquad\qquad \text{if } \widehat{\beta}_i^j \neq 0, \tag{11}$$

$$\frac{1}{n}\|\Phi^\top(\mathbf{C}_i - \Phi\widehat{\beta}_i)\|^2 \leq \frac{\lambda\sqrt{p^j}}{2} \qquad\qquad \text{if } \widehat{\beta}_i^j = 0. \tag{12}$$

To ensure that our results hold even in the case where $n < pd$, we introduce a standard assumption on the data from the high-dimensional statistics literature. Intuitively, this assumption ensures that the data is "variable enough" in the directions that matter.

**Assumption A.1.** The Restricted Eigen Value (RE(1)) is satisfied by the data matrix $\Phi \in \mathbb{R}^{n \times k\overline{p}}$ if there exists a $\kappa > 0$ such that for all $\Delta \in \mathbb{R}^{k\overline{p}} \setminus \{0\}$ and $j = 1, \ldots, d$ with $\sum_{i\neq j}\|\Delta^i\|\sqrt{p^i} \leq 3\|\Delta^j\|\sqrt{p^j}$ it holds that

$$\frac{\|\Phi\Delta\|}{\sqrt{n}\|\Delta^j\|} \mid \geq \kappa.$$

This property is satisfied for any $\Delta \in \mathbb{R}^{k\overline{p}} \setminus \{0\}$ if $\widehat{\Sigma} = \frac{1}{n}\Phi^\top\Phi$ has a positive minimal eigenvalue. Let $\lambda_{\min} > 0$ be the minimal eigenvalue of $\widehat{\Sigma}$, then

$$\|\Phi\Delta\|^2 = \Delta^\top\Phi^T\Phi\Delta = n\Delta^\top\widehat{\Sigma}\Delta \geq n\lambda_{\min}\Delta^\top\Delta = n\lambda_{\min}\|\Delta\|^2.$$

Now divide by $n$ and take the square root on both sides. This gives us

$$\frac{\|\Phi\Delta\|}{\sqrt{n}} \geq \sqrt{\lambda_{\min}}\|\Delta\| = \sqrt{\lambda_{\min}}\sqrt{\sum_{i=1}^d\|\Delta^i\|^2} \geq \sqrt{\frac{\lambda_{\min}}{d}}\sum_{i=1}^d\|\Delta^i\| \geq \sqrt{\frac{\lambda_{\min}}{d}}\|\Delta^j\|.$$

The second inequality follows from an application of Jensen's inequality. Dividing both sides by $\|\Delta^j\|$ gives the desired result.

The matrix $\widehat{\Sigma}$ is the empirical covariance matrix and will be positive definite almost surely whenever $n \geq k\overline{p}$ and hence RE(1) will be satisfied if $n \geq k\overline{p}$.

Finally, define $p_{\min} = \min_{j=1,\ldots d} p^j$ and $p_{\max} = \max_{j=1,\ldots,d} p^j$. The following theorems and proofs are adapted from Chapter 8 in Bühlmann and Van De Geer [10] and sections 3 and 5 in Lounici et al. [35].

**Theorem A.2.** *Assume that for all $\ell = 1, \ldots, n$, $\varepsilon_i^{(\ell)} \sim \mathcal{N}(0, \sigma^2)$ independently, $\sigma^2 > 0$, the RE(1) condition is satisfied with $\kappa > 0$ and consider the Group Lasso estimator*

$$\widehat{\beta}_i = \arg\min_{\beta \in \mathbb{R}^{k\overline{p}}} \frac{1}{n}\|\mathbf{C}_i - \Phi\beta\|^2 + \lambda\|\beta\|_{2,1},$$

*where $\lambda \geq 4\lambda_0$ with*

$$\lambda_0 = \frac{2\sigma}{\sqrt{n}}\sqrt{1 + \sqrt{\frac{8\log(d/\delta)}{p_{\min}}} + \frac{8\log(d/\delta)}{p_{\min}}}.$$

*Then, for any $\delta \in (0, 1)$, with probability at least $1 - \frac{\delta}{d}$,*

$$\frac{1}{n}\|\mathbf{C}_i - \Phi\widehat{\beta}_i\|^2 + \lambda\|\widehat{\beta}_i - \beta_i^\star\|_{2,1} \leq \frac{24\lambda^2 p^{\pi(i)}}{\kappa^2} \tag{13}$$

$$\|(\widehat{\Sigma}(\widehat{\beta} - \beta^\star))^j\| \leq \lambda\sqrt{p^j} \qquad \text{for all } j = 1, \ldots, d \tag{14}$$

$$\|\widehat{\beta} - \beta^\star\|_{2,1} \leq \frac{24\lambda p^{\pi(i)}}{\kappa^2}. \tag{15}$$

This theorem offers us a several things. Equation 13 gives us a bound on the true prediction error. The last two equations, (14, 15), are needed to prove that we find accurate parameter values using the Group Lasso approach. The fact that the last equation gives a bound in the $(2, 1)$-norm, allows us to use a duality argument later on to provide a bound on the $(2, \infty)$-norm of the difference between the learned and true parameter. Knowing that only one of the groups has to be non-zero combined with this uniform bound enables us to conclude that the correct group has been identified in the proof of Theorem A.5.

*Proof.* First let us define for every $j = 1, \ldots, d$ the random events $\mathcal{A}_j = \{\frac{1}{n}\|(\Phi^\top \varepsilon_i)^j\| \leq \frac{\lambda\sqrt{p^j}}{2}\}$ and their intersection $\mathcal{A} = \bigcap_{j=1}^d \mathcal{A}_j$. Most importantly, we see from Lemma C.2 that this event has probability at least $1 - \frac{\delta}{d}$. We get the $1/d$ factor by using $\widetilde{\delta} = \frac{\delta}{d}$ in Lemma C.2 and noticing that this only adds a factor of 2 in the log terms. The first assertion (13) is true on the event $\mathcal{A}$ and follows from the proof of Theorem 8.1 in [10] and noting that in our setting their oracle parameter is given by our $\beta_i^\star$ and that $\mathbf{f}_0 = \Phi\beta_i^\star$.

Moving on towards (14), by the optimality condition (11) and (12) we have for each $j = 1, \ldots, d$

$$\frac{1}{n}\|(\Phi(\mathbf{C}_i - \Phi\widehat{\beta})^j\| \leq \frac{\lambda\sqrt{p^j}}{2}.$$

Let us rewrite the expression in (14) into

$$\|(\widehat{\Sigma}(\widehat{\beta}_i - \beta_i^\star))^j\| = \frac{1}{n}\|(\Phi^\top(\Phi\widehat{\beta} - \Phi\beta^\star))^j\|.$$

Substituting $\Phi\beta_i^\star = \mathbf{C}_i - \varepsilon_i$ into this expression gives

$$\|(\widehat{\Sigma}(\widehat{\beta}_i - \beta_i^\star))^j\| \leq \frac{1}{n}\|(\Phi^\top(\Phi\widehat{\beta}_i - \mathbf{C}_i))^j\| + \frac{1}{n}\|(\Phi^\top \varepsilon_i)^j\|$$
$$\leq \frac{\lambda\sqrt{p^j}}{2} + \frac{\lambda\sqrt{p^j}}{2} = \lambda\sqrt{p^j}.$$

Note that this inequality only holds on $\mathcal{A}$.

The final assertion is a direct consequence of the first,

$$\lambda\|\widehat{\beta}_i - \beta_i^\star\|_{2,1} \leq \frac{1}{n}\|\Phi(\widehat{\beta}_i - \beta_i^\star)\|^2 + \lambda\|\widehat{\beta} - \beta^\star\|_{2,1} \leq \frac{24\lambda^2 p^{\pi(i)}}{\kappa^2}$$
$$\|\widehat{\beta}_i - \beta_i^\star\|_{2,1} \leq \frac{24\lambda p^{\pi(i)}}{\kappa^2}.$$

$\square$

To state and prove the general version of Theorem 4.2 we also need to generalize Assumption 4.1.

**Assumption A.3.** There exists some constant $a > 1$ such that for any $j \neq j'$, it holds that

$$\max_{1 \leq t \leq \min(p^j, p^{j'})} |(\widehat{\Sigma}_{jj'})_{tt}| \leq \frac{1}{14a}\sqrt{\frac{p_{\min}}{p_{\max}}} \tag{16}$$

and

$$\max_{1 \leq t \leq p^j, 1 \leq t' \leq p^{j'}, t \neq t'} |(\widehat{\Sigma}_{jj'})_{tt'}| \leq \frac{1}{14a}\sqrt{\frac{p_{\min}}{p_{\max}}}\frac{1}{\sqrt{p^j p^{j'}}}. \tag{17}$$

The previous assumption is stronger than the RE(1 ) property, as shown by the following lemma:

**Lemma A.4.** *Let Assumption A.3 be satisfied. Then RE(1) is satisfied with $\kappa = \sqrt{1 - 1/a}$.*

*Proof.* This is Lemma B.3 in [35]. $\square$

The following theorem is a modification of Theorem 5.1 by Lounici et al. [35], where some adaptations are made to adjust the result to our setting.

**Theorem A.5.** *Let Assumption (A.3) hold, for $\ell = 1, \ldots, d$, $\varepsilon_i^\ell \sim \mathcal{N}(0, \sigma^2)$ independently, $\sigma^2 > 0$, and with $\delta \in (0, 1)$ set $\lambda \geq 4\lambda_0$, where*

$$\lambda_0 = \frac{2\sigma}{\sqrt{n}}\sqrt{1 + \sqrt{\frac{8\log(d/\delta)}{p_{\min}}} + \frac{8\log(d/\delta)}{p_{\min}}}.$$

*Furthermore, set $c = \left(1 + \frac{24}{7(a-1)}\right)$. Then, for any $\delta \in (0, 1)$, with probability at least $1 - \frac{\delta}{d}$, any solution $\widehat{\beta}_i$ of (10) satisfies*

$$\|\widehat{\beta}_i - \beta_i^\star\|_{2,\infty} \leq c\lambda\sqrt{p_{\max}}. \tag{18}$$

*If, in addition, $\|(\beta_i^\star)^{\pi(i)}\| > 2c\lambda\sqrt{p_{\max}}$, then* (18) *implies that*

$$\widehat{J}_i = \arg\max_{j=1,\ldots,d} \|\widehat{\beta}_i^j\|$$

*estimates $\pi(i)$ correctly.*

*Proof.* Most of the proof is similar to the proof of Theorem 5.1 in Lounici et al. [35]. We supply a full proof for completeness and because our setting is slightly different. We will need more notation to prove this statement. Set $p_\infty = \max_{1 \leq d} p^j$ and define the extended covariance matrices $\widetilde{\Sigma}_{jj'}$ of size $p_\infty \times p_\infty$ as

$$\widetilde{\Sigma}_{jj'} = \left[\begin{array}{c|c} \widehat{\Sigma}_{jj'} & 0 \\ \hline 0 & 0 \end{array}\right] \text{ if } j \neq j' \text{ and } \widetilde{\Sigma}_{jj} = \left[\begin{array}{c|c} \widehat{\Sigma}_{jj} - I_{p^j \times p^j} & 0 \\ \hline 0 & 0 \end{array}\right] \text{ if } j = j'.$$

We also define for any $j = 1, \ldots, d$ and $\Delta \in \mathbb{R}^{k\overline{p}}$ the vector $\widetilde{\Delta}^j \in \mathbb{R}^{p_\infty}$ such that

$$\widetilde{\Delta}^j = \left[\begin{array}{c} \Delta^j \\ 0 \end{array}\right].$$

Now set $\Delta = \widehat{\beta}_i - \beta_i^\star$ and bound

$$\|\Delta\|_{2,\infty} = \|\widehat{\Sigma}\Delta - (\widehat{\Sigma} - I_{k\overline{p} \times k\overline{p}})\Delta\|_{2,\infty} \leq \|\widehat{\Sigma}\Delta\|_{2,\infty} + \|(\widehat{\Sigma} - I_{k\overline{p} \times k\overline{p}})\Delta\|_{2,\infty}.$$

The first term is controlled by (14) from Lemma A.2. The latter term can be bounded by noticing that only the off-diagonal elements will contribute to the norm. We can bound it using Cauchy-Schwarz:

$$\|(\widehat{\Sigma} - I_{k\overline{p} \times k\overline{p}})\Delta\|_{2,\infty} = \max_{j=1,\ldots,d} \|((\widehat{\Sigma} - I_{k\overline{p} \times k\overline{p}})\Delta)^j\|$$

$$= \max_{j=1,\ldots,d} \left[\sum_{t=1}^{p^j} \left(\sum_{j'=1}^{d} \sum_{t'=1}^{p^{j'}} (\widetilde{\Sigma}_{jj'})_{tt'} \widetilde{\Delta}_{t'}^{j'}\right)^2\right]^{1/2}$$

$$\leq \max_{j=1,\ldots,d} \left[\sum_{t=1}^{p^j} \left(\sum_{j'=1}^{d} (\widetilde{\Sigma}_{jj'})_{tt} \widetilde{\Delta}_t^{j'}\right)^2\right]^{1/2} + \max_{j=1,\ldots,d} \left[\sum_{t=1}^{p^j} \left(\sum_{j'=1}^{d} \sum_{t'=1, t' \neq t}^{p^{j'}} (\widetilde{\Sigma}_{jj'})_{tt'} \widetilde{\Delta}_{t'}^{j'}\right)^2\right]^{1/2}.$$

We now bound both terms separately. The first term can be bounded using an application of Assumption A.3 and then Minkowski's inequality. The Minkowski's inequality is true for $L^p$ norms and tells us

$$\|x + y\|_p \leq \|x\|_p + \|y\|_p.$$

In our case this generalises to

$$\left[\sum_{t=1}^{p_\infty} \left(\sum_{j'=1}^{d} |\widetilde{\Delta}_t^{j'}|\right)^2\right]^{1/2} = \|\sum_{j=1}^{d} \widetilde{\Delta}^j\| \leq \sum_{j=1}^{d} \|\widetilde{\Delta}^j\| \leq \frac{1}{\sqrt{p_{\min}}} \sum_{j=1}^{d} \sqrt{p^j} \|\widetilde{\Delta}^j\| = \frac{1}{\sqrt{p_{\min}}} \|\widetilde{\Delta}\|_{2,1}.$$

Combining Assumption A.3 with the above inequality gives us

$$\max_{j=1,\ldots,d} \left[\sum_{t=1}^{p_j} \left(\sum_{j'=1}^{d} (\widetilde{\Sigma}_{jj'})_{tt} \widetilde{\Delta}_t^{j'}\right)^2\right]^{1/2} \leq \frac{1}{14a} \sqrt{\frac{p_{\min}}{p_{\max}}} \left[\sum_{t=1}^{p_\infty} \left(\sum_{j'=1}^{d} |\widetilde{\Delta}_t^{j'}|\right)^2\right]^{1/2}$$

$$\leq \frac{1}{14a} \sqrt{\frac{p_{\min}}{p_{\max}}} \frac{1}{\sqrt{p_{\min}}} \|\widetilde{\Delta}\|_{2,1}$$

$$\leq \frac{1}{14a} \sqrt{\frac{1}{p_{\max}}} \|\Delta\|_{2,1}.$$

The second term can now be bounded by another application of Cauchy-Schwarz:

$$\max_{j=1,\ldots,d}\left[\sum_{t=1}^{p^j}\left(\sum_{j'=1}^{d}\sum_{t'=1,t'\neq t}^{p^{j'}}(\widetilde{\Sigma}_{jj'})_{tt'}\widetilde{\Delta}_{t'}^{j'}\right)^2\right]^{1/2}$$

$$\leq\frac{1}{14a}\sqrt{\frac{p_{\min}}{p_{\max}}}\max_{j=1,\ldots,d}\left[\frac{1}{p^j}\sum_{t=1}^{p^j}\left(\sum_{j'=1}^{d}\sum_{t'=1}^{p^{j'}}\frac{|\widetilde{\Delta}_{t'}^{j'}|}{\sqrt{p^{j'}}}\right)^2\right]^{1/2}$$

$$\leq\frac{1}{14a}\sqrt{\frac{p_{\min}}{p_{\max}}}\sum_{j'=1}^{d}\sum_{t'=1}^{p^{j'}}\frac{|\widetilde{\Delta}_{t'}^{j'}|}{\sqrt{p^{j'}}}$$

$$\leq\frac{1}{14a}\sqrt{\frac{p_{\min}}{p_{\max}}}\frac{1}{\sqrt{p_{\min}}}\|\widetilde{\Delta}\|_{2,1}$$

$$\leq\frac{1}{14a}\sqrt{\frac{1}{p_{\max}}}\|\Delta\|_{2,1}.$$

The $(2,1)$-norm term is now bounded using (15). Putting everything together we get

$$\|\widehat{\beta}_i-\beta_i^\star\|_{2,\infty}\leq\|\widehat{\Sigma}(\widehat{\beta}_i-\beta_i^\star)\|_{2,\infty}+\|(\widehat{\Sigma}-I_{pd\times pd})(\widehat{\beta}_i-\beta_i^\star)\|_{2,\infty}$$

$$\leq\lambda\sqrt{p_{\max}}+\frac{2}{14a}\sqrt{\frac{1}{p_{\max}}}\left(\frac{24\lambda p^{\pi(i)}}{\kappa^2}\right)$$

$$\leq\left(1+\frac{24}{7\kappa^2 a}\right)\lambda\sqrt{p_{\max}}.$$

To satisfy both assumptions (A.1, 4.1), we need to set $a\kappa^2=(a-1)$ as per Lemma A.4.

Finally, to prove the final claim, note that (18) combined with our sparsity assumption on the true parameters implies that for all $j'\neq\pi(i)$ it must be that $\|\widehat{\beta}_i^{j'}\|=\|\widehat{\beta}_i^{j'}-(\beta_i^\star)^{j'}\|<c\lambda\sqrt{p_{\max}}$. We will show that for $\pi(i)$ it must be that $\|\widehat{\beta}_i^{\pi(i)}\|>c\lambda\sqrt{p_{\max}}$. Hence, the estimator gets the correct index with high probability. Indeed, if $\|(\beta_i^\star)^{\pi(i)}\|>2c\lambda\sqrt{p_{\max}}$ we get

$$\|\widehat{\beta}_i^{\pi(i)}\|=\|(\beta_i^\star)^{\pi(i)}-((\beta_i^\star)^{\pi(i)}-\widehat{\beta}_i^{\pi(i)})\|$$

$$\geq\left|\|(\beta_i^\star)^{\pi(i)}\|-\|((\beta_i^\star)^{\pi(i)}-\widehat{\beta}_i^{\pi(i)})\|\right|$$

$$\geq 2c\lambda\sqrt{p_{\max}}-c\lambda\sqrt{p_{\max}}$$

$$=c\lambda\sqrt{p_{\max}}.$$

$\square$

Let us restate the specific version of Theorem 4.2 again for clarity. This theorem is now a corollary of Theorem A.5.

**Theorem 4.2.** *Suppose the data have been pre-processed to satisfy* (4) *and let Assump.* (4.1) *hold. Take any $\delta\in(0,1)$ and set $\lambda\geq 4\lambda_0$, where*

$$\lambda_0=\frac{2\sigma}{\sqrt{n}}\sqrt{1+\sqrt{\frac{8\log(d/\delta))}{p}}+\frac{8\log(d/\delta)}{p}},$$

*and set $c=\left(1+\frac{24}{7(a-1)}\right)$. Then, any solution $\widehat{\beta}_i$ of the Group Lasso objective* (3) *satisfies*

$$\|\widehat{\beta}_i-\beta_i^\star\|_{2,\infty}\leq c\lambda\sqrt{p}\tag{5}$$

*with probability at least $1-\frac{\delta}{d}$. If, in addition, $\|(\beta_i^\star)^{\pi(i)}\|>2c\lambda\sqrt{p}$, then* (5) *implies that $\widehat{J}_i=\arg\max_{j=1,\ldots,d}\|\widehat{\beta}_i^j\|$ estimates $\pi(i)$ correctly.*

*Proof.* The result follows from Theorem A.5, where in this case $k = d$, and $p^j = p$ for all $j = 1, \ldots, d$. $\qquad\square$

**Corollary A.6.** *Assume the same setting as Theorem 4.2 such that for each $i = 1, \ldots, d, \|(\beta_i^\star)^{\pi(i)}\| > 2c\lambda\sqrt{p}$ and consider the estimator $\widehat{\pi}$ as defined in (6). Then $\widehat{\pi} = \pi$ with probability at least $1 - \delta$.*

*Proof.* Consider the following estimators

$$\widehat{\beta}_i = \underset{\beta \in \mathbb{R}^{dp}}{\arg\min} \|\mathbf{C}_i - \Phi\beta\|^2 + \lambda\sqrt{p}\|\beta\|_{2,1},$$

$$\widehat{J}_i = \underset{j=1,\ldots d}{\arg\max} \|\widehat{\beta}_i^j\|,$$

$$\widetilde{\pi} : [d] \to [d], i \mapsto \widehat{J}_i.$$

We will first show that $\widetilde{\pi}$ estimates $\pi$ with probability at least $1 - \delta$. Afterwards, we will show that the event on which $\widetilde{\pi}$ is correct, is contained in the event that $\widehat{\pi}$ estimates $\pi$ correctly, implying a lower bound on the requested probability.

We apply a union bound

$$\begin{aligned}
\mathbb{P}(\widetilde{\pi} = \pi) &= \mathbb{P}(\forall i = 1, \ldots, d \mid \widehat{J}_i = \pi(i)) \\
&= 1 - \mathbb{P}(\exists i = 1, \ldots, d \mid \widehat{J}_i \neq \pi(i)) \\
&> 1 - \sum_{i=1}^d \frac{\delta}{d} \\
&= 1 - \delta.
\end{aligned}$$

We proceed to the second step. If $\widetilde{\pi}$ estimates $\pi$ correctly, then $\widetilde{\pi}$ is already a valid permutation and $\widetilde{\pi} = \widehat{\pi}$. Indeed, if $\widetilde{\pi}$ is correct then that means that $\|\widehat{\beta}_i^{\widetilde{\pi}(i)}\| = \|\widehat{\beta}_i^{\pi(i)}\|$ is the maximum norm for each $i$. Coincidentally, by $\pi$ being a correct permutation, $\widetilde{\pi}$ describes a correct matching with largest values, which means that $\widetilde{\pi}(i) = \widehat{\pi}(i)$ for each $i = 1, \ldots, d$ and

$$\mathbb{P}(\widehat{\pi} = \pi) \geq \mathbb{P}(\widetilde{\pi} = \pi) \geq 1 - \delta.$$

$\qquad\square$

# B   Proofs and Implementation Details for Kernelized Alignment Learning

In this section we provide proof details and the algorithm for Section 5. We will make one adjustment to the Group Lasso regularization in the optimization problem compared to (7), which is that we square the regularization term. This form is theoretically more appealing, but is still equivalent to the

---

**Algorithm 2** Estimating the permutation using kernels

---

1: Input: reg. parameter $\lambda > 0$, kernels $\kappa_1, \ldots, \kappa_d$
2: Data: $\{(C^{(\ell)}, M^{(\ell)})\}_{\ell=1}^n$
3: **for** $j = 1, \ldots, d$ **do**
4: $\quad (K_j)_{\ell k} \leftarrow \kappa_j(M^{(\ell)}, M^{(k)})$
5: $\quad L_j \leftarrow \text{CholeskyDecomposition}(K_j)$
6: **end for**
7: **for** $i = 1, \ldots d$ **do**

8: $\quad \widehat{\gamma}_i \leftarrow \underset{\gamma \in \mathbb{R}^{np}}{\arg\min} \frac{1}{n}\|\mathbf{C}_i - \sum_{j=1}^d L_j\gamma^j\|^2 + \lambda\|\gamma\|_{2,1}$

9: **end for**

10: $\widehat{\pi} \leftarrow \underset{\pi \in \Pi}{\arg\max} \sum_{i=1}^d \|\widehat{\gamma}_i^{\pi(i)}\|$

---

standard formulation: as Bach [4] argues, the two versions of the optimization problem have the same sets of solutions when varying the regularization parameters. For $\mu > 0$, the objective is given by

$$\inf_{\beta^1,\ldots,\beta^d} \frac{1}{n} \|\mathbf{C}_i - \beta(\mathbf{M})\|^2 + \mu \left( \sum_{j=1}^{d} \|\beta^j\|_{\mathcal{H}_j} \right)^2. \tag{19}$$

Let $\widehat{\beta}_i^1, \ldots, \widehat{\beta}_i^d$ be the solutions of the above optimization problem. The translation between regularization parameters that give the same solutions for (7) and (19) is given by $\lambda = \mu \left( \sum_{j=1}^{d} \|\widehat{\beta}_i^j\|_{\mathcal{H}_j} \right)$.

## B.1 Representer Theorem

The squared version of the optimization problem allows us to prove the Representer theorem from the main text:

**Theorem 5.1.** *Let $\varphi_1, \ldots, \varphi_d$ be the feature maps associated with $\mathcal{H}_1, \ldots, \mathcal{H}_d$. Then there exist $\hat{c}_i^1, \ldots, \hat{c}_i^d \in \mathbb{R}^n$ such that the optimization problem in (7) has solution $\widehat{\beta}_i$ with each $\widehat{\beta}_i^j$ of the form*

$$\widehat{\beta}_i^j = \sum_{\ell=1}^{n} \varphi_j(M_j^{(\ell)})(\hat{c}_i^j)_\ell.$$

*Proof.* First we state the following result about a variational equality for positive numbers

$$\left( \sum_{j=1}^{d} \|\beta^j\| \right)^2 = \inf_{\eta \in \Delta_d} \sum_{j=1}^{d} \frac{\|\beta^j\|^2}{\eta^j}. \tag{20}$$

A proof of this statement can be found in section 1.5 of [5]. Using (20) and switching to the squared version of (7) we rewrite (19) as

$$\inf_{\beta_1,\ldots,\beta_d} \frac{1}{n} \|\mathbf{C}_i - \beta(\mathbf{M})\|^2 + \mu \left( \sum_{j=1}^{d} \|\beta^j\|_{\mathcal{H}_j} \right)^2$$

$$= \inf_{\eta \in \Delta_d} \inf_{\beta_1,\ldots,\beta_d} \frac{1}{n} \|\mathbf{C}_i - \beta(\mathbf{M})\|^2 + \mu \sum_{j=1}^{d} \frac{\|\beta^j\|_{\mathcal{H}_j}^2}{\eta_j}$$

$$= (\text{OPT}_1).$$

We can rewrite this expression further, using the reproducing property of the RKHSs $\mathcal{H}_j$, which gives $\beta^j(M_j) = \langle \beta^j, \varphi_j(M_j) \rangle$. Furthermore, defining $\widetilde{\beta^j} = \frac{\beta^j}{\sqrt{\eta_j}}$ and $\widetilde{\varphi}_j = \sqrt{\eta_j}\varphi_j$ we rewrite

$$(\text{OPT}_1) = \inf_{\eta \in \Delta_d} \inf_{\beta_1,\ldots,\beta_d} \frac{1}{n} \sum_{\ell=1}^{n} \left( C_i^\ell - \sum_{j=1}^{d} \left\langle \beta^j, \varphi_j(M_i^{(\ell)}) \right\rangle \right)^2 + \mu \sum_{j=1}^{d} \frac{\|\beta^j\|_{\mathcal{H}_j}^2}{\eta_j}$$

$$= \inf_{\eta \in \Delta_d} \inf_{\widetilde{\beta}_1,\ldots,\widetilde{\beta}_d} \frac{1}{n} \sum_{\ell=1}^{n} \left( C_i^{(\ell)} - \sum_{j=1}^{d} \left\langle \sqrt{\eta_j}\widetilde{\beta^j}, \varphi_j(M_j^{(\ell)}) \right\rangle \right)^2 + \mu \sum_{j=1}^{d} \|\widetilde{\beta^j}\|_{\mathcal{H}_j}^2$$

$$= \inf_{\eta \in \Delta_d} \inf_{\widetilde{\beta}_1,\ldots,\widetilde{\beta}_d} \frac{1}{n} \sum_{i=1}^{n} \left( C_i^{(\ell)} - \sum_{j=1}^{d} \left\langle \widetilde{\beta^j}, \widetilde{\varphi}_j(M_j^{(\ell)}) \right\rangle \right)^2 + \mu \sum_{j=1}^{d} \|\widetilde{\beta^j}\|_{\mathcal{H}_j}^2$$

$$= (\text{OPT}_2).$$

This final expression should be recognized as the feature representation of the Representer theorem [48] applied to the kernel described by

$$\kappa(\eta)(M, M') = \sum_{j=1}^{d} \left\langle \widetilde{\varphi_j}(M_j), \widetilde{\varphi_j}(M'_j) \right\rangle$$

$$= \sum_{j=1}^{d} \eta_j \left\langle \varphi_j(M_j), \varphi_j(M'_j) \right\rangle$$

$$= \sum_{j=1}^{d} \eta_j \kappa_j(M_j, M'_j).$$

The Representer theorem then gives us that the solution of the inner optimization problem in (OPT$_2$) can be described by

$$\widetilde{\beta^j} = \sum_{\ell=1}^{n} \widetilde{\varphi_j}(M_j^{\ell})(c)_\ell \iff \frac{\beta^j}{\sqrt{\eta_j}} = \sqrt{\eta_j} \sum_{\ell=1}^{n} \varphi_j(M_j^{(\ell)})(c)_\ell \iff \beta^j = \sum_{\ell=1}^{n} \varphi_j(M_j^{(\ell)})\eta_j(c)_\ell$$

with $c \in \mathbb{R}^n$ and $\eta \in \Delta_d$. Alternatively interpreted this says that there exist $c^1, \ldots, c^d \in \mathbb{R}^n$ such that $\beta^j = \sum_{\ell=1}^{n} \varphi_j(M_j^{(\ell)})(c^j)_\ell$. We invoke again the equivalence between the squared and un-squared versions of the optimization problem using the translation of regularization parameters $\lambda = \mu \left( \sum_{j=1}^{d} \|\widehat{\beta}_i^j\|_{K_j} \right)$ and conclude that the solutions of (7) are of the same form. $\qquad\square$

To get the finite-dimensional optimization problem as stated in (8) we substitute the correct forms of $\widehat{\beta}_i^j$ back into the original optimization problem. Define the Gramm matrices $(K_j)_{\ell k} = \kappa_j(M_j^{(\ell)}, M_j^k) = \left\langle \varphi_j(M_j^{(\ell)}, \varphi_j(M_j^k) \right\rangle$ and observe

$$\inf_{\beta_1, \ldots, \beta_d} \frac{1}{n} \sum_{\ell=1}^{n} \left( C_i^{(\ell)} - \sum_{j=1}^{d} \left\langle \beta^j, \varphi_j(M_j^{(\ell)}) \right\rangle \right)^2 + \lambda \sum_{j=1}^{d} \|\beta^j\|_{\mathcal{H}_j}$$

$$= \inf_{c^1, \ldots, c^d \in \mathbb{R}^n} \frac{1}{n} \sum_{\ell=1}^{n} \left( C_i^{(\ell)} - \sum_{j=1}^{d} (K_j c^j)_i \right)^2 + \lambda \sum_{j=1}^{d} \sqrt{(c^j)^\top K_j c^j}$$

$$= \inf_{c^1, \ldots, c^d \in \mathbb{R}^n} \frac{1}{n} \|\mathbf{C}_i - \sum_{j=1}^{d} K_j c^j\|^2 + \lambda \sum_{j=1}^{d} \|c^j\|_{K_j}.$$

## B.2  Estimator consistency

As stated in the main text, the assumptions in Theorem 5.2 are explained in this section. The assumptions stated in (A-D) ensure that the RKHSs that we work with are nice enough and that the function we want to estimate is not too miss specified. For a more complete discussion on the assumptions, we refer to Bach [4]. To remind ourselves, we are given $d$ random variables $M = (M_1, \ldots, M_d)$, where each random variable lives in $\mathcal{Z}_j$, and $d$ RKHSs $\mathcal{H}_1, \ldots, \mathcal{H}_d$ associated with $d$ kernels $\kappa_1, \ldots, \kappa_j$. The cross-covariance operator, $\Sigma_{ij}$ for $\mathcal{H}_j$ to $\mathcal{H}_i$ is defined such that for all $(\beta^i, \beta^j) \in \mathcal{H}_i \times \mathcal{H}_j$,

$$\left\langle \beta^j, \Sigma_{ij}\beta^j \right\rangle = \mathbb{E}[\beta^i(M_i)\beta^j(M_j)] - \mathbb{E}[\beta^i(M_i)]\mathbb{E}[\beta^j(M_j)]. \qquad (21)$$

The bounded correlation operators $\rho_{ij}$ are defined through the decomposition $\Sigma_{ij} = \Sigma_{ii}^{1/2} \rho_{ij} \Sigma_{jj}^{1/2}$ [6].

(A) For each $j = 1, \ldots, d$, the Hilbert space $\mathcal{H}_j$ is a separable reproducing kernel Hilbert space associated with kernel $\kappa_j$ and the random variables $\kappa_j(\cdot, M_j)$ are not constant and have finite fourth-order moments.

(B) For all $i, j = 1, \ldots, d$, the cross correlation operators are compact $\rho_{ij}$ and the joint correlation operator is invertible.

(C) For each $i = 1, \ldots, d$, there exist functions $\beta_i^{\star 1}, \ldots, \beta_i^{\star d} \in \mathcal{H}_1, \ldots, \mathcal{H}_d$, $b_i \in \mathbb{R}$ and a function $f_i$ of $M$ such that

$$C_i = \sum_{j=1}^{d} \beta_i^{\star j}(M_j) + b_i + f_i(M) + \varepsilon_i,$$

where $\mathbb{E}[\varepsilon_i \mid M] = 0$ and $\sigma_{\min}^2 < \mathbb{E}[\varepsilon_i^2 \mid M] < \sigma_{\max}^2$ with $\mathbb{E}[f_i(M)^2] < \infty$, $\mathbb{E}[f_i(M)] = 0$ and $\mathbb{E}[f_i(M)\beta_i^{\star j}(M_j) = 0]$ for all $j = 1 \ldots, d$. We define $\pi(i)$ to be the one index for which $\beta_i^{\star \pi(i)} \neq 0$.

(D) For all $i, j = 1, \ldots, d$, there exists $g_i^j \in \mathcal{H}_j$ such that $\beta_i^{\star j} = \Sigma_{jj}^{1/2} g_i^j$.

For each function, $\beta_i^{\star \pi(i)}$, hat is non-zero we will require the following condition

$$\max_{j \neq \pi(i)} \left\| \Sigma_{jj}^{1/2} \rho_{i\pi(i)} \rho_{\pi(i)\pi(i)}^{-1} D g_{\pi(i)} \right\|_{\mathcal{H}_i} < 1, \tag{22}$$

Where $D$ is a block diagonal operator where each block consists of the operators $\frac{1}{\|\beta_i^{\star j}\|_{\mathcal{H}_j}} I_{\mathcal{H}_j}$. Condition (B) can be seen as an analogue to the correlation assumption in Assumptions 4.1 and A.3, as it ensures that the variables are not too dependent.

Before we prove Theorem 5.2, we will first prove that each individual index $\pi(i)$ can be estimated consistently. This follows from an asymptotic result by Bach [4].

**Theorem 5.2.** *Assume (A–D) in App. B.2. Then, for any sequence of regularization parameters $\lambda_n$ such that $\lambda_n \to 0$ and $\sqrt{n}\lambda_n \to +\infty$ when $n \to \infty$, the estimated permutation $\widehat{\pi}$ converges in probability to $\pi$.*

*Proof.* To prove this result we first define the estimator of each individual index $\pi(i)$ as

$$\widehat{J}_i = \arg\max_{j=1,\ldots,d} \|\widehat{\beta}_i^j\|_{\mathcal{H}_j}. \tag{23}$$

Theorem 11 in [4] gives consistency for the estimated parameters $\widehat{\beta}_i^j$ and estimated index $\widehat{J}_i$. However, his result is stated for the squared version of the Group Lasso and has as assumption that the $\mu_n$ regularization parameters have the property that $\mu_n \to \infty$ and $\sqrt{n}\mu_n \to +\infty$ as the number of data points $n \to \infty$. The translation factor between regularization parameters $\lambda_n = \mu_n(\sum_{j=1}^{d} \|\widehat{\beta}_i^j\|_{K_j})$ convergence to a constant in probability, by the consistency of the estimated parameters. This shows that the scalings for $\lambda_n$ and $\mu_n$ are the same asymptotically. We conclude that $\widehat{J}_i$ estimates $\pi(i)$ with probability tending to 1.

Remember that the norms of $\widehat{\beta}_i^j$ and $\hat{c}_i^j$ are the same through $\|\widehat{\beta}_i^j\|_{\mathcal{H}_j} = \|\hat{c}_i^j\|_{K_j}$. This means that we have consistency for estimators of $\pi(i)$ that are based on $\|\hat{c}_i^j\|_{K_j}$ as well. For each $i = 1, \ldots, d$ we can repeat the above argumentation to get consistency for each $\widehat{J}_i$ separately. To combine the conclusions, we apply the same argument as in the finite dimensional case and a a union bound that finishes our proof,

$$\mathbb{P}(\widehat{\pi} = \pi) \geq 1 - \mathbb{P}(\exists i = 1, \ldots, d \mid \widehat{J}_i \neq \pi(i))$$

$$\geq 1 - \sum_{i=1}^{d} \mathbb{P}(\widehat{J}_i \neq \pi(i)) \to 1.$$

$\square$

## B.3 Implementation

**Lemma B.1.** *For each $j = 1, \ldots, d$ let $L_j$ be the Cholesky decomposition of the Gramm matrix $K_j$, then $\hat{c}_i^j = (L_j^\top)^{-1}\hat{\gamma}_i^j$ if the parameters $\hat{\gamma}_i^1, \ldots, \hat{\gamma}_i^d$ are minimizers of*

$$\min_{\gamma^1,\ldots,\gamma^d \in \mathbb{R}^n} \frac{1}{n}\|\mathbf{C}_i - \sum_{j=1}^{d} L_j \gamma^j\|^2 + \lambda\|\gamma^j\|_{2,1}. \tag{9}$$

*Proof.* Applying the Cholesky decomposition $K_j = L_j L_j^\top$ for each $j = 1, \ldots, d$ and substituting this into (9) gives

$$\inf_{c^1,\ldots,c^d\in\mathbb{R}^n} \frac{1}{n}\|\mathbf{C}_i - \sum_{j=1}^d K_j c^j\|^2 + \lambda\sum_{j=1}^d \|c^j\|_{K_j} = \inf_{c^1,\ldots,c^d\in\mathbb{R}^n} \frac{1}{n}\|\mathbf{C}_i - \sum_{j=1}^d L_j L_j^\top c^j\| + \lambda\sum_{j=1}^d \|L_j^\top c^j\|$$

$$= \inf_{\gamma^1,\ldots,\gamma^d\in\mathbb{R}^n} \frac{1}{n}\|\mathbf{C}_i - \sum_{j=1}^d L_j\gamma^j\|^2 + \lambda\sum_{j=1}^d \|\gamma^j\|.$$

$\square$

The number of parameters now scales with the number of data samples. Computationally, this quickly becomes unwieldy. We apply a Nyström style approximation by sub-sampling $m \ll n$ columns of each $K_j$ Gramm matrix and using those to approximate the full Gramm matrix [52].

## C  Probability Results

**Lemma C.1.** *For $j = 1, \ldots, d$ and $\sigma^2 > 0$ let $\frac{\chi_j^2}{\sigma^2}$ be independent chi-square distributed random variables with $p^j$ degrees of freedom for $j = 1, \ldots, d$. Then, with $\delta \in (0,1)$ and for*

$$\lambda_0 = \frac{2\sigma}{\sqrt{n}}\sqrt{1 + \sqrt{\frac{4\log(d/\delta)}{p_{\min}}} + \frac{4\log(d/\delta)}{p_{\min}}},$$

*we have*

$$\mathbb{P}\left(\max_{1\le j\le d} \frac{\chi_j}{\sqrt{np^j}} \le \frac{\lambda_0}{2}\right) \ge 1 - \delta.$$

*Proof.* This is Lemma 8.1 in [10] and substituting $x = \log(1/\delta)$. $\square$

The previous lemma is the general version of a concentration inequality that is needed in the proof of Theorem A.5. The concentration inequality that we want to use is the following.

**Lemma C.2.** *Let $\sigma^2 > 0$ and assume that $\varepsilon^{(1)}, \ldots \varepsilon^{(n)}$ are independently $\mathcal{N}(0, \sigma^2)$ distributed, $\Phi$ as in Appendix A, and with $\delta \in (0,1)$ set $\lambda \ge 4\lambda_0$ for*

$$\lambda_0 = \frac{2\sigma}{\sqrt{n}}\sqrt{1 + \sqrt{\frac{4\log(d/\delta)}{p_{\min}}} + \frac{4\log(d/\delta)}{p_{\min}}}$$

*Then, $\mathbb{P}(\mathcal{A}) \ge 1 - \delta$, where $\mathcal{A} = \bigcap_{j=1}^d \mathcal{A}_j$ with the events $\mathcal{A}_j = \{\frac{1}{n}\|(\Phi^\top\varepsilon)^j\| \le \frac{\lambda\sqrt{p^j}}{2}\}$ for all $j = 1, \ldots d$ and $\boldsymbol{\varepsilon} = [\varepsilon^{(1)}, \ldots \varepsilon^{(n)}]^\top$.*

*Proof.* By assumption, $I_{p^j\times p^j} = \widehat{\Sigma}_j = \frac{1}{n}(\Phi_j)^\top\Phi_j$ and the fact that $\varepsilon \sim \mathcal{N}(0, \sigma^2 I_{n\times n})$, we first see

$$\frac{1}{\sigma\sqrt{n}}(\Phi\varepsilon)^j = \frac{1}{\sigma\sqrt{n}}\Phi_j^\top\varepsilon$$
$$\sim \mathcal{N}(0, \frac{1}{n}\Phi_j^\top I_{n\times n}\Phi_j)$$
$$\sim \mathcal{N}(0, I_{p^j\times p^j}).$$

This shows us that $\frac{1}{\sigma^2 n}\|(\Phi^\top\varepsilon)^j\|^2$ has a chi-squared distribution. We can now apply Lemma C.1 by noticing that it holds also holds for $\lambda \ge 4\lambda_0 \ge \lambda_0$ and that

$$\bigcap_{j=1}^d \mathcal{A}_j = \left\{\max_{1\le j\le d} \frac{1}{\sqrt{np^j}}\frac{1}{\sqrt{n}}\|(\Phi\varepsilon)^j\| \le \frac{\lambda}{2}\right\}.$$

$\square$

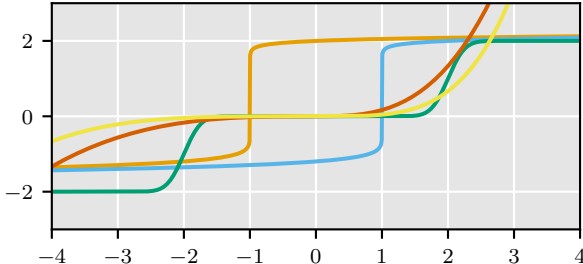

Figure 4: The different types of diffeomorphisms used in the misspecified case.

## D   Experimental Details

All the code to reproduce the experiments and figures in this paper is provided as a GitHub repository at https://github.com/HiddeFok/sample-efficient-learning-of-concepts. We perform synthetic experiments on a 192-core CPU node (AMD Genoa) with 336GB of RAM. We perform the image benchmark experiments on single 18-core GPU node (NVIDIA A100 GPU and Intel XEON CPU) with 120GB of RAM.

### D.1   Toy Dataset Experiments

The synthetic experiments can be subdivided into 4 sets, based on which features mapping is used. These features mappings are linear features, spline features , random Fourier features and kernels. Here, we describe the data generatation and the hyperparameters of the features and kernels.

#### D.1.1   Data Generation

For the synthetic experiment we sample the $C \in \mathbb{R}^d$ variables from a $\mathcal{N}(0, (1-\rho)I_{d \times d} + \rho\mathbb{1})$ distribution, where $\mathbb{1}$ denotes a matrix filled with only 1's. The $\rho \in (0,1)$ parameter controls the amount of correlation between the variables. For the experiments with continuous concepts, we sample $n$ data points, on which we perform a $80/20$ train/test data split. For the binary concept experiments, we perform a $50/50$ train/test split, to ensure that both labels of each concept appear in the test set. We also keep the correlation coefficient in the test set at $\rho = 0$, while varying the correlation in the train set.

We now simulate the $M$ variables directly from the $C$ variables, instead of using a causal representation learning method that would learn them from the observations $X$. This allows us to control the different sources of noise more carefully. There are 2 settings in which we generate the $M \in \mathbb{R}^d$ variables, well-specified (following all of our assumptions) and misspecified (settings in which we consider more general mixing functions beyond our theoretical assumptions.)

**Wellspecified.**   In the wellspecified setting we generate the $M$ variables by applying a map consisting of the features and kernels used to estimate the permutation. This setting is a sanity check to see if our estimator works in a setting that satisfies all the required assumptions. For each dimension $j = 1, \ldots, d$, a random weight vector $\beta_j^\star \in \mathbb{R}^p$ is sampled, such that $\|\beta_j^\star\| \in [16\lambda_0, 32\lambda_0]$ uniformly. A permutation $\pi \colon \{1, \ldots, d\} \to \{1, \ldots, d\}$ is uniformly sampled from all possible permutations. Finally, with independent $\varepsilon_i \sim \mathcal{N}(0, \sigma^2)$ noise variables we get

$$C_i = \varphi(M_{\pi(i)})^\top \beta_{\pi(i)}^\star + \varepsilon_i.$$

**Misspecified.**   In the misspecified setting we generate the $M$ variables by first sampling $d$ diffeomorphisms uniformly, $\{f_i \colon \mathbb{R} \to \mathbb{R}\}_{i=1}^d$, from a set of pre-specified diffeomorphisms. The set of possible functions is plotted in Figure 4. Each function gets a random scaling $w_i$ uniformly in $[-2, 2]$. Finally, we get

$$C_i = w_{\pi(i)} f_{\pi(i)}(M_{\pi(i)}) + \varepsilon_i,$$

with $\varepsilon_i \sim \mathcal{N}(0, \sigma^2)$ again independent noise variables.

**Target Label.** In the experiment where we need binary concepts, the concept labels were created by thresholding each variable at the midpoint. The underlying continuous score was generated using the misspecified setting from above. In this case the midpoint is at 0 and we get

$$\overline{C_i} = \mathbb{1}[C_i \leq 0].$$

The $M$ variables are kept continuous and a logistic regression is used to map $M$ to $\overline{C}$. To create the target labels, a subset of size $k = \min(3, \lfloor \frac{d}{5} \rfloor)$ is sampled from $\{1, \ldots, d\}$, without replacement. To determine if the target label is 0 or 1, we sum the binary concepts in the $k$ selected variables and check if the sum is at least $\max(1, \lfloor \frac{d}{10} \rfloor)$.

**Metrics.** For each of the experiments with continuous variables we report the the mean error in the learned permutation $\pi$ of the variables w.r.t to the true permutation $\widetilde{\pi}$ of the variables, defined as MPE $= \frac{1}{d} \sum_{i=1}^{d} \mathbb{1}\{\widetilde{\pi}(i) \neq \pi(i)\}$. Moreover, we report the $R^2$ score and execution time. To evaluate different settings of our estimator, we compare with the MPE, $R^2$ and execution time of using only the purely linear version of our estimator on that particular data setting. We also report the MPE and execution times of assigning the variables based on Pearson or Spearman correlations as naive baselines. In the binary experiments, we report the MPE, the mean concept accuracy, the label accuracy, the OIS-metric and the NIS-metric by Zarlenga et al. [58].

**Parameters of the experiments.** We vary the following parameters: the regularization parameter $\lambda$, the dimension $d$, the correlation $\rho$ and the number of data points $n$. As stated before, in each of the continuous experiment we look at the wellspecified and misspecified case. The settings in the continuous case are:

- The regularization parameter varies in $\lambda \in \{0.001, 0.005, 0.01, 0.05, 0.1, 0.5, 1\}$. The other settings are set to $d \in \{20, 60, 100\}$, $\rho = 0$ and $n = 1250$.
- The dimension is varied in $d \in \{5, 30, 60, 80, 100\}$. The other settings are set to $\lambda \in \{0.001, 0.01, 0.1\}$, $\rho = 0$ and $n = 1250$.
- The correlation parameter varies in $\rho \in \{0, 0.2, 0.4, 0.6, 0.8, 0.95, 0.99\}$. The other settings are set to $\lambda \in \{0.001, 0.01, 0.1\}$, $d = 60$ and $n = 1250$.
- The total number of data points is varied in $n \in \{65, 125, 1250, 2500, 5000\}$. The other settings are set to $\lambda \in \{0.001, 0.01, 0.1\}$, $d = 60$ and $\rho = 0$.

In the case of the binary concepts, we have fewer combinations, because we focus more on the downstream task and comparing the estimator to baselines:

- The dimension is varied in $d \in \{5, 10, 15, 20, 30\}$. The other settings are set to $\lambda \in \{0.001, 0.01, 0.1\}$, $\rho = 0.5$ and $n = 2000$.
- The total number of data points is varied in $n \in \{100, 200, 2000, 4000, 10000\}$. The other settings are set to $\lambda \in \{0.001, 0.01, 0.1\}$, $d = 20$ and $\rho = 0.5$.

### D.1.2 Feature and Kernel setting

We now describe the types of models we consider in our evaluation on the synthetic data.

**Linear features** In the linear case no transformation is applied to the $M$ variables.

**Spline features** In the spline features case we perform the regression using a spline basis transformation, either piecewise linear or cubic splines. We expect this method to work especially well, because the cubic splines form a dense subset in the space of twice-differentiable functions, of which the diffeomorphisms are a subset. To calculate these features we use the `SplineTransformer` class of the `scikit-learn` package. The total number of feature parameters is calculated as $p = n_k + n_d - 1$, where $n_k$ is the number of knots and $n_d$ is the degree of each spline. In each of the toy dataset experiments the number of knots was $n_l \in \{4, 8\}$ and the degrees were $n_d \in \{1, 3\}$.

**Random Fourier features** For the random Fourier features we use a varying number of random features. We sample random features that approximate the RBF kernel. To sample these features

we use the `RBFSampler` class of the `scikit-learn` package. The total number of feature parameters in this case is the number of random Fourier features. The number of features in the toy dataset experiments is $p \in \{2, 4, 6, 8\}$.

**Kernels** For the kernel experiments we perform the experiments for several kernels: the polynomial kernel, the RBF kernel, the Brownian kernel and a Sobolev kernel. These kernels are given by

$$\kappa_{\text{pol}}(x, y) = (1 + \langle x, y \rangle)^3$$
$$\kappa_{\text{RBF}}(x, y) = e^{-(x-y)^2}$$
$$\kappa_{\text{Lap}}(x, y) = e^{-|x-y|}$$
$$\kappa_{\cos}(x, y) = \cos(\langle x, y \rangle).$$

## D.2 Action/Temporal Datasets Experiments

We consider a synthetic data benchmark from the causal representation learning literature [31]. The data generation settings, model architectures and training hyperparameters are taken from the original paper [31]. The implementation can be found at https://github.com/slachapelle/disentanglement_via_mechanism_sparsity/tree/main, and is available under the Apache License 2.0.

### D.2.1 Dataset Generation Details

The benchmark consists of temporal data sequences, $\{(X^t, z^t, a^t)\}_{t=1}^T$, where $X^t \in \mathbb{R}^{20}$ is the observed data, $a^t \in \mathbb{R}^{10}$ is an action, which is seen as an auxiliary variable in the ICA framework developed in [24], and $z^t \in \mathbb{R}^{10}$ is the latent causal variable. The ground-truth mixing function $f$ is a random neural network with three hidden layers of 20 units with Leaky-ReLU activations with negative slope of $0.2$. The weight matrices are sampled independently according to $\mathcal{N}(0, 1)$ and the weight matrices are then orthogonalized to ensure invertability of the mixing function. The observational noise $\varepsilon$ in each dimension is sampled according to $\mathcal{N}(0, 10^{-4})$ and is added to $f(z^t)$. The transitions from $(z^{t-1}, a^{t-1})$ to $z^t$ are sampled according to $\mathcal{N}(\mu(z^{t-1}, a^{t-1}), 10^{-4} I_{10 \times 10})$. The mean function $\mu$ will be different between the Action Sparsity dataset and the Temporal Sparsity datasets.

**Action Sparsity Dataset** In this case, the sequences have length $T = 1$ and the mean function is given by

$$\mu(z^{t-1}, a^{t-1})_i := \sin(\tfrac{2+i}{\pi} a_i^{t-1} + (i-1)) + \sin(\tfrac{2+i}{\pi} a_{i-1}^{t-1} + (i-1)),$$

where the index $i = -1$ is periodically identified with $i = 10$.

**Temporal Sparsity Dataset** In this case, the sequences have length $T = 2$ and the mean function is given by

$$\mu(z^{t-1}, a^{t-1})_i := z_i^{t-1} + 0.5 \sum_{j=1}^{i} \sin(\tfrac{2+i}{\pi} z_j^{t-1} + (i-1)).$$

**Target Labels** We employ the same tactic as before to create the target labels. The true latent variables $z_i^t$ are binarized by looking at the empirical range over the whole dataset, and then thresholded at the midpoint. The target label is then determined by sampling 3 dimensions in $\{1, \ldots, 10\}$ and setting the label to 1 if at least 2 of the 3 latent variables labels are 1.

In both datasets we sample $10^6$ points and split the data $80/20$ for the train/test split.

### D.2.2 Model Architectures

As the first step of our pipeline we compare three models: TCVAE [11], iVAE [24] and DMS-VAE [31]. The same encoder and decoder architecture is used for all models: an MLP with 6 layers of 512 units with LeakyReLU activations with negative slope $0.2$. The encoder $f_{\text{enc}}(x; \theta)$ outputs the mean and standard deviation of $q_\theta(z^t \mid x^t)$, which are the densities of normal distributions. The latent transition distribution $\hat{p}_\lambda(z_i^t \mid z^{<t}, a^{<t})$, where $z^{<t} = (z^{t'})_{t'=1}^{t-1}$ and $a^{<t} = (a^{t'})_{t'=1}^{t-1}$, is also

Table 2: Architecture details for the encoder and decoder used in the temporal and action sparsity dataset experiments.

|  | Layer | Hidden Size | Activation Function |
|---|---|---|---|
|  | Linear | 512 | LeakyReLU(0.2) |
|  | Linear | 512 | LeakyReLU(0.2) |
|  | Linear | 512 | LeakyReLU(0.2) |
| Encoder | Linear | 512 | LeakyReLU(0.2) |
|  | Linear | 512 | LeakyReLU(0.2) |
|  | Linear | 512 | LeakyReLU(0.2) |
|  | Linear | $2 \cdot 10$ | - |
|  | Linear | 512 | LeakyReLU(0.2) |
|  | Linear | 512 | LeakyReLU(0.2) |
|  | Linear | 512 | LeakyReLU(0.2) |
| Decoder | Linear | 512 | LeakyReLU(0.2) |
|  | Linear | 512 | LeakyReLU(0.2) |
|  | Linear | 512 | LeakyReLU(0.2) |
|  | Linear | 20 | - |

learned by a fully connected neural network. The decoder $f_{\text{dec}}(z; \psi)$ tries to reconstruct the original data from the learned encodings. A minibatch size of $1024$ is used for training. See Table 2 for a detailed description.

The differences between the three methods come from the loss function that is optimized. The common term in each of the optimizations is the Evidence Lower Bound (ELBO) objective, which is given by

$$\text{ELBO}(\theta, \psi, \lambda) = \sum_{t=1}^{T} \mathop{\mathbb{E}}_{z^t \sim q_\theta(\cdot | x^t)} \left[ \log p_\psi(x^t \mid z^t) \right] \tag{24}$$

$$- \mathop{\mathbb{E}}_{z^{<t} \sim q_\theta(\cdot | x^{<t})} \left[ \text{KL}(q_\theta(z^t \mid x^t) \parallel \hat{p}_\lambda(z^t \mid z^{<t}, a^{<t})) \right], \tag{25}$$

where $\text{KL}(\cdot \parallel \cdot)$ is the Kullback-Leibler divergence.

**TCVAE**   We use the implementation of TCVAE [11] by Lachapelle et al. [31]. The loss function consists of the same components as the ELBO objective, but they decompose it into 3 terms and add a weight parameter to each of the terms. The hyperparameters for the training procedure can be found in Table 3.

**iVAE**   We use the implementation iVAE [24] by Lachapelle et al. [31]. The loss function here is similar to the ELBO objective, but it adds one parameter $\beta$ to the KL-term in the objective. The hyperparameters for the training procedure can be found in Table 3.

**DMS-VAE**   Lachapelle et al. [31] introduce a sparsity regularization in the ELBO objective to prove the identifiability in temporal and action settings. The new objective is given by

$$\text{ELBO}(\theta, \psi, \lambda) + \alpha_z \|\hat{G}_z\|_0 + \alpha_a \|\hat{G}_a\|_0.$$

The variable $\hat{G}_z$ is a learned matrix, representing the relations between the latent variables between two time steps. The variable $\hat{G}_a$ is a learned matrix representing the relations between the actions and the latent variables. The norm $\|\cdot\|_0$ counts the number of non-zero terms. This is a discrete objective and can transformed into a continuous objective using the Gumbel-Softmax trick [37, 22].

Instead of using the regularized objective, the authors also propose a constraint-based optimization procedure on the ELBO objective, where the constraint is determined by the number of edges in the learned graph. We use their constrained optimization method, which provides an optimization schedule that we set by enabling the `--constraint_schedule` flag. The other hyperparameters for the training procedure can be found in Table 3.

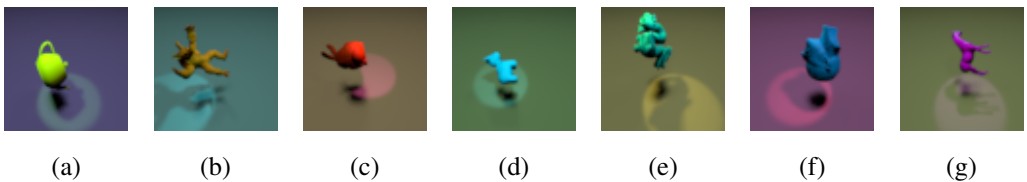

|    (a)    |    (b)    |    (c)    |    (d)    |    (e)    |    (f)    |    (g)    |

Figure 5: Examples of the 7 shapes in the Temporal Causal3DIdent dataset. From left to right: teapot, armadillow, bunny, cow, dragon, head and horse.

### D.3 Temporal Causal3DIdent

We consider an image data benchmark from the causal representation learning literature [33]. The dataset, model architectures and training hyper parameters are taken from [33]. The implementation can be found at https://github.com/phlippe/CITRIS/tree/main, which is available under the BSD 3-Clause Clear License.

#### D.3.1 Dataset Details

The data comes from a setting which is referred to as Temporal Intervened Sequences. The assumption is that there are $d$ causal variables $(G_1, \ldots, G_d)$ and a corresponding causal graph $\mathcal{G} = (V, E)$ where each node $i \in V$ represents a causal variable $G_i$. The variables can be real-valued or vector-valued, and each edge $(i, j) \in E$ represents a relation between $G_i$ and $G_j$. We assume that there are $T$ time steps and for every $t = 1, \ldots, T$ the causal variables follow a stochastic process. We therefore have a sequence $\{(G_1^t, \ldots, G_d^t)\}_{t=1}^T$, where only the causal variables in $t-1$ are the parents of the causal variables in time step $t$. The observations $X^t$ at each time step are created through a mixing function $X^t = f(G_1^t, \ldots G_d^t, \varepsilon^t)$, where $\varepsilon^t$ is an i.i.d. noise variable. Additionally, we have access to a $d$-dimensional binary vector at each time step $I^t \in \{0, 1\}^d$ that tells us which causal variables have been intervened on, but not with which value.

The causal variables that are used in the data generating process are the following:

- The **object position** (pos_o) is modelled in 3 dimensions $(x, y, z) \in [-2, 2]^3$. The values are forced to be in this interval to ensure that the object does not disappear from the image, becomes too small or covers the whole image.

- The **object rotation** (rot_o) is modelled in 2 dimensions $(\alpha, \beta) \in [0, 2\pi)^2$. Distances for angles are calculated in a periodic fashion, ensuring that angles close to 0 and $2\pi$ are close together.

- The **spotlight rotation** (rot_s) is the positioning of the spotlight that shines on the object. The value range is $[0, 2\pi)$, where distances are again calculated in a periodic fashion.

- The **spotlight hue** (hue_s) is the color of the spotlight. The range of the value is $[0, 2\pi)$, where 0 corresponds to red.

- The **background hue** (hue_b) is the color of the background. The value range is $[0, 2\pi)$ with 0 corresponding to red again.

Table 3: The hyperparameters used for the training of the DMS-VAE, TCVAE and iVAE for the action and temporal sparsity datasets

| Hyperparameter | Value |
| --- | --- |
| Batch Size | 1024 |
| Optimizer | Adam [27] and Cooper [12] |
| Learning rate | 5e-4 (DMS-VAE), 1e-4 (iVAE), 1e-3 (TCVAE) |
| KL divergence factor $\beta$ | 1.0 |
| Number of latents | 20 |
| Number of epochs | 500 |
| Gumbel Softmax temperature | 1.0 |

- The **object hue** (hue_o) is the color of the object, with value range is $[0, 2\pi)$ and again with 0 representing red.

Lippe et al. [33] generate the data using Blender, a setup inspired by von Kügelgen et al. [51] and using code provided by Zimmermann et al. [59]. They generate the dataset by starting with an initial random set of causal variables. They then sample the causal variables in each subsequent time step by following a specific conditional distribution, which is given by the set of equations in (26).

$$
\begin{aligned}
f(a, b, c) &= \frac{a - b}{2} + c \\
\text{pos\_x}^{t+1} &= f(1.5 \cdot \sin(\text{rot\_}\beta^t), \text{pos\_x}^t, \varepsilon_x^t) \\
\text{pos\_y}^{t+1} &= f(1.5 \cdot \sin(\text{rot\_}\alpha^t), \text{pos\_y}^t, \varepsilon_y^t) \\
\text{pos\_z}^{t+1} &= f(1.5 \cdot \sin(\text{rot\_}\alpha^t), \text{pos\_z}^t, \varepsilon_z^t) \\
\text{rot\_}\alpha^{t+1} &= f(\text{hue\_b}^t, \text{rot\_}\alpha^t, \varepsilon_\alpha^t) \\
\text{rot\_}\beta^{t+1} &= f(\text{hue\_o}^t, \text{rot\_}\beta^t, \varepsilon_\beta^t) \\
\text{rot\_s}^{t+1} &= f(\text{atan2}(\text{pos\_x}^t, \text{pos\_y}^t, \text{rot\_s}^t, \varepsilon_{rs}^t) \\
\text{hue\_s}^{t+1} &= f(2\pi - \text{hue\_b}^t, \text{hue\_s}^t, \varepsilon_{hs}^t) \\
\text{hue\_b}^{t+1} &= \text{hue\_b}^t + \varepsilon_b^t \\
\text{hue\_b}^{t+1} &= f(g(i), \text{hue\_o}^t, \varepsilon_{ho}^t)
\end{aligned}
\tag{26}
$$

All the noise variables, $\varepsilon$, are independently $\mathcal{N}(0, 10^{-2})$ distributed for the position and $\mathcal{N}(0, (0.15)^2)$ distributed for the angles. The $g$ function in the final line maps the object shapes to specific values detailed in Table 4.

The object shape is changed in each time step with a probability of $0.05$. If it is changed, a new shape is sampled uniformly over the 7 shapes.

They then sample for each time step the intervention targets $I_i^{t+1} \sim \text{Bernoulli}(0.1)$. If a causal variable is intervened on, it is replaced with a random sample from $U(-2, 2)$ for the position values or $U(0, 2\pi)$ for the angles. For the object shape a uniform distribution over the 7 shapes is used. They run this simulation for $250,000$ steps, which is the full dataset.

We use the already generated dataset downloaded from https://zenodo.org/records/6637749#.YqcWCnVBxCA, which is available under the Creative Commons Attribution 4.0 International license.

**Target Labels** We employ the same tactic as before to create the target labels. The true latent variables $G_i$, are binarized by looking at the empirical range over the whole dataset, and then thresholded at the midpoint. The target label is then determined by sampling three dimensions in $\{1, \ldots, 7\}$ and setting the label to 1 if at least two of the three latent variables labels are 1.

Table 4: Output of the $g$ function for each object shape. The avg function for angles is defined as $\text{avg}(\alpha, \beta) = \text{atan2}\left(\frac{\sin(\alpha) + \sin(\beta)}{2}, \frac{\cos(\alpha) + \cos(\beta)}{2}\right)$

| Object shape | Object hue goal |
|---|---|
| Teapot Size | $0$ |
| Armadillo | $\frac{2\pi}{5}$ |
| Hare | $\text{avg}(\text{hue\_s}^t, \text{hue\_b}^t)$ |
| Cow | $\frac{4\pi}{5}$ |
| Dragon | $\pi + \text{avg}(\text{hue\_s}^t, \text{hue\_b}^t)$ |
| Head | $\frac{6\pi}{5}$ |
| Horse | $\frac{8\pi}{5}$ |

Table 5: Architecture details for the encoder and decoder used in the Temporal Causal3DIdent experiments.

|  | Layer | Feature Dimension (H × W × C) | Kernel | Stride | Activation Function |
|---|---|---|---|---|---|
| Encoder | Conv | $32 \times 32 \times 64$ | 3 | 2 | BatchNorm+SiLU |
|  | Conv | $32 \times 32 \times 64$ | 3 | 1 | BatchNorm+SiLU |
|  | Conv | $16 \times 16 \times 64$ | 3 | 2 | BatchNorm+SiLU |
|  | Conv | $16 \times 16 \times 64$ | 3 | 1 | BatchNorm+SiLU |
|  | Conv | $8 \times 8 \times 64$ | 3 | 2 | BatchNorm+SiLU |
|  | Conv | $8 \times 8 \times 64$ | 3 | 1 | BatchNorm+SiLU |
|  | Conv | $4 \times 4 \times 64$ | 3 | 2 | BatchNorm+SiLU |
|  | Conv | $4 \times 4 \times 64$ | 3 | 1 | BatchNorm+SiLU |
|  | Reshape | $1 \times 1 \times 1024$ | - | - | - |
|  | Linear | $1 \times 1 \times 256$ | - | - | LayerNorm+SiLU |
|  | Linear | $1 \times 1 \times 2 \cdot$`num_latents` | - | - | - |
| Decoder | Linear | $1 \times 1 \times 256$ | - | - | LayerNorm+SiLU |
|  | Linear | $1 \times 1 \times 1024$ | - | - | - |
|  | Reshape | $4 \times 4 \times 1024$ | - | - | - |
|  | Upsample | $8 \times 8 \times 64$ | - | - | - |
|  | ResidualBlock | $8 \times 8 \times 64$ | 3 | 1 | - |
|  | Upsample | $16 \times 16 \times 64$ | - | - | - |
|  | ResidualBlock | $16 \times 16 \times 64$ | 3 | 1 | - |
|  | Upsample | $32 \times 32 \times 64$ | - | - | - |
|  | ResidualBlock | $32 \times 32 \times 64$ | 3 | 1 | - |
|  | Upsample | $64 \times 64 \times 64$ | - | - | - |
|  | ResidualBlock | $64 \times 64 \times 64$ | 3 | 1 | - |
|  | Pre-Activations | $64 \times 64 \times 64$ | - | - | BatchNorm+SiLU |
|  | Conv | $64 \times 64 \times 64$ | 1 | 1 | BatchNorm+SiLU |
|  | Conv | $64 \times 64 \times 3$ | 1 | 1 | Tanh |

### D.3.2 Model Architectures

In this setting we use as the first part of our pipeline both CITRIS [33], in particular CITRIS-VAE, and iVAE [24]. In both models, the encoder and decoder architecture are set to be the same. The encoder is a convolutional neural network, which outputs two parameters per latent variable. These will be the mean and the log of the standard deviation for the normal distribution that models the latent variable. The decoder uses bilinear upsampling and residual blocks to reconstruct the image. The full architecture is described in Table 5.

As an additional optimization step, an autoencoder is pre-trained to map the high-dimensional images to lower-dimensional feature vectors, but without enforcing disentanglement. This is done separately from the main training procedure, as Lippe et al. [33] mention that this improves performance. During training a small amount of Gaussian noise is added to the encodings to prevent a collapse of the encoding distribution. No prior is enforced for this encoder. This autoencoder has 2 ResidualBlocks instead of 1 per resolution in the decoder part. The training hyperparameters are described in Table 6 and the autoencoder is trained using the MSE reconstruction loss.

**CITRIS-VAE** CITRIS [33] allows for multidimensional causal variables. The number of latent variables is allowed to be bigger than the number of causal variables $d$, but the model subdivides the latent variables into $d$ possibly uneven blocks that get mapped to the causal variables. This means that an assignment $\psi \colon \{1, \ldots, k\} \to \{1, \ldots, d\}$ is learned between the learned latent variables and the learned causal variables, so that multiple latent variables can represent one causal variables. This is done by assuming that each $\psi(i)$ follows a Gumbel-Softmax distribution and we learn the continuous parameters that govern these distributions. CITRIS-VAE optimizes the following objective:

$$\text{ELBO}(\theta, \varphi, \gamma) = - \underset{z^{t+1} \sim q_\theta(\cdot | x^{t+1})}{\mathbb{E}} \left[ \log p_\theta(x^{t+1} \mid z^{t+1}) \right]$$

$$+ \underset{\substack{z^t \sim q_\theta(\cdot | x^t) \\ \pi \sim \text{GS}(\gamma)}}{\mathbb{E}} \left[ \sum_{i=1}^d \text{KL}(q_\theta(z^{t+1}_{\psi(i)} \mid x^{t+1}) \parallel p_\varphi(z^{t+1}_{\psi(i)} \mid z^t, I^{t+1}_i)) \right].$$

Table 6: The hyperparameters used for the training of autoencoder used as preprocessing by both the CITRIS-VAE and iVAE.

| Hyperparameter | Value |
|---|---|
| Batch Size | 512 |
| Optimizer | Adam [27] |
| Learning rate | 1e-3 |
| Learning rate scheduler | Cosine Warmup (100 steps) |
| Number of latents | 32 |
| Gaussian noise $\sigma$ | 0.05 |
| Number of epochs | 1000 |

Table 7: The hyperparameters used for the training of both the CITRIS-VAE and iVAE models.

| Hyperparameter | Value |
|---|---|
| Batch Size | 512 |
| Optimizer | Adam [27] |
| Learning rate | 1e-3 |
| Learning rate scheduler | Cosine Warmup (100 steps) |
| KL divergence factor $\beta$ | 1.0 |
| KL divergence factor $\psi_0(\lambda)$ | 0.01 |
| Number of latents | 32 |
| Number of epochs | 600 |
| Target classifier weight | 2.0 |
| Gumbe Softmax temperature | 2.0 |

Here $p_\theta$ models the encoder, $q_\theta$ the decoder, $p_\varphi(z^{t+1} \mid z^t, I^{t+1})$ the transition prior and GS is the Gumbel-Softmax distribution of the causal variables between time steps given the intervention targets. Finally, $\pi$ is the target assignment between learned encoding variables and the causal variables. During training a latent-to-causal variable assignment is sampled, while during testing the argmax is used. The transition prior $p_\varphi$ is learned by an autoregressive model, which for each $z^{t+1}_{\psi(i)}$ takes $z^t, I^{t+1}_i$ and $z^{t+1}$ as inputs and outputs a Gaussian random variable. The autoregressive model follows a MADE architecture [23], with 16 neurons per layer for each encoding, and the input to these neurons are the features of all previous encodings. The prior is 2 layers deep, and uses SiLU activation functions. Finally, a small network is trained to predict the intervention targets, given $z^t$ and $z^{t+1}_{\psi(i)}$ for each $i = 1, \ldots k$.

**iVAE**  To adapt the iVAE model [24] for this setting, the auxiliary variable $u$ is given by the previous observation $x^t$ and intervention targets $I^{t+1}$. Another alteration that is made, is that the prior with the iVAE model only conditions on $(x^t, I^{t+1})$. The main difference between iVAE and the CITRIS-VAE is the structure of the prior $p(z^{t+1} \mid z^t, I^{t+1})$. Another difference is that no target assignment is learned during the training, but only after. For iVAE a 2-layer MLP with hidden dimensionality of 128 is used for the transition prior.

All the hyperparameters for training CITRIS-VAE and iVAE are reported in Table 7.

### D.4  Concept-Based Models

Concept-bottleneck models were introduced and popularized by Koh et al. [28]. They offer a deep learning architecture that is inherently interpretable, by letting the input go through a concept layer. Each node in that layer indicates either the value of a predefined concept when using continuous concepts if the concept is present or not when the concepts are binary. Then, a linear layer from these concepts to the final target layer is applied. Formally, given the observational data $\{(x_i, c_i, y_i)\}_{i=1}^n$, where $x_i \in \mathcal{X}$, $c_i \in \mathcal{G}_i$, $y_i \in \mathbb{R}$, a concept encoder $p_\theta(c \mid x)$ and a label predictor $p_\varphi(y \mid c)$ are learned. The total prediction is then performed by combining these 2 probabilities. Different choices are possible for this combination, which lead to different concept-based models. We will discuss the Concept-Bottleneck Model (CBM) [28], the Concept-Embedding Model (CEM) [57] and the HardCBM [17], which are all three used in the binary versions of our experiments. For all three models, we use the same encoders as in the VAE models, to make the comparison fair.

The linear layer allows the user to construct explanations for each prediction. When a prediction is made, we can look at the concept activations, $\hat{c}_i$, and the corresponding weights in the final layer $\psi$. An explanation is given by the set

$$\{(\psi_i, \hat{c}_i) \mid i \in \{1, \ldots, d\}\}.$$

The interpretation is that $\psi_i$ tells us how important that concept is in general, and $\hat{c}_i$ tells us how active that concept is for this particular prediction.

**CBM**  In the Concept-Bottleneck Model (CBM) [28] the output of the concept layer will be denoted by $g$ and the function from the concepts to the labels will be $f$. The output of $g$ are the logits of that concept being 1, meaning $\sigma(g(x)_i) = p_\theta(c_i = 1 \mid x)$ for $i = 1, \ldots, d$, where $\sigma$ is the sigmoid function. The function $f$ is a linear combination of the probabilities for all the concepts so together $f(\sigma(g(x))) = \psi^\top \sigma(g(x)) = p_{\theta,\psi}(y \mid x)$.

**CEM**  Concept-Embedding Model (CEM) [57] show that the performance of CBMs can be improved by letting the concepts be encoded in a higher dimensional space, i.e., $\hat{c}_i \in \mathbb{R}^k$ instead of $\hat{c}_i \in \mathbb{R}$. These embeddings are then concatenated and fed to the label predictor. For each concept, a function $s_i$ is learned that predicts the concept label from these embeddings. These predictions can then be used for explanations, but the embeddings are passed to the label predictor.

**HardCBM**  CBMs can also be susceptible to concept leakage, when unrelated information is encoded in each concept. HardCBM [17] propose a way to address this issue. In particular, to discourage information from other sources leaking through the concepts, the HardCBM label predictor only takes binary values from the concept layer as input. To make a prediction, we have to marginalize over the possible concepts, $p_{\theta,\psi}(y \mid x) = \mathbb{E}_{c \sim p_\theta(c|x)}[p_\varphi(y \mid c)]$. To model possible dependencies between the concepts in the distribution of $X \mid C$, the conditional distribution is decomposed as

$$p_\theta(c \mid x) = \prod_{i=1}^{d} p(c_i \mid x, c_1, \ldots c_{i-1}).$$

In the HardCBM such a decomposition can be achieved by using an autoregressive architecture.

In all the experiments, each model is trained for 100 epochs, with batch sizes of 256 and optimised by Adam with a learning rate of 1e-3.

### D.5  Performance metrics

To assess the performance of our estimator we report the mean permutation error of the estimated permutation (MPE), together with the execution time. Whenever the concepts are continuous, we also report the $R^2$-score. Some of the alignment learning baselines are created by regressing every input variable onto every output variable and using the individual $R^2$-scores to extract an alignment. In those cases, we first match the encodings and ground truth causal variables according to the highest $R^2$-scores. The $R^2$-score that is reported is the average of the $R^2$-scores that are chosen to be matched. This is also referred to as the $R^2$-score on the diagonal. This gives

$$R^2 = \frac{1}{d} \sum_{i=1}^{d} \left( 1 - \frac{\|\mathbf{C}_i - \alpha_i\left(\mathbf{M}_{\widehat{\pi}(i)}\right)\|^2}{\|\mathbf{C}_i - \overline{\mathbf{C}_i}\|^2} \right), \qquad \overline{\mathbf{C}_i} = \frac{1}{n} \sum_{\ell=1}^{n} C_i^{(\ell)}.$$

Here, $\alpha$ and $\widehat{\pi}$ are the estimated alignment map and estimated permutation. The alignment map is applied to the whole vector $\mathbf{M}_j$. As in the main text, $\mathbf{C}$ and $\mathbf{M}$ are matrices where the rows are the data samples and the columns the concepts and encodings respectively.

In the downstream classification experiments we also report various other metrics. The mean accuracy of all the concepts and the accuracy of the downstream label prediction is reported. Furthermore, two additional metrics for concept-based models are reported, the OIS and NIS metric. The exact definitions can be found in Zarlenga et al. [58]. The OIS metric measures how well concept $i$ can be used to reconstruct concept $j$. The NIS metric measures how much information about concept $i$ is contained in the other concepts jointly. A low OIS metric, but high NIS metric indicates that it is not possible to reconstruct one concept from any of the others individually, but combining the concept embeddings would allow to reconstruct the concept from the others.

**Estimator settings**  We use various settings of our estimator, to assess if there are particular advantages for certain versions. The versions that we use are

- **Linear**, no feature map is applied.

- **Random Fourier Features**, we sample 8 random Fourier features from the RBF kernel.

- **Spline**, we calculate cubic spline features with 4 knots.

- **Laplacian**, we use the Laplacian kernel with $\min\{n, 20\}$ components. If the number of components is less than the number of datapoints a Nyström sampling procedure is used.
- **Two stage** we apply a two stage approach, where we use $20\%$ of the data to estimate the permutation using no additional features and then use the rest of the data to perform ridge regression with cubic spline features using $4$ knots.

We define an array of regularization parameters and report the results of the best choice for each $n$. The parameters that we consider are $\lambda \in \{0.0001, 0.0005, 0.001, 0.005, 0.01, 0.05, 0.1, 0.2\}$.

**Alignment Learning Baselines.** We estimate the permutations using the Pearson and Spearman correlations. We also add another baseline for the experiments with the encodings learned by a VAE. This baseline is given by training multiple neural networks. Each neural network takes an individual encoding as input and tries to predict all the causal variables. The individual $R^2$-scores are used to construct a matching again. This neural network is a 2-layer MLP with $128$ hidden nodes in each layer and tanh is used as an activation function. The network is trained for $100$ epochs with Adam and a learning rate of 4e-3. We considered using $32$ and $64$ hidden nodes per layer, but concluded that $128$ nodes per layer offers the best baseline.

In the Temporal Causal3DIdent dataset experiments, both VAE models learn groups of latent variables that are matched with a causal variable. To use the Pearson and Spearman correlations in this case, we first sum the latent variables in the groups and then calculate the correlation coefficients.

The baselines in the binary versions of the experiment in the two DL datasets are given by the three concept-based models discussed in Section D.4.

# E  Additional Results

In this section, we show all the results obtained in the experiments with the Toy Dataset, Action/Temporal Sparsity datasets and Temporal Causal3DIdent dataset.

## E.1  Toy Dataset

**Continuous Concepts.** We provide plots for experiments performed with linear features, spline features, random Fourier features and kernels. The concepts are continuous in these plots. All results are depicted in Figures 6–14.

In each version of our estimator we see that the MPEs are good, especially in the misspecified setting. It is interesting to note that the estimator does not perform well in the wellspecified case, when the regularization parameter is not tuned correctly. This can be explained by the non-invertability of the functions in this setting. This makes the identification of each matching more noisy and difficult, which can also be noted by the fact that the Pearson and Spearman correlation approaches are not able to find the correct permutation, while our estimator is able to do it correctly.

The non-invertibility of the functions in the wellspecified setting also explains why the $R^2$ scores in that setting are worse than the $R^2$-scores in the misspecified setting. Another reason for that observation is that the norms of the true parameters have to be quite large, to strictly adhere to the assumptions of our theoretical results. This increases the variability of the output data by a large amount and makes regression more difficult.

Finally, we also see that our estimator does work well, even when the correlation is high, but the regularization parameter has to be tuned correctly. This does come at the cost of an increased computation time.

**Binary Concepts Ablation.** We also perform an ablation study, with binary concepts. How the continuous scores are binarized is explained in Appendix D.1 and sampling the diffeomorphisms from the functions depicted in Figure 4. In these experiments we added the CBM [28], CEM [57] and HardCBM [17] as baselines. We report the concept accuracy, label accuracy, OIS and NIS scores [58] and execution times. The results are depicted in Figures 15–19. The ablation comes from the fact that the correlation in the training set differs from the correlation in the test set; in the former it is $\rho = 0.5$ and in the latter it is $\rho = 0$. This is done to mimic the situation where a classifier could pick up on spurious correlations in the train set that are not present in the test set. We see that our

estimator performs well in this case in terms of concept accuracy and label accuracy, and better than the concept based-models whenever the regularization parameter is set to be not too large. Our estimator performs similarly as the concept-based models in terms of OIS, but consistently scores better when considering the NIS metric.

It is interesting to note that CEM [57] performs the worst in terms of the NIS and OIS metrics. An explanation for this is that it has the capacity to store more information about the other concepts in the embeddings. CBM [28] and HardCBM [17] suffer less from this problem, as the concepts are stored as one-dimensional objects. All concept-based models perform similarly in terms of label and concept accuracy.

One caveat is that the training times of our estimator do increase faster than the training times of the concept-based models for increasing number of dimensions. We think this can be attributed to the fact that to perform the logistic alignment learning procedure, a loop over all dimensions has to be done. In each step one logistic Group Lasso regression is performed. In the continuous case, this was not necessary, as we could see the output as multi-dimensional, and all regressions could be done in parallel. In principle this should be possible for the logistic Group Lasso as well, but implementing this parallel computation was beyond the scope of this paper. The implementation of the logistic Group Lasso we use, does not have this feature for multiple labels.

**Mixtures of Causal Variables.** We conducted an ablation with the synthetic toy dataset to evaluate performance when we have mixtures of causal variables as inputs to the alignment, which simulates the case in which CRL methods might not provide complete disentanglement, but only partial disentanglement. In these experiments, we set the dimension $d$ to an even number and then randomly created $d/2$ pairs of $M_j$ variables. For each pair $(M_i, M_j)$, we then created a new mixture variable $M'_j = (1-a)M_i + aM_j$, where $a$ represents the mixing parameter. We then used these $M'_j$ variables as input to the alignment estimation step. We ran 10 random repetitions for each setting and show the permutation error for the original variables in Figure 20. We additionally show a paired mean permutation error, where we consider the estimated permutation correct if the matching identifies the pairs correctly. These results are shown in Figure 21.

What we see is that our estimator keeps performing perfectly up until slightly below $a = 1/2$, but afterwards does not get the permutation right. However, we do see that the pairs are matched correctly for almost all values of $a$.

### E.2 Action/Temporal Sparsity Datasets and Temporal Causal3DIdent Dataset

**Continuous Concepts.** Here, we report all results obtained in the Action/Temporal Sparsity datasets and Temporal Causal3DIdent dataset. The MPEs are reported in Table 8, the $R^2$-scores in Table 9 and the execution times are plotted in Figure 22.

In terms of MPEs, we always see that a version of our estimator performs the best or as good as the best in each of the datasets for each number of training data samples considered. For the $R^2$ scores, we see that we perform well in some cases in the low data regime, but when using all the data available, the neural network often performs the best. This does come at a computational cost, where the neural network approach requires two to three orders of magnitude more computation time.

It is interesting to note that the estimator typically works better for the more advanced models developed. This can be explained by the fact that these models achieve a better disentanglement, which should make it easier to find the correct matching between the encodings and the causal variables.

**Binary Concepts.** We create binary concepts and labels in a similar fashion as detailed in Appendix D.1. The number of active columns selected is now fixed at $k = 3$ and at least two of them have to be non-zero for the label to be one. The permutation errors are also reported in this case and can be found in Table 10. The baselines are CBM [28], CEM [57] and HardCBM [17]. The concept accuracy and label accuracy are reported in Table 11, and the OIS- and NIS-metrics in Table 12. The execution times are plotted in Figure 23.

We see that the standard concept-based models perform worse with respect to all metrics with only a few labels. However, in the action and temporal datasets, they quickly perform well in terms of concept accuracy and OIS. Our estimator performs especially well when using non-tabular data, such

as in the Temporal Causal3Dident dataset. This shows the added benefit of the CRL phase of our framework.

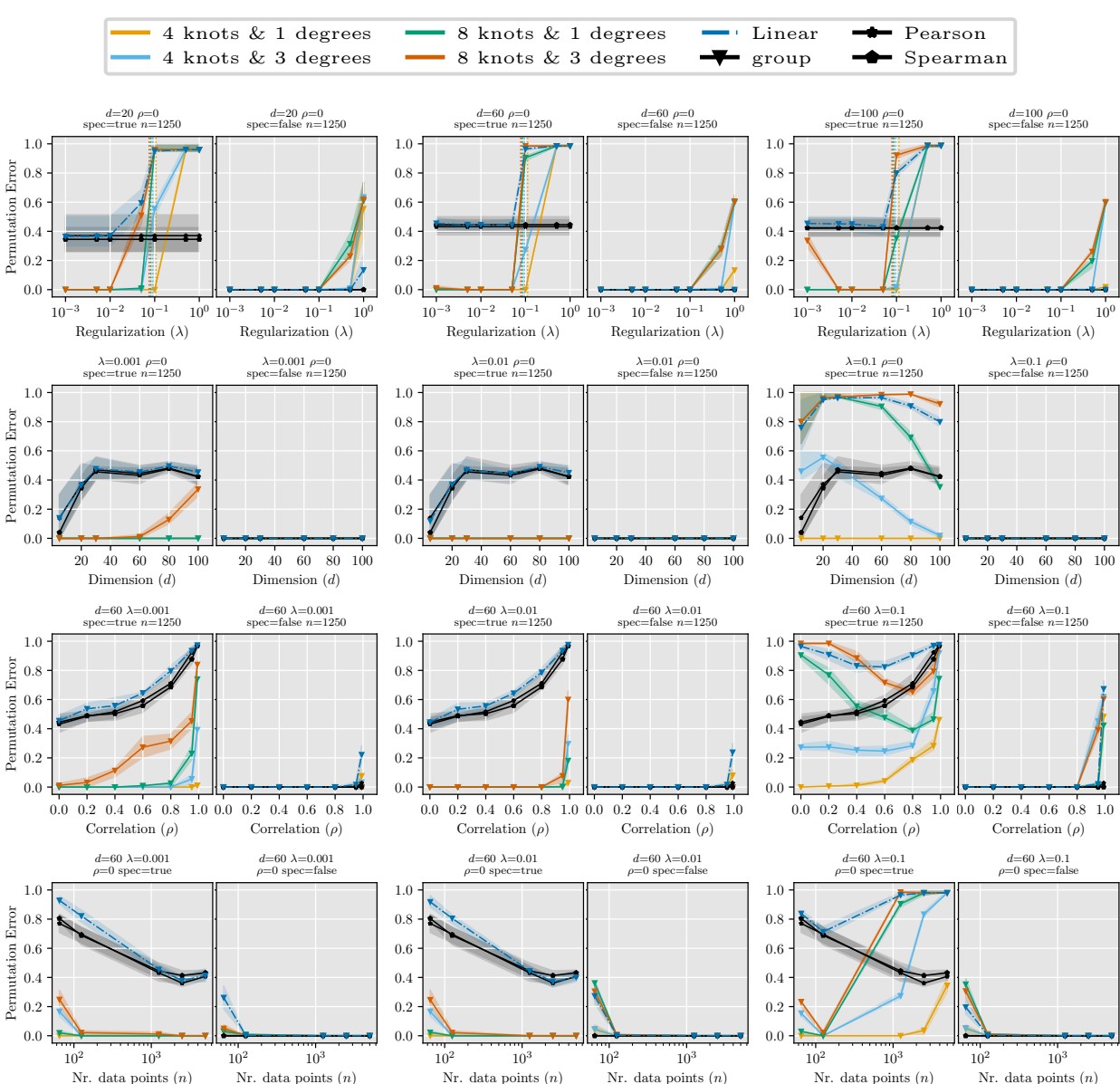

Figure 6: Permutation Error using Spline Features for all parameters considered in the experiments. Each plot is paired, where the left is wellspecified case and the right is the misspecified case.

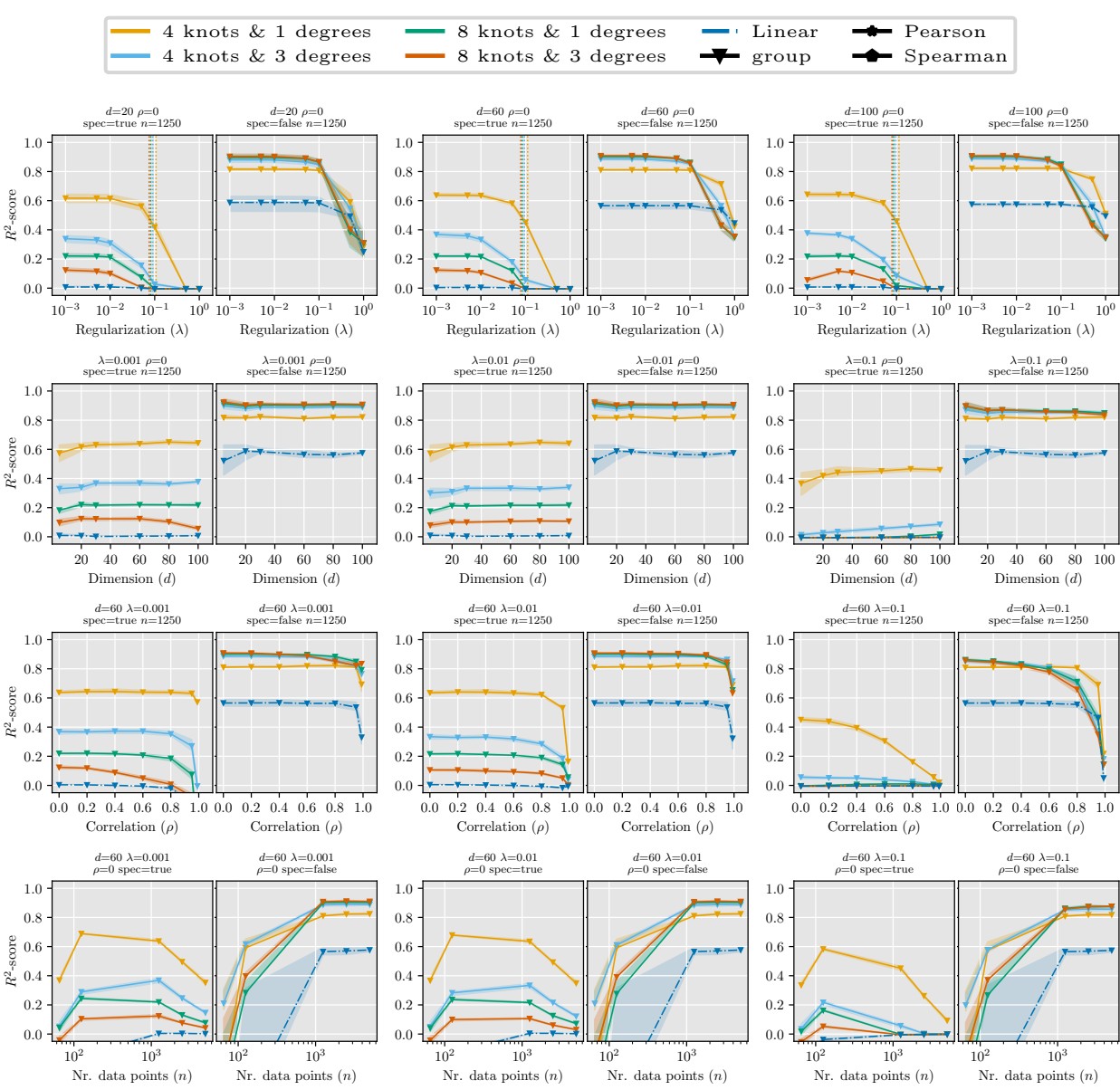

Figure 7: $R^2$-score on the diagonal using Spline Features for all parameters considered in the experiments. Each plot is paired, where the left is wellspecified case and the right is the misspecified case.

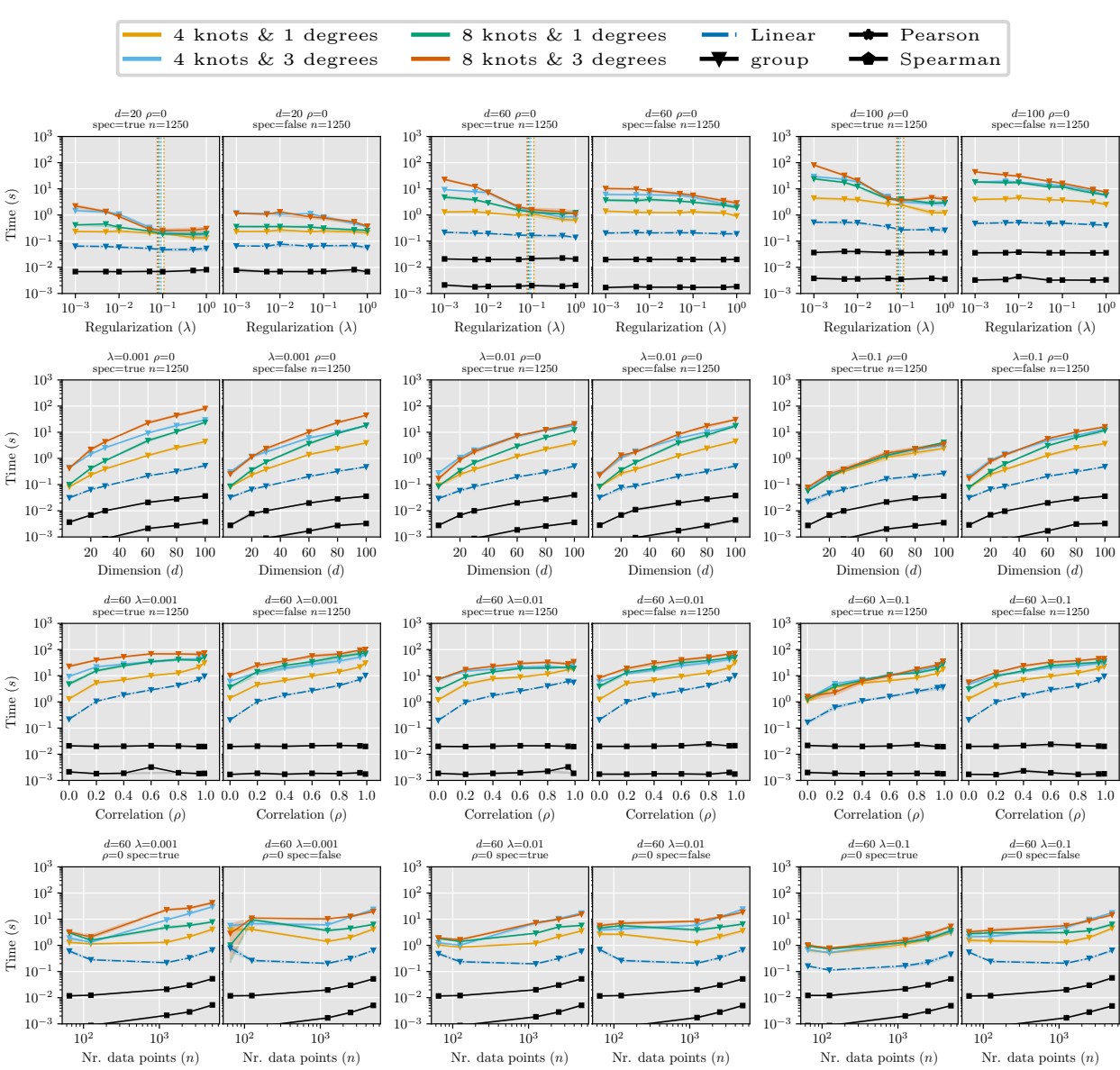

Figure 8: Execution times using Spline Features for all parameters considered in the experiments. Each plot is paired, where the left is wellspecified case and the right is the misspecified case.

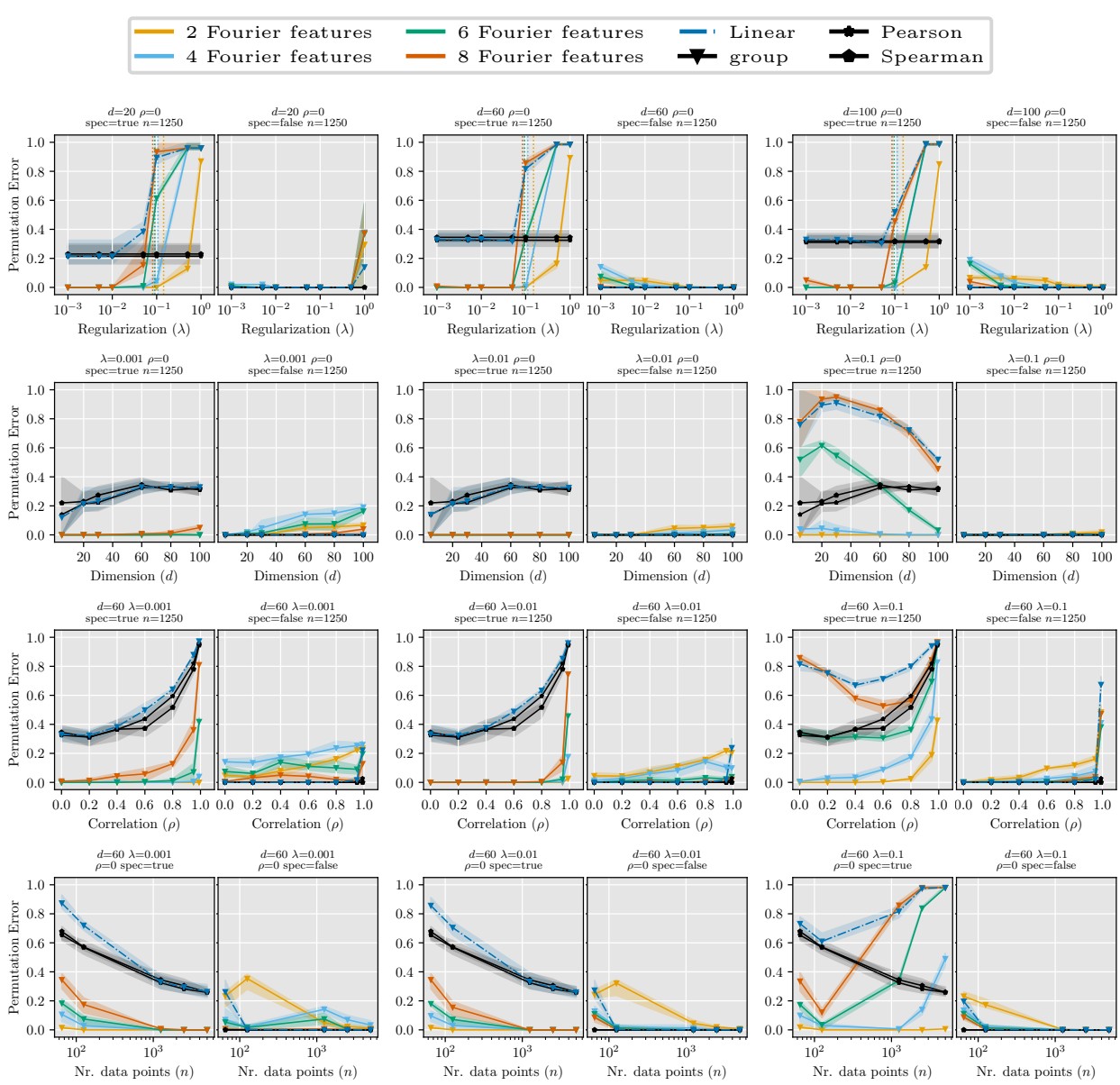

Figure 9: Permutation Errors using Random Fourier Features for all parameters considered in the experiments. Each plot is paired, where the left is wellspecified case and the right is the misspecified case.

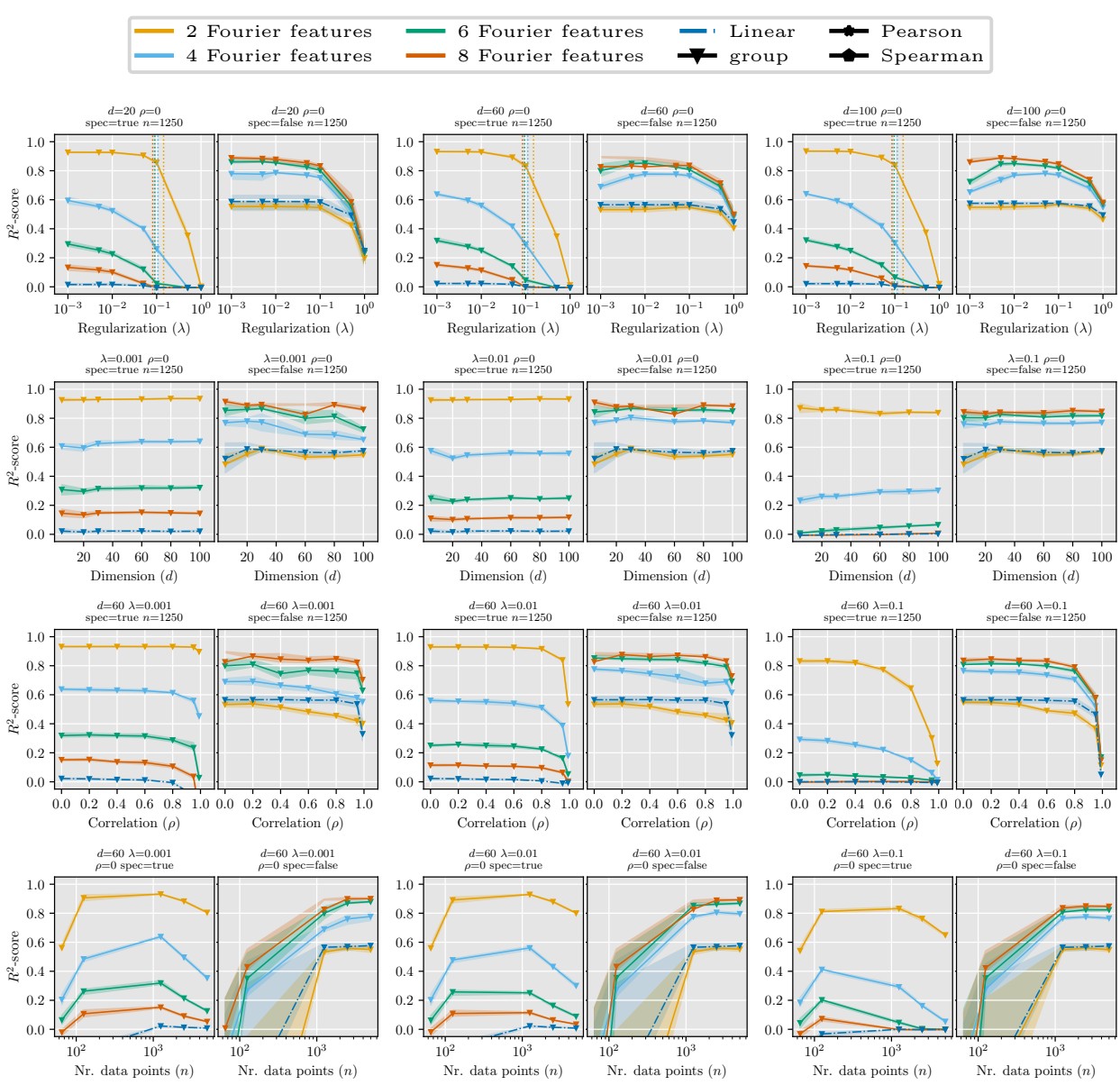

Figure 10: $R^2$-score on the diagonal using Random Fourier Features for all parameters considered in the experiments. Each plot is paired, where the left is wellspecified case and the right is the misspecified case.

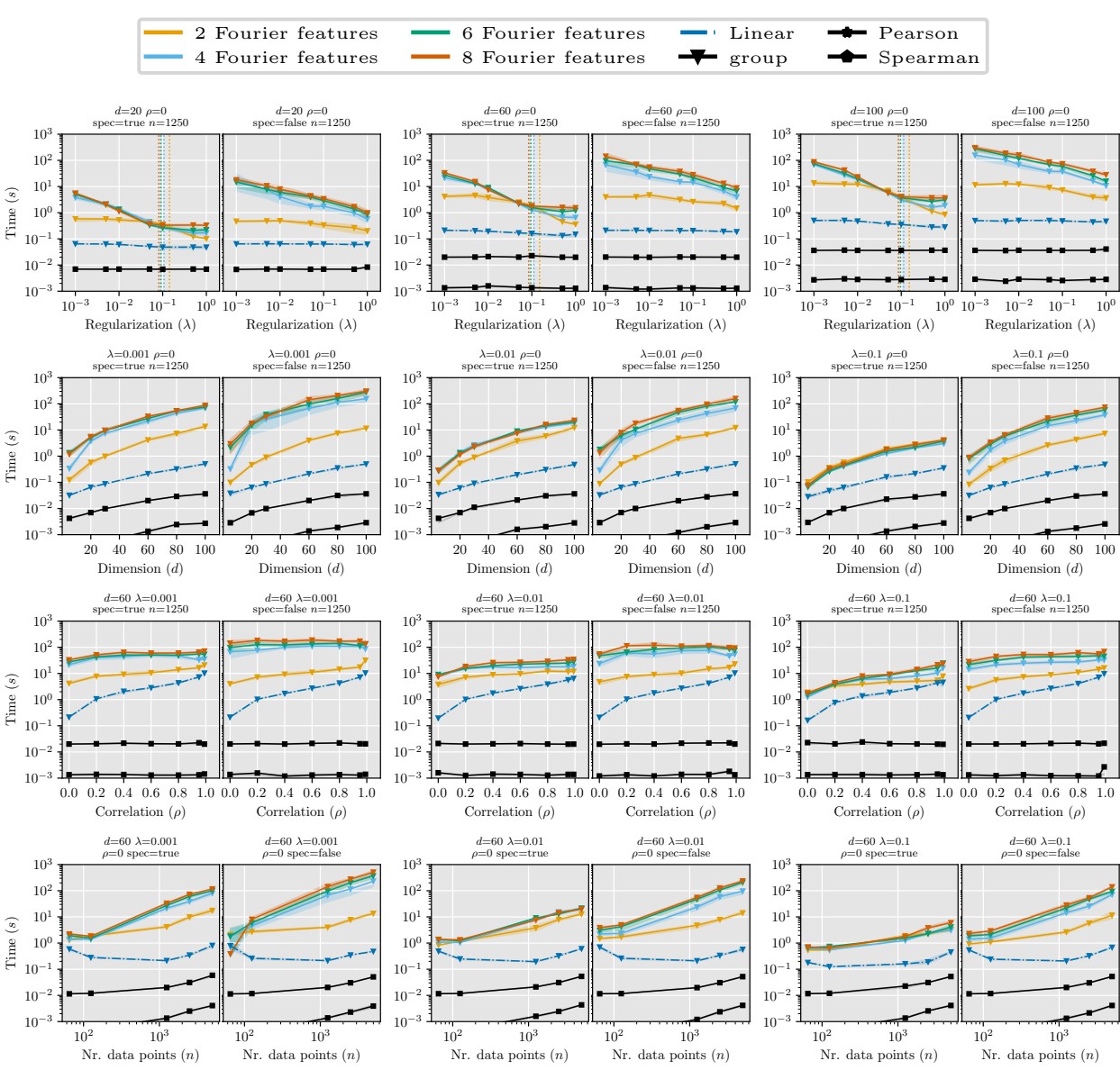

Figure 11: Execution times using Random Fourier Features for all parameters considered in the experiments. Each plot is paired, where the left is wellspecified case and the right is the misspecified case.

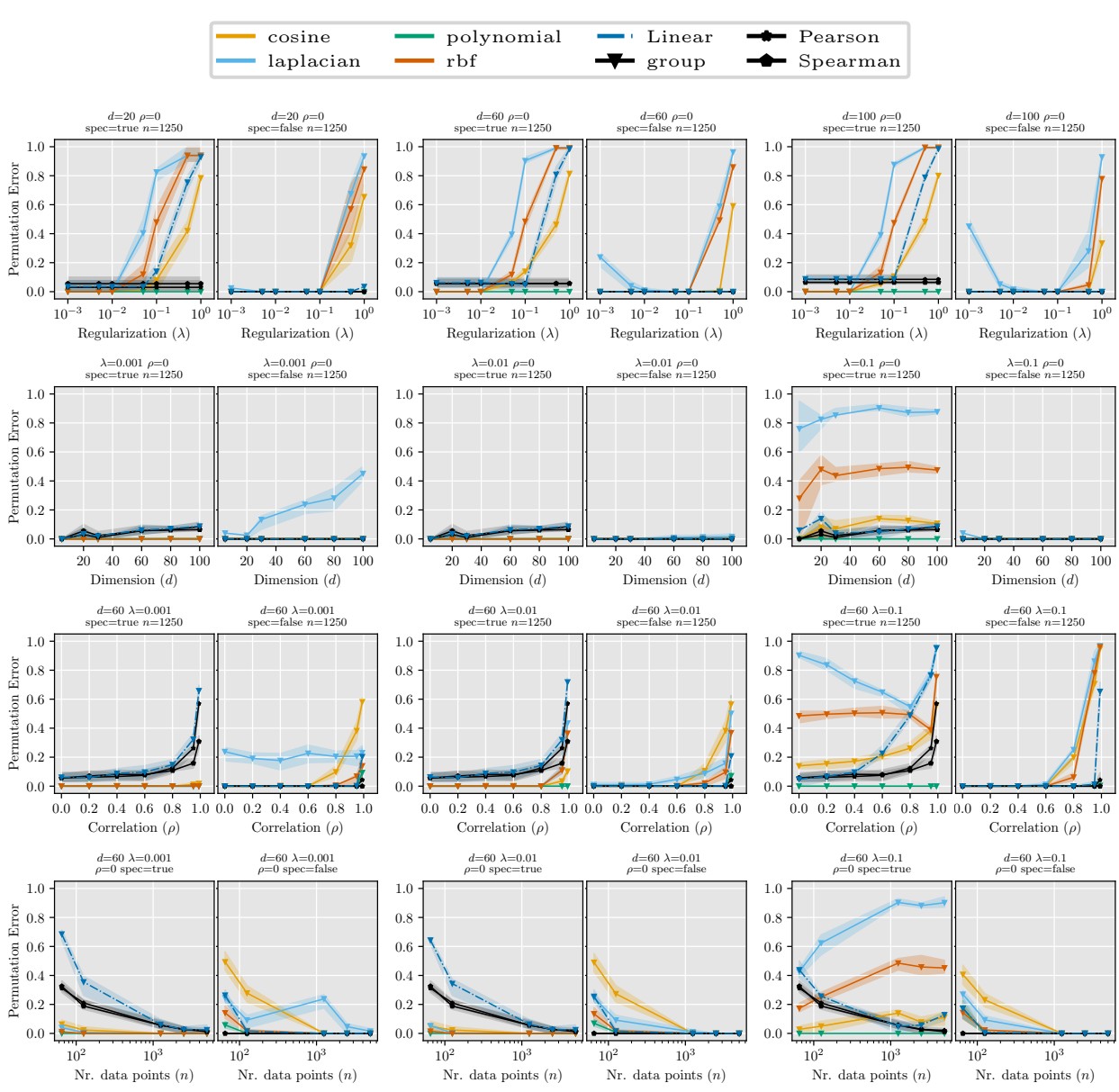

Figure 12: Permutation Errors using Kernels

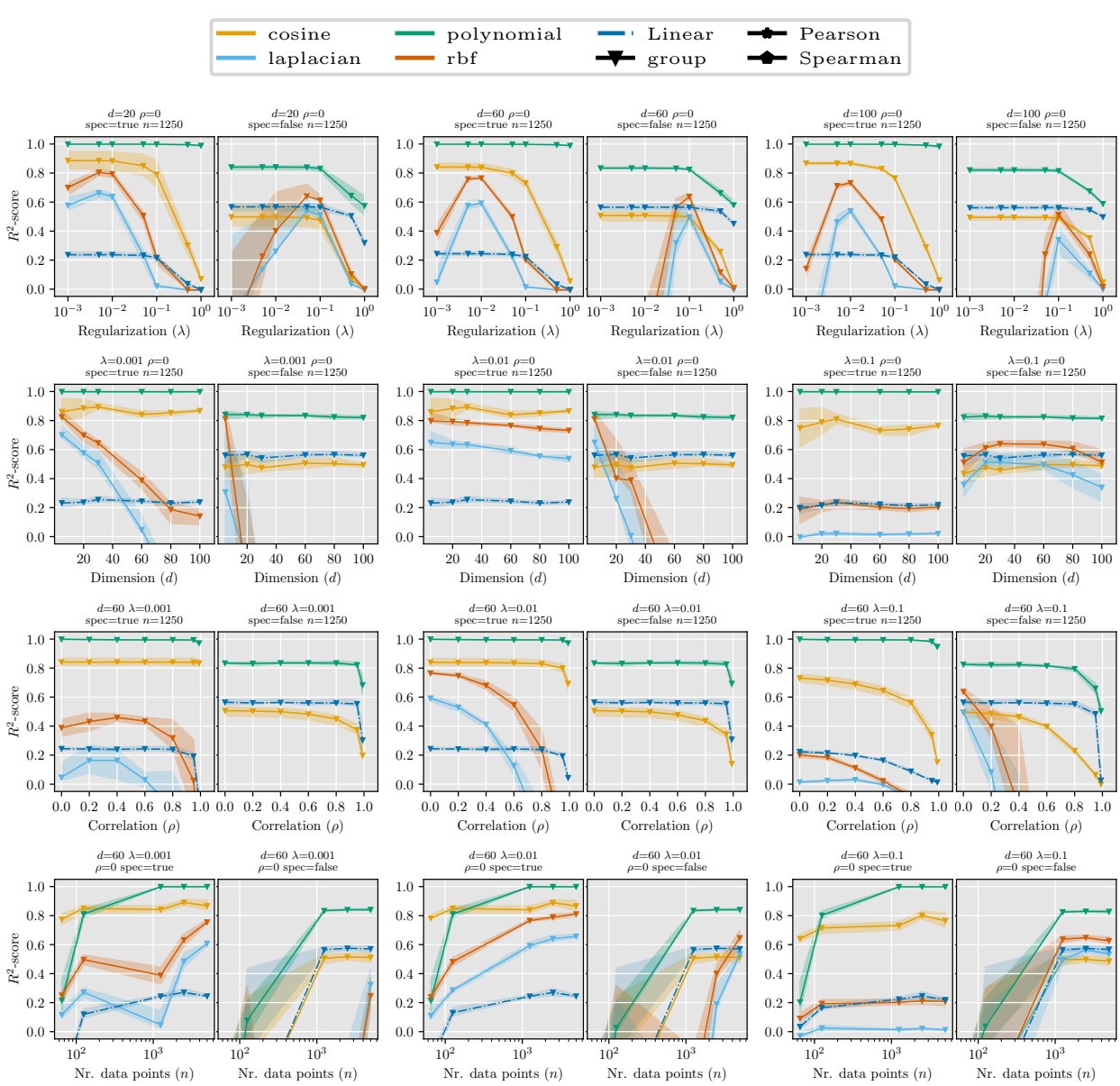

Figure 13: $R^2$-score on the diagonal using Kernels for all parameters considered in the experiments. Each plot is paired, where the left is wellspecified case and the right is the misspecified case.

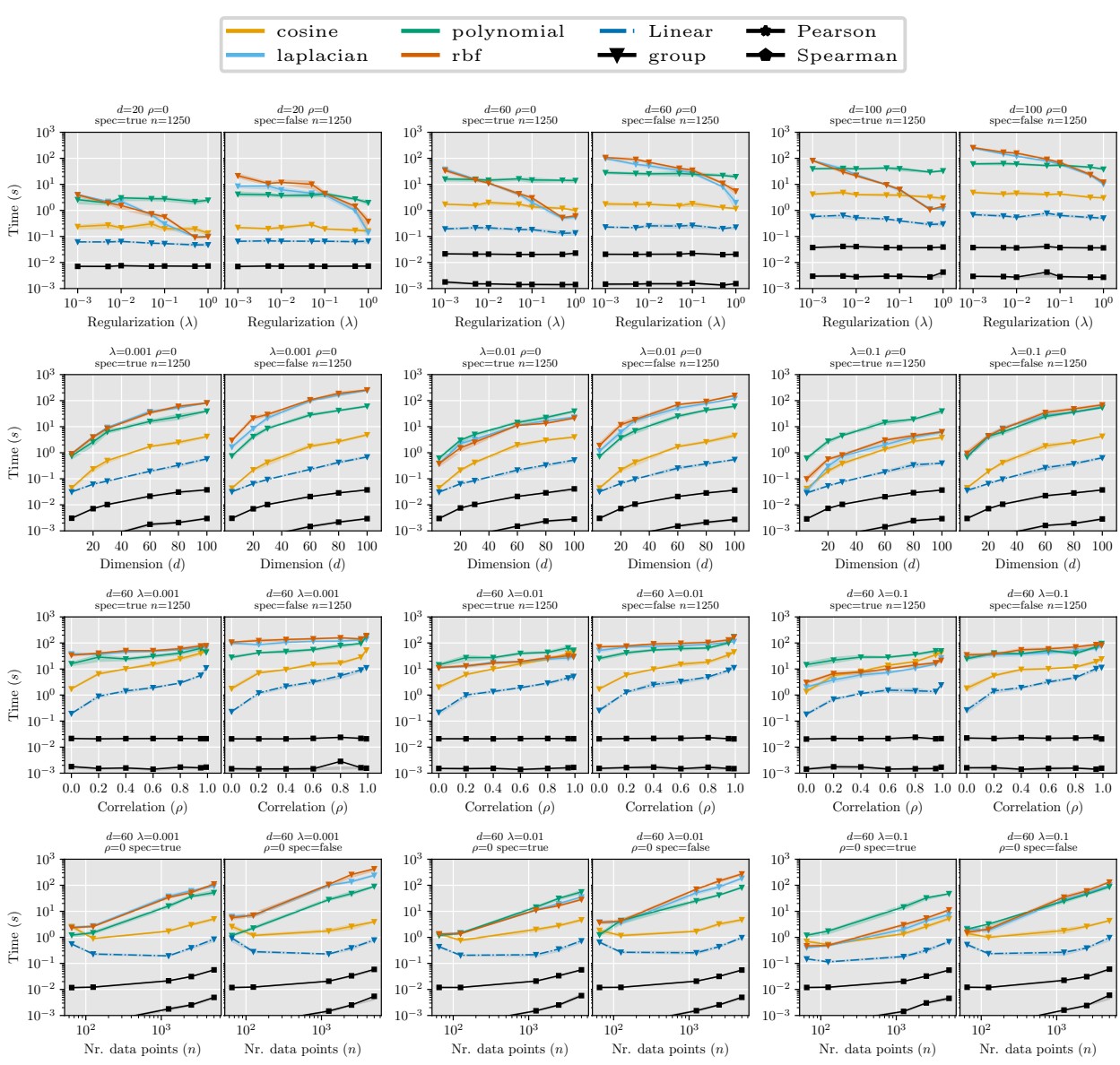

Figure 14: Execution times using kernels for all parameters considered in the experiments. Each plot is paired, where the left is wellspecified case and the right is the misspecified case.

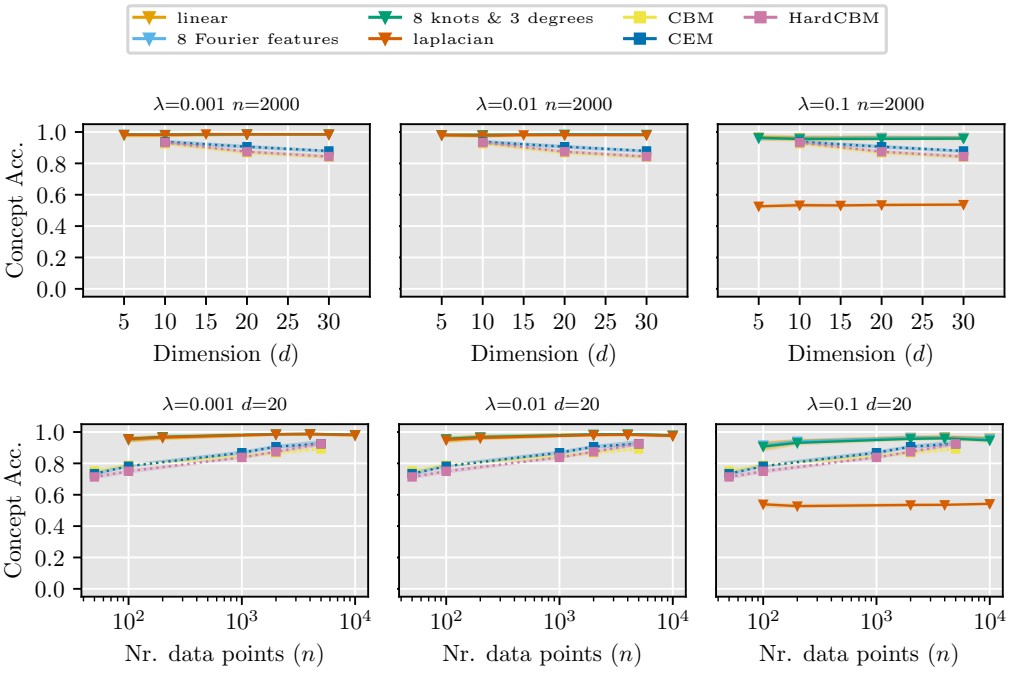

Figure 15: Concept accuracy on the Toy Dataset when the training correlation is $0.5$ and the test correlation is $0$. Several versions of our estimator are compared to the performance of several concept-based models.

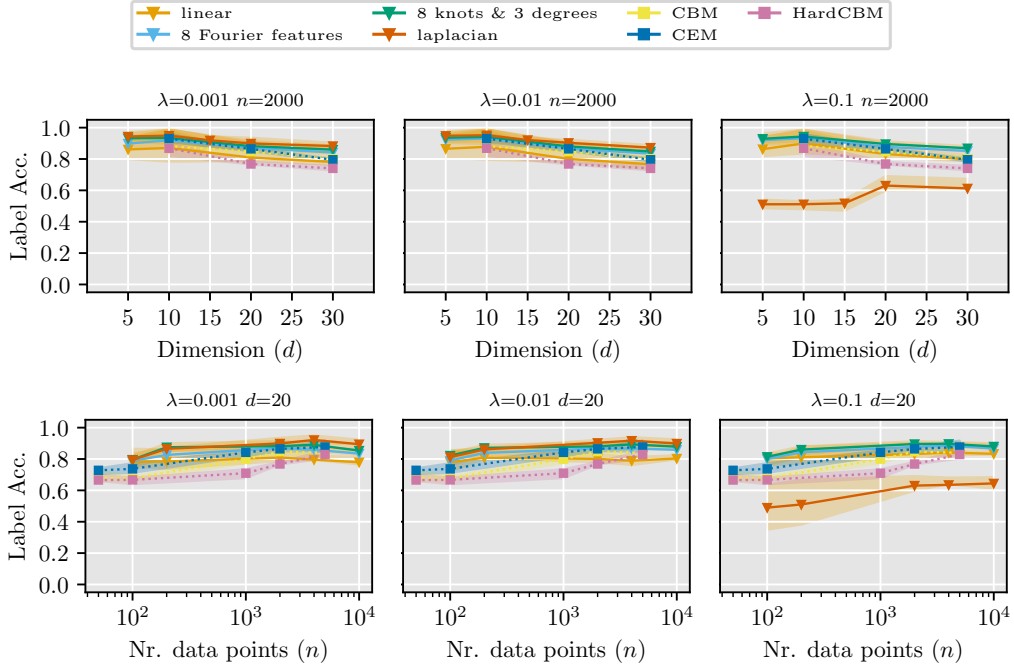

Figure 16: Label accuracy on the Toy Dataset when the training correlation is $0.5$ and the test correlation is $0$. Several versions of our estimator are compared to the performance of several concept-based models.

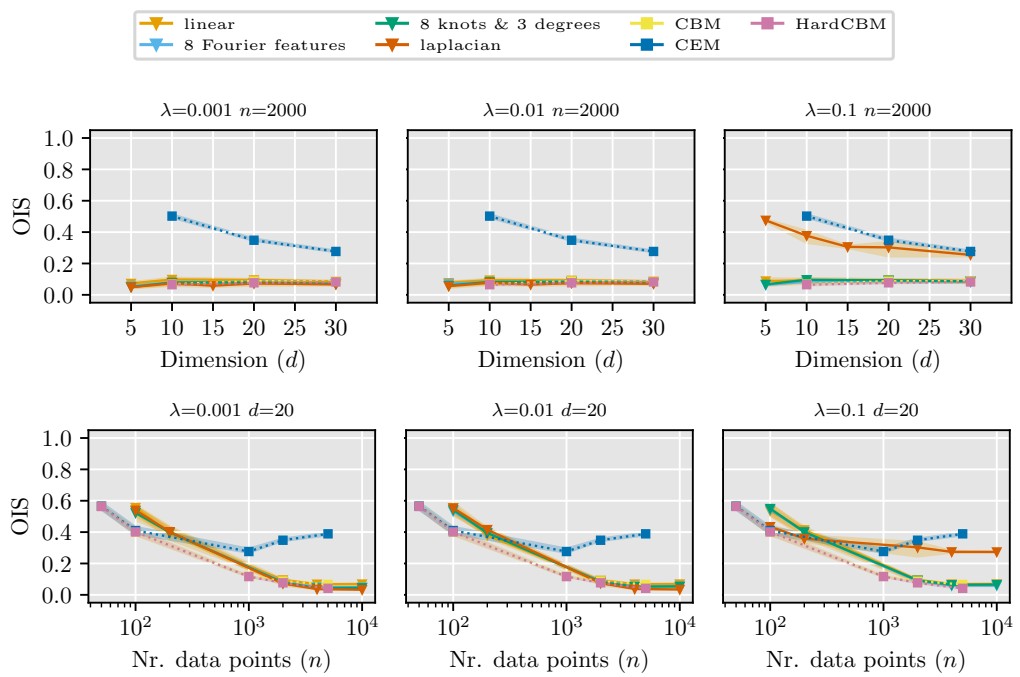

Figure 17: OIS metric on the Toy Dataset when the training correlation is $0.5$ and the test correlation is $0$. Several versions of our estimator are compared to the performance of several concept-based models.

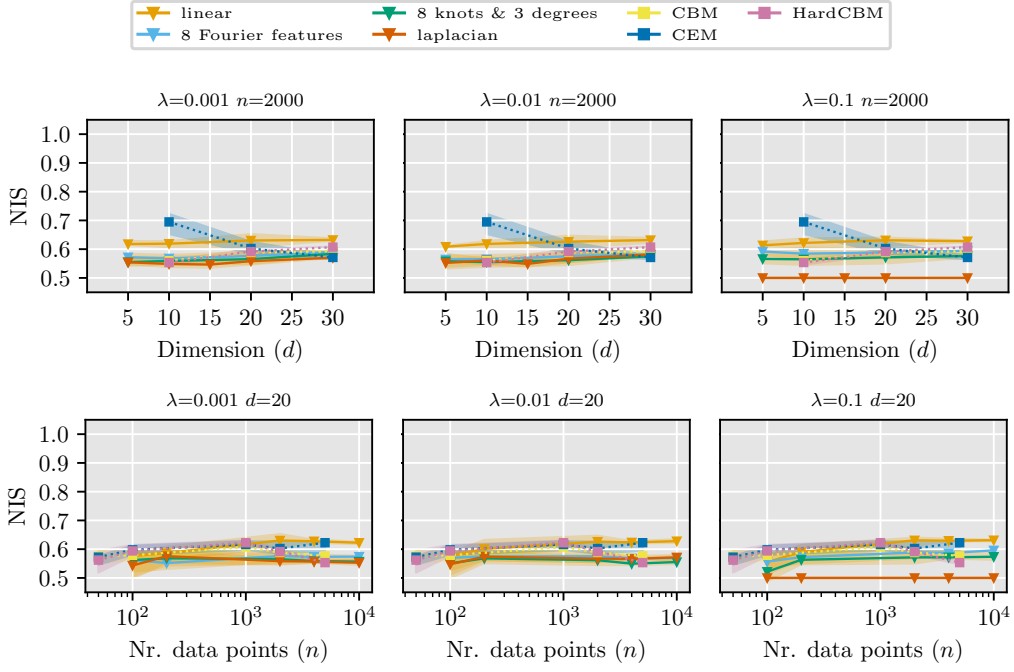

Figure 18: NIS metric on the Toy Dataset when the training correlation is $0.5$ and the test correlation is $0$. Several versions of our estimator are compared to the performance of several concept-based models.

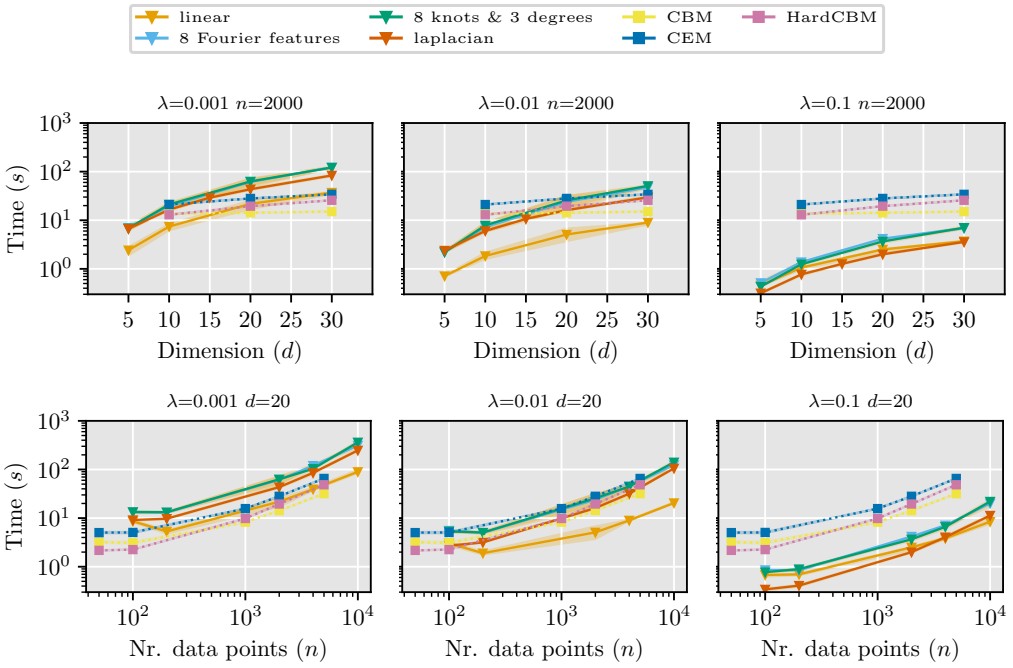

Figure 19: Execution times on the Toy Dataset when the training correlation is 0.5 and the test correlation is 0. Several versions of our estimator are compared to the performance of several concept-based models.

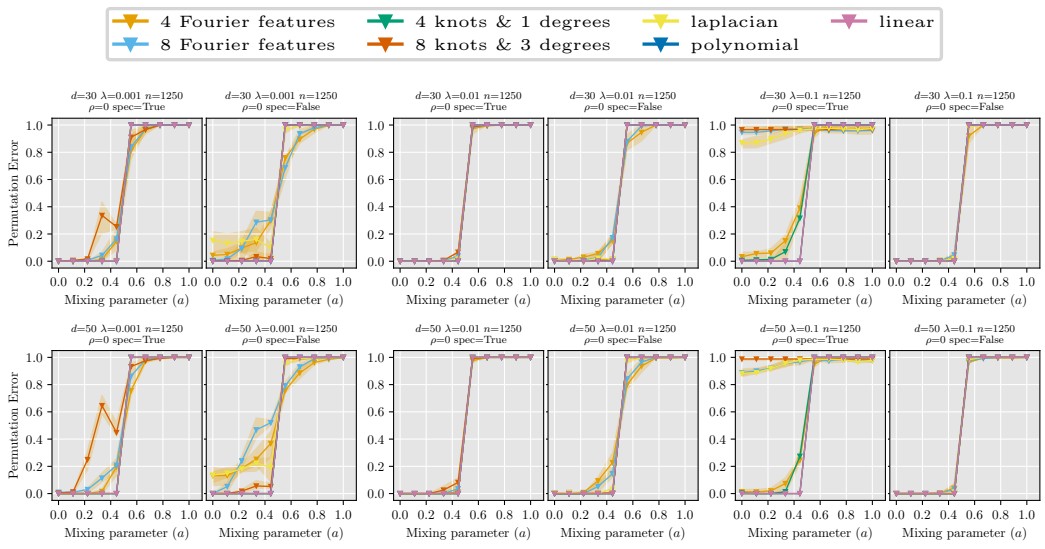

Figure 20: Mean permutation errors on the Toy Dataset when the causal variables are mixed according to a parameter $a$.

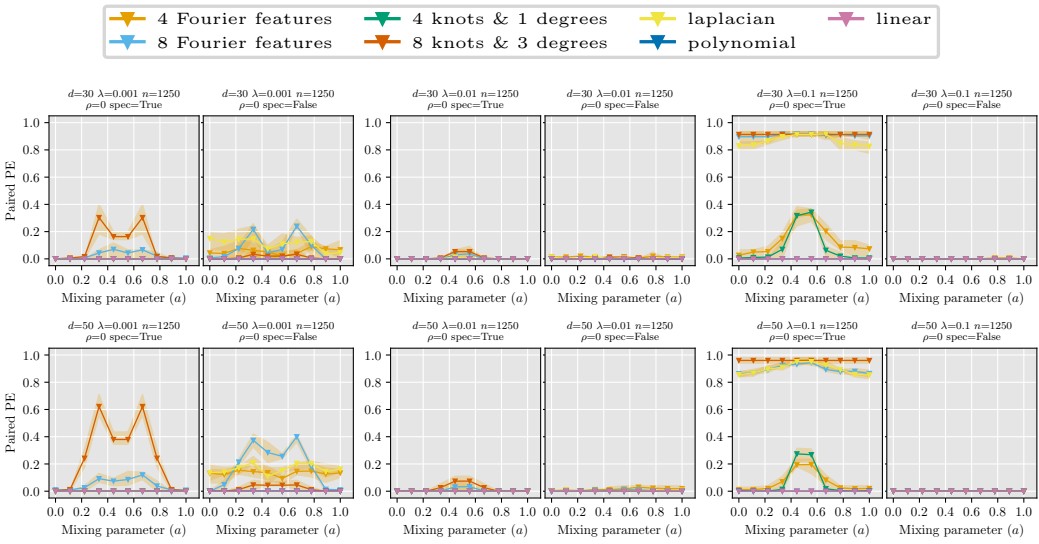

Figure 21: Mean paired permutation errors on the Toy Dataset when the causal variables are mixed according to a parameter $a$.

Table 8: Permutation Errors for the encodings learned in the Action/Temporal sparsity datasets and the Temporal Causal3DIdent datasets. The $(n)$ indicates the number of train and test points used in each column. We report the mean and standard deviation over 50 random seeds and in each column we indicate the best method by **bold**.

| Model | Method | Permutation Error ↓ $(n)$ | | | | | |
|---|---|---|---|---|---|---|---|
| | | 5 | 10 | 20 | 100 | 1000 | 10000 |
| **Action Sparsity Dataset** | | | | | | | |
| DMS-VAE | NN | **0.18** ± 0.02 | **0.03** ± 0.01 | 0.01 ± 0.01 | **0.00** ± 0.00 | **0.00** ± 0.00 | **0.00** ± 0.00 |
| | Pearson | 0.31 ± 0.02 | 0.04 ± 0.01 | **0.00** ± 0.00 | **0.00** ± 0.00 | **0.00** ± 0.00 | **0.00** ± 0.00 |
| | Spearman | 0.28 ± 0.03 | 0.04 ± 0.01 | 0.00 ± 0.00 | **0.00** ± 0.00 | **0.00** ± 0.00 | **0.00** ± 0.00 |
| | Linear | 0.25 ± 0.03 | 0.03 ± 0.01 | **0.00** ± 0.00 | **0.00** ± 0.00 | **0.00** ± 0.00 | **0.00** ± 0.00 |
| | Spline | 0.36 ± 0.02 | 0.28 ± 0.03 | 0.02 ± 0.01 | **0.00** ± 0.00 | **0.00** ± 0.00 | **0.00** ± 0.00 |
| | RFF | 0.47 ± 0.02 | 0.11 ± 0.02 | 0.02 ± 0.01 | **0.00** ± 0.00 | **0.00** ± 0.00 | **0.00** ± 0.00 |
| | Laplacian | 0.64 ± 0.03 | 0.28 ± 0.03 | 0.03 ± 0.01 | **0.00** ± 0.00 | **0.00** ± 0.00 | **0.00** ± 0.00 |
| | Two Stage | 0.46 ± 0.03 | 0.06 ± 0.02 | 0.02 ± 0.01 | **0.00** ± 0.00 | **0.00** ± 0.00 | **0.00** ± 0.00 |
| iVAE | NN | 0.82 ± 0.02 | 0.72 ± 0.02 | 0.53 ± 0.02 | 0.38 ± 0.02 | 0.24 ± 0.01 | 0.20 ± 0.00 |
| | Pearson | 0.76 ± 0.02 | 0.56 ± 0.02 | **0.39** ± 0.02 | 0.35 ± 0.02 | 0.16 ± 0.02 | 0.14 ± 0.01 |
| | Spearman | **0.75** ± 0.02 | 0.64 ± 0.02 | 0.50 ± 0.02 | 0.31 ± 0.03 | 0.18 ± 0.03 | 0.10 ± 0.02 |
| | Linear | 0.78 ± 0.02 | **0.56** ± 0.02 | 0.41 ± 0.02 | **0.20** ± 0.03 | **0.02** ± 0.01 | **0.00** ± 0.00 |
| | Spline | 0.84 ± 0.02 | 0.77 ± 0.02 | 0.61 ± 0.02 | 0.32 ± 0.02 | 0.12 ± 0.02 | 0.00 ± 0.00 |
| | RFF | 0.83 ± 0.02 | 0.76 ± 0.02 | 0.63 ± 0.02 | 0.37 ± 0.03 | 0.19 ± 0.02 | 0.18 ± 0.02 |
| | Laplacian | 0.84 ± 0.02 | 0.78 ± 0.02 | 0.65 ± 0.02 | 0.34 ± 0.02 | 0.03 ± 0.01 | 0.00 ± 0.00 |
| | Two Stage | 0.78 ± 0.02 | 0.65 ± 0.02 | 0.54 ± 0.02 | 0.32 ± 0.03 | 0.21 ± 0.02 | 0.22 ± 0.03 |
| **Temporal Sparsity Dataset** | | | | | | | |
| DMS-VAE | NN | 0.78 ± 0.02 | 0.70 ± 0.02 | 0.48 ± 0.02 | 0.46 ± 0.01 | 0.20 ± 0.00 | **0.00** ± 0.00 |
| | Pearson | 0.68 ± 0.02 | **0.40** ± 0.02 | 0.23 ± 0.03 | 0.20 ± 0.03 | 0.03 ± 0.02 | **0.00** ± 0.00 |
| | Spearman | 0.72 ± 0.02 | 0.53 ± 0.02 | 0.40 ± 0.02 | 0.11 ± 0.02 | **0.00** ± 0.00 | **0.00** ± 0.00 |
| | Linear | **0.67** ± 0.02 | 0.41 ± 0.03 | **0.20** ± 0.03 | **0.07** ± 0.02 | **0.00** ± 0.00 | **0.00** ± 0.00 |
| | Spline | 0.72 ± 0.02 | 0.70 ± 0.02 | 0.47 ± 0.03 | 0.17 ± 0.03 | 0.03 ± 0.01 | **0.00** ± 0.00 |
| | RFF | 0.76 ± 0.02 | 0.59 ± 0.02 | 0.37 ± 0.03 | 0.10 ± 0.02 | 0.03 ± 0.02 | 0.01 ± 0.01 |
| | Laplacian | 0.80 ± 0.02 | 0.67 ± 0.02 | 0.45 ± 0.02 | 0.15 ± 0.03 | 0.01 ± 0.01 | **0.00** ± 0.00 |
| | Two Stage | 0.78 ± 0.02 | 0.59 ± 0.03 | 0.44 ± 0.02 | 0.15 ± 0.02 | 0.01 ± 0.01 | **0.00** ± 0.00 |
| TCVAE | NN | 0.83 ± 0.02 | 0.81 ± 0.02 | 0.65 ± 0.02 | 0.33 ± 0.01 | 0.25 ± 0.02 | 0.60 ± 0.00 |
| | Pearson | **0.74** ± 0.02 | 0.54 ± 0.03 | **0.30** ± 0.03 | 0.36 ± 0.03 | 0.24 ± 0.03 | 0.23 ± 0.02 |
| | Spearman | 0.77 ± 0.02 | 0.61 ± 0.03 | 0.47 ± 0.03 | 0.28 ± 0.03 | **0.14** ± 0.03 | **0.01** ± 0.01 |
| | Linear | 0.76 ± 0.02 | **0.50** ± 0.02 | 0.36 ± 0.03 | **0.26** ± 0.03 | 0.18 ± 0.02 | 0.19 ± 0.03 |
| | Spline | 0.79 ± 0.02 | 0.72 ± 0.02 | 0.48 ± 0.02 | 0.32 ± 0.03 | 0.24 ± 0.03 | 0.15 ± 0.02 |
| | RFF | 0.81 ± 0.02 | 0.60 ± 0.02 | 0.44 ± 0.03 | 0.30 ± 0.03 | 0.29 ± 0.03 | 0.23 ± 0.03 |
| | Laplacian | 0.83 ± 0.02 | 0.70 ± 0.02 | 0.53 ± 0.03 | 0.32 ± 0.03 | 0.22 ± 0.03 | 0.12 ± 0.03 |
| | Two Stage | 0.80 ± 0.02 | 0.64 ± 0.02 | 0.52 ± 0.02 | 0.32 ± 0.03 | 0.22 ± 0.03 | 0.08 ± 0.03 |
| **Temporal Causal3DIdent Dataset** | | | | | | | |
| CITRIS-VAE | NN | 0.72 ± 0.02 | **0.37** ± 0.03 | **0.11** ± 0.02 | **0.00** ± 0.00 | **0.00** ± 0.00 | **0.00** ± 0.00 |
| | Pearson | 0.78 ± 0.03 | 0.64 ± 0.03 | 0.39 ± 0.02 | 0.30 ± 0.02 | 0.14 ± 0.00 | 0.01 ± 0.01 |
| | Spearman | 0.73 ± 0.03 | 0.63 ± 0.03 | 0.56 ± 0.04 | 0.25 ± 0.04 | 0.09 ± 0.03 | **0.00** ± 0.00 |
| | Linear | **0.56** ± 0.03 | 0.41 ± 0.03 | 0.27 ± 0.02 | 0.01 ± 0.00 | **0.00** ± 0.00 | **0.00** ± 0.00 |
| | Spline | 0.69 ± 0.02 | 0.59 ± 0.03 | 0.33 ± 0.03 | **0.00** ± 0.00 | **0.00** ± 0.00 | **0.00** ± 0.00 |
| | RFF | 0.79 ± 0.02 | 0.63 ± 0.02 | 0.42 ± 0.02 | 0.10 ± 0.02 | 0.01 ± 0.01 | **0.00** ± 0.00 |
| | Laplacian | 0.71 ± 0.02 | 0.56 ± 0.02 | 0.35 ± 0.02 | 0.11 ± 0.01 | **0.00** ± 0.00 | **0.00** ± 0.00 |
| | Two Stage | 0.79 ± 0.02 | 0.66 ± 0.03 | 0.59 ± 0.04 | 0.25 ± 0.05 | 0.09 ± 0.03 | **0.00** ± 0.00 |
| iVAE | NN | 0.73 ± 0.03 | 0.46 ± 0.03 | 0.29 ± 0.02 | **0.00** ± 0.00 | **0.00** ± 0.00 | **0.00** ± 0.00 |
| | Pearson | **0.63** ± 0.04 | 0.47 ± 0.03 | 0.26 ± 0.03 | 0.24 ± 0.03 | **0.00** ± 0.00 | **0.00** ± 0.00 |
| | Spearman | 0.71 ± 0.03 | 0.50 ± 0.03 | 0.39 ± 0.02 | 0.11 ± 0.02 | **0.00** ± 0.00 | **0.00** ± 0.00 |
| | Linear | 0.70 ± 0.03 | **0.45** ± 0.03 | **0.24** ± 0.03 | **0.00** ± 0.00 | **0.00** ± 0.00 | **0.00** ± 0.00 |
| | Spline | 0.69 ± 0.03 | 0.62 ± 0.03 | 0.41 ± 0.03 | 0.02 ± 0.01 | **0.00** ± 0.00 | **0.00** ± 0.00 |
| | RFF | 0.73 ± 0.03 | 0.54 ± 0.03 | 0.39 ± 0.03 | 0.02 ± 0.01 | **0.00** ± 0.00 | **0.00** ± 0.00 |
| | Laplacian | 0.69 ± 0.03 | 0.49 ± 0.03 | 0.32 ± 0.03 | 0.01 ± 0.01 | **0.00** ± 0.00 | **0.00** ± 0.00 |
| | Two Stage | 0.75 ± 0.03 | 0.55 ± 0.03 | 0.36 ± 0.02 | 0.08 ± 0.02 | **0.00** ± 0.00 | **0.00** ± 0.00 |

Table 9: $R^2$ scores on the diagonal for the encodings learned in the Action/Temporal sparsity datasets and the Temporal Causal3DIdent datasets. The $(n)$ indicates the number of train and test points used in each column. We report the mean and standard deviation over 50 random seeds and in each column we indicate the best method by **bold**. If a score was below $-100$, we indicate this with †.

| Model | Method | \multicolumn{6}{c}{$R^2$-score on the diagonal ↑ $(n)$} |
|---|---|---|---|---|---|---|---|
| | | 5 | 10 | 20 | 100 | 1000 | 10000 |
| \multicolumn{8}{c}{**Action Sparsity Dataset**} |
| DMS-VAE | NN | **0.21** ±0.03 | 0.56 ±0.01 | 0.68 ±0.00 | **0.78** ±0.00 | 0.79 ±0.00 | **0.79** ±0.00 |
| | Pearson | - | - | - | - | - | - |
| | Spearman | - | - | - | - | - | - |
| | Linear | -0.29 ±0.13 | **0.56** ±0.02 | **0.72** ±0.00 | 0.77 ±0.00 | 0.77 ±0.00 | 0.77 ±0.00 |
| | Spline | † | 0.08 ±0.02 | 0.53 ±0.01 | 0.78 ±0.00 | 0.78 ±0.00 | 0.79 ±0.00 |
| | RFF | -0.69 ±0.12 | 0.24 ±0.02 | 0.59 ±0.01 | 0.77 ±0.00 | 0.78 ±0.00 | 0.78 ±0.00 |
| | Laplacian | -0.72 ±0.08 | -0.03 ±0.02 | 0.33 ±0.01 | 0.64 ±0.01 | 0.70 ±0.00 | 0.70 ±0.00 |
| | Two Stage | † | -1.25 ±1.41 | 0.53 ±0.08 | 0.77 ±0.00 | **0.79** ±0.00 | 0.79 ±0.00 |
| iVAE | NN | **-0.36** ±0.03 | -0.14 ±0.03 | 0.10 ±0.01 | **0.29** ±0.00 | **0.31** ±0.00 | **0.31** ±0.00 |
| | Pearson | - | - | - | - | - | - |
| | Spearman | - | - | - | - | - | - |
| | Linear | -0.86 ±0.11 | **-0.10** ±0.02 | **0.14** ±0.01 | 0.25 ±0.00 | 0.27 ±0.00 | 0.28 ±0.00 |
| | Spline | -19.24 ±13.32 | -0.18 ±0.02 | 0.05 ±0.01 | 0.24 ±0.00 | 0.27 ±0.00 | 0.27 ±0.00 |
| | RFF | -0.93 ±0.10 | -0.22 ±0.02 | -0.02 ±0.01 | 0.19 ±0.00 | 0.25 ±0.00 | 0.25 ±0.00 |
| | Laplacian | -0.83 ±0.06 | -0.23 ±0.02 | -0.05 ±0.01 | 0.12 ±0.00 | 0.17 ±0.00 | 0.17 ±0.00 |
| | Two Stage | † | -1.49 ±0.19 | -0.28 ±0.03 | 0.19 ±0.01 | 0.30 ±0.00 | 0.31 ±0.00 |
| \multicolumn{8}{c}{**Temporal Sparsity Dataset**} |
| DMS-VAE | NN | -2.78 ±0.30 | -0.26 ±0.03 | 0.17 ±0.01 | 0.38 ±0.00 | 0.42 ±0.00 | 0.43 ±0.00 |
| | Pearson | - | - | - | - | - | - |
| | Spearman | - | - | - | - | - | - |
| | Linear | **-0.94** ±0.11 | **0.11** ±0.02 | **0.32** ±0.01 | **0.41** ±0.00 | 0.43 ±0.00 | 0.43 ±0.00 |
| | Spline | -6.88 ±4.87 | -0.16 ±0.03 | 0.17 ±0.01 | 0.40 ±0.00 | 0.42 ±0.00 | 0.43 ±0.00 |
| | RFF | -1.00 ±0.11 | -0.10 ±0.03 | 0.21 ±0.01 | 0.39 ±0.00 | 0.41 ±0.00 | 0.42 ±0.00 |
| | Laplacian | -1.11 ±0.12 | -0.22 ±0.03 | 0.04 ±0.01 | 0.24 ±0.01 | 0.30 ±0.00 | 0.31 ±0.00 |
| | Two Stage | † | -1.24 ±0.19 | -0.11 ±0.05 | 0.32 ±0.01 | **0.43** ±0.00 | 0.43 ±0.00 |
| TCVAE | NN | -3.02 ±0.24 | -0.44 ±0.04 | 0.05 ±0.01 | 0.28 ±0.00 | 0.33 ±0.00 | 0.34 ±0.00 |
| | Pearson | - | - | - | - | - | - |
| | Spearman | - | - | - | - | - | - |
| | Linear | -1.00 ±0.15 | **0.00** ±0.03 | **0.23** ±0.01 | **0.32** ±0.00 | **0.33** ±0.00 | 0.33 ±0.00 |
| | Spline | † | -0.18 ±0.02 | 0.10 ±0.01 | 0.30 ±0.00 | 0.33 ±0.00 | 0.34 ±0.00 |
| | RFF | **-0.84** ±0.08 | -0.13 ±0.03 | 0.15 ±0.01 | 0.31 ±0.00 | 0.33 ±0.00 | 0.33 ±0.00 |
| | Laplacian | -0.92 ±0.10 | -0.21 ±0.03 | -0.01 ±0.01 | 0.17 ±0.01 | 0.23 ±0.00 | 0.23 ±0.00 |
| | Two Stage | † | -1.54 ±0.21 | -0.25 ±0.07 | 0.21 ±0.01 | 0.33 ±0.00 | **0.34** ±0.00 |
| \multicolumn{8}{c}{**Temporal Causal3DIdent Dataset**} |
| CITRIS-VAE | NN | -4.11 ±0.32 | -0.17 ±0.04 | **0.22** ±0.02 | 0.42 ±0.01 | 0.59 ±0.00 | **0.65** ±0.00 |
| | Pearson | - | - | - | - | - | - |
| | Spearman | - | - | - | - | - | - |
| | Linear | -0.82 ±0.15 | **-0.17** ±0.04 | 0.20 ±0.01 | **0.44** ±0.00 | 0.49 ±0.00 | 0.50 ±0.00 |
| | Spline | † | -0.21 ±0.03 | 0.12 ±0.01 | 0.44 ±0.01 | **0.60** ±0.00 | 0.62 ±0.00 |
| | RFF | -0.86 ±0.15 | -0.26 ±0.03 | -0.02 ±0.01 | 0.27 ±0.01 | 0.46 ±0.01 | 0.50 ±0.01 |
| | Laplacian | **-0.80** ±0.14 | -0.24 ±0.03 | -0.02 ±0.01 | 0.36 ±0.00 | 0.54 ±0.00 | 0.57 ±0.00 |
| | Two Stage | † | -3.59 ±1.13 | -0.81 ±0.12 | -0.60 ±0.05 | 0.52 ±0.01 | 0.62 ±0.00 |
| iVAE | NN | -4.29 ±0.36 | -0.23 ±0.05 | 0.13 ±0.02 | **0.62** ±0.00 | **0.71** ±0.00 | **0.74** ±0.00 |
| | Pearson | - | - | - | - | - | - |
| | Spearman | - | - | - | - | - | - |
| | Linear | -0.88 ±0.16 | **-0.20** ±0.03 | **0.14** ±0.01 | 0.43 ±0.00 | 0.47 ±0.00 | 0.48 ±0.00 |
| | Spline | † | -0.41 ±0.15 | 0.01 ±0.01 | 0.43 ±0.00 | 0.55 ±0.00 | 0.56 ±0.00 |
| | RFF | -0.84 ±0.14 | -0.26 ±0.03 | -0.01 ±0.01 | 0.36 ±0.01 | 0.52 ±0.00 | 0.54 ±0.00 |
| | Laplacian | **-0.81** ±0.14 | -0.24 ±0.03 | 0.00 ±0.01 | 0.34 ±0.01 | 0.49 ±0.00 | 0.51 ±0.00 |
| | Two Stage | † | † | -0.74 ±0.11 | -0.50 ±0.06 | 0.54 ±0.00 | 0.56 ±0.00 |

Table 10: Permutation errors when using binary concepts created based on the ground truth latent variables in the Action/Temporal sparsity datasets and the Temporal Causal3DIdent datasets. The $(n)$ indicates the number of train and test points used in each column. We report the mean and standard deviation over 10 random seeds in each column and we indicate the best method by **bold**.

| Model | Method | Permutation Errors ↓ $(n)$ | | | |
| | | 20 | 100 | 1000 | 10000 |
|---|---|---|---|---|---|
| **Action Sparsity Dataset** | | | | | |
| DMS-VAE | Linear | **0.08** ± **0.03** | **0.00** ± **0.00** | **0.00** ± **0.00** | **0.00** ± **0.00** |
| | Spline | 0.08 ± 0.03 | **0.00** ± **0.00** | **0.00** ± **0.00** | **0.00** ± **0.00** |
| | RFF | 0.08 ± 0.03 | **0.00** ± **0.00** | **0.00** ± **0.00** | **0.00** ± **0.00** |
| | Laplacian | 0.08 ± 0.03 | **0.00** ± **0.00** | **0.00** ± **0.00** | **0.00** ± **0.00** |
| | Two Stage | 0.08 ± 0.03 | 0.04 ± 0.03 | **0.00** ± **0.00** | **0.00** ± **0.00** |
| iVAE | Linear | 0.65 ± 0.07 | 0.38 ± 0.06 | 0.10 ± 0.05 | **0.00** ± **0.00** |
| | Spline | 0.73 ± 0.04 | 0.41 ± 0.05 | 0.14 ± 0.05 | **0.00** ± **0.00** |
| | RFF | 0.75 ± 0.03 | 0.45 ± 0.04 | 0.27 ± 0.04 | 0.09 ± 0.05 |
| | Laplacian | 0.73 ± 0.02 | 0.52 ± 0.04 | 0.25 ± 0.05 | 0.06 ± 0.04 |
| | Two Stage | 0.65 ± 0.06 | 0.65 ± 0.04 | 0.31 ± 0.04 | **0.00** ± **0.00** |
| **Temporal Sparsity Dataset** | | | | | |
| DMS-VAE | Linear | **0.37** ± **0.04** | **0.13** ± **0.05** | **0.00** ± **0.00** | **0.00** ± **0.00** |
| | Spline | 0.58 ± 0.04 | 0.16 ± 0.07 | **0.00** ± **0.00** | **0.00** ± **0.00** |
| | RFF | 0.56 ± 0.05 | 0.20 ± 0.06 | 0.02 ± 0.02 | **0.00** ± **0.00** |
| | Laplacian | 0.60 ± 0.05 | 0.30 ± 0.07 | 0.02 ± 0.02 | **0.00** ± **0.00** |
| | Two Stage | 0.46 ± 0.04 | 0.47 ± 0.06 | 0.06 ± 0.03 | **0.00** ± **0.00** |
| TCVAE | Linear | 0.54 ± 0.05 | 0.38 ± 0.06 | 0.32 ± 0.04 | 0.20 ± 0.00 |
| | Spline | 0.66 ± 0.05 | 0.36 ± 0.07 | 0.21 ± 0.04 | 0.18 ± 0.05 |
| | RFF | 0.63 ± 0.04 | 0.36 ± 0.04 | 0.21 ± 0.07 | 0.12 ± 0.08 |
| | Laplacian | 0.69 ± 0.05 | 0.43 ± 0.06 | 0.18 ± 0.07 | 0.26 ± 0.08 |
| | Two Stage | 0.63 ± 0.03 | 0.51 ± 0.03 | 0.31 ± 0.06 | 0.36 ± 0.08 |
| **Temporal Causal3DIdent Dataset** | | | | | |
| CITRISVAE | Linear | 0.10 ± 0.05 | 0.06 ± 0.04 | **0.00** ± **0.00** | **0.00** ± **0.00** |
| | Spline | 0.07 ± 0.03 | **0.00** ± **0.00** | **0.00** ± **0.00** | **0.00** ± **0.00** |
| | RFF | 0.33 ± 0.06 | 0.10 ± 0.04 | **0.00** ± **0.00** | **0.00** ± **0.00** |
| | Laplacian | 0.17 ± 0.04 | **0.00** ± **0.00** | **0.00** ± **0.00** | **0.00** ± **0.00** |
| | Two Stage | 0.27 ± 0.05 | 0.34 ± 0.05 | 0.09 ± 0.02 | 0.10 ± 0.02 |
| iVAE | Linear | **0.06** ± **0.04** | **0.00** ± **0.00** | **0.00** ± **0.00** | **0.00** ± **0.00** |
| | Spline | 0.07 ± 0.05 | 0.03 ± 0.03 | **0.00** ± **0.00** | **0.00** ± **0.00** |
| | RFF | 0.17 ± 0.06 | 0.03 ± 0.03 | **0.00** ± **0.00** | **0.00** ± **0.00** |
| | Laplacian | 0.10 ± 0.05 | 0.03 ± 0.03 | **0.00** ± **0.00** | **0.00** ± **0.00** |
| | Two Stage | 0.11 ± 0.04 | 0.14 ± 0.05 | **0.00** ± **0.00** | **0.00** ± **0.00** |

Table 11: Concept Accuracy and Label accuracy when using binary concepts and binary downstream labels created based on the ground truth latent variables in the Action/Temporal sparsity datasets and the Temporal Causal3DIdent datasets. The $(n)$ indicates the number of train and test points used in each column. We report the mean and standard deviation over 10 random seeds in each column and we indicate the best method by **bold**.

| Model | Method | Concept Acc ↑ $(n)$ | | | | Label Acc ↑ $(n)$ | | | |
|---|---|---|---|---|---|---|---|---|---|
| | | 20 | 100 | 1000 | 10000 | 20 | 100 | 1000 | 10000 |
| **Action Sparsity Dataset** | | | | | | | | | |
| DMS-VAE | Linear | **0.83** ± **0.01** | 0.86 ± 0.00 | 0.85 ± 0.00 | 0.86 ± 0.00 | **0.77** ± **0.03** | 0.81 ± 0.01 | 0.83 ± 0.01 | 0.84 ± 0.01 |
| | Spline | 0.83 ± 0.01 | **0.86** ± **0.00** | 0.85 ± 0.00 | 0.86 ± 0.00 | 0.72 ± 0.03 | **0.83** ± **0.02** | **0.85** ± **0.01** | 0.86 ± 0.01 |
| | RFF | 0.82 ± 0.00 | 0.85 ± 0.00 | 0.85 ± 0.00 | 0.86 ± 0.00 | 0.77 ± 0.02 | 0.82 ± 0.02 | 0.84 ± 0.01 | 0.85 ± 0.01 |
| | Laplacian | 0.81 ± 0.01 | 0.85 ± 0.00 | 0.85 ± 0.00 | 0.86 ± 0.00 | 0.75 ± 0.03 | 0.82 ± 0.02 | 0.84 ± 0.01 | 0.85 ± 0.01 |
| | Two Stage | 0.83 ± 0.01 | 0.85 ± 0.00 | **0.86** ± **0.00** | 0.86 ± 0.00 | 0.72 ± 0.03 | 0.80 ± 0.02 | 0.80 ± 0.01 | 0.82 ± 0.02 |
| iVAE | Linear | 0.66 ± 0.01 | 0.68 ± 0.00 | 0.70 ± 0.00 | 0.70 ± 0.00 | 0.73 ± 0.03 | 0.79 ± 0.02 | 0.81 ± 0.01 | 0.83 ± 0.01 |
| | Spline | 0.63 ± 0.01 | 0.68 ± 0.00 | 0.70 ± 0.00 | 0.70 ± 0.00 | 0.69 ± 0.02 | 0.76 ± 0.02 | 0.81 ± 0.01 | 0.83 ± 0.01 |
| | RFF | 0.59 ± 0.01 | 0.66 ± 0.01 | 0.68 ± 0.00 | 0.69 ± 0.00 | 0.59 ± 0.04 | 0.70 ± 0.02 | 0.78 ± 0.01 | 0.81 ± 0.01 |
| | Laplacian | 0.59 ± 0.01 | 0.65 ± 0.01 | 0.68 ± 0.00 | 0.68 ± 0.00 | 0.60 ± 0.04 | 0.70 ± 0.02 | 0.78 ± 0.01 | 0.79 ± 0.01 |
| | Two Stage | 0.64 ± 0.01 | 0.65 ± 0.00 | 0.69 ± 0.00 | 0.70 ± 0.00 | 0.67 ± 0.03 | 0.70 ± 0.02 | 0.79 ± 0.01 | 0.82 ± 0.01 |
| CBM [28] | | 0.64 ± 0.01 | 0.73 ± 0.01 | 0.85 ± 0.00 | 0.91 ± 0.00 | 0.60 ± 0.03 | 0.60 ± 0.02 | 0.73 ± 0.01 | 0.88 ± 0.01 |
| CEM [57] | | 0.64 ± 0.02 | 0.72 ± 0.00 | 0.86 ± 0.00 | 0.91 ± 0.00 | 0.66 ± 0.02 | 0.69 ± 0.03 | 0.82 ± 0.01 | 0.88 ± 0.01 |
| HardCBM [17] | | 0.66 ± 0.01 | 0.73 ± 0.01 | 0.85 ± 0.00 | **0.91** ± **0.00** | 0.56 ± 0.03 | 0.61 ± 0.01 | 0.68 ± 0.01 | **0.89** ± **0.00** |
| **Temporal Sparsity Dataset** | | | | | | | | | |
| DMS-VAE | Linear | **0.69** ± **0.01** | 0.73 ± 0.00 | 0.74 ± 0.00 | 0.74 ± 0.00 | **0.75** ± **0.03** | **0.79** ± **0.01** | **0.84** ± **0.02** | 0.85 ± 0.02 |
| | Spline | 0.64 ± 0.01 | 0.73 ± 0.01 | 0.74 ± 0.00 | 0.74 ± 0.00 | 0.69 ± 0.03 | 0.79 ± 0.02 | 0.84 ± 0.01 | 0.85 ± 0.01 |
| | RFF | 0.66 ± 0.01 | 0.72 ± 0.00 | 0.74 ± 0.00 | 0.74 ± 0.00 | 0.74 ± 0.02 | 0.79 ± 0.01 | 0.82 ± 0.01 | 0.84 ± 0.01 |
| | Laplacian | 0.64 ± 0.01 | 0.71 ± 0.00 | 0.73 ± 0.00 | 0.73 ± 0.00 | 0.69 ± 0.02 | 0.75 ± 0.02 | 0.80 ± 0.01 | 0.83 ± 0.01 |
| | Two Stage | 0.67 ± 0.01 | 0.69 ± 0.00 | 0.74 ± 0.00 | 0.74 ± 0.00 | 0.66 ± 0.03 | 0.71 ± 0.02 | 0.80 ± 0.02 | 0.84 ± 0.02 |
| TCVAE | Linear | 0.67 ± 0.01 | 0.70 ± 0.01 | 0.71 ± 0.00 | 0.71 ± 0.00 | 0.72 ± 0.03 | 0.78 ± 0.02 | 0.84 ± 0.01 | 0.84 ± 0.01 |
| | Spline | 0.63 ± 0.01 | 0.70 ± 0.01 | 0.71 ± 0.00 | 0.71 ± 0.00 | 0.67 ± 0.03 | 0.77 ± 0.01 | 0.83 ± 0.02 | 0.84 ± 0.01 |
| | RFF | 0.63 ± 0.01 | 0.69 ± 0.01 | 0.71 ± 0.00 | 0.71 ± 0.00 | 0.68 ± 0.03 | 0.79 ± 0.02 | 0.82 ± 0.01 | 0.84 ± 0.01 |
| | Laplacian | 0.61 ± 0.01 | 0.69 ± 0.01 | 0.70 ± 0.00 | 0.70 ± 0.00 | 0.65 ± 0.03 | 0.74 ± 0.01 | 0.80 ± 0.01 | 0.82 ± 0.01 |
| | Two Stage | 0.63 ± 0.01 | 0.67 ± 0.01 | 0.71 ± 0.00 | 0.71 ± 0.00 | 0.71 ± 0.03 | 0.72 ± 0.02 | 0.80 ± 0.02 | 0.84 ± 0.01 |
| CBM [28] | | 0.64 ± 0.01 | 0.74 ± 0.01 | 0.86 ± 0.00 | 0.92 ± 0.00 | 0.53 ± 0.03 | 0.60 ± 0.02 | 0.75 ± 0.01 | 0.89 ± 0.01 |
| CEM [57] | | 0.66 ± 0.01 | **0.75** ± **0.01** | **0.87** ± **0.00** | 0.92 ± 0.00 | 0.62 ± 0.04 | 0.68 ± 0.02 | 0.82 ± 0.01 | 0.89 ± 0.01 |
| HardCBM [17] | | 0.64 ± 0.01 | 0.72 ± 0.01 | 0.86 ± 0.00 | **0.92** ± **0.00** | 0.50 ± 0.04 | 0.58 ± 0.02 | 0.71 ± 0.02 | **0.90** ± **0.01** |
| **Temporal Causal3DIdent Dataset** | | | | | | | | | |
| CITRISVAE | Linear | **0.81** ± **0.01** | 0.85 ± 0.00 | 0.85 ± 0.00 | 0.85 ± 0.00 | 0.74 ± 0.06 | 0.75 ± 0.06 | 0.80 ± 0.04 | 0.81 ± 0.04 |
| | Spline | 0.80 ± 0.01 | **0.87** ± **0.00** | **0.88** ± **0.00** | **0.89** ± **0.00** | 0.74 ± 0.06 | **0.80** ± **0.04** | **0.82** ± **0.03** | **0.83** ± **0.03** |
| | RFF | 0.71 ± 0.01 | 0.81 ± 0.01 | 0.84 ± 0.01 | 0.85 ± 0.01 | 0.70 ± 0.06 | 0.76 ± 0.05 | 0.79 ± 0.04 | 0.81 ± 0.04 |
| | Laplacian | 0.76 ± 0.01 | 0.84 ± 0.01 | 0.87 ± 0.00 | 0.88 ± 0.00 | 0.72 ± 0.06 | 0.78 ± 0.04 | 0.80 ± 0.03 | 0.82 ± 0.03 |
| | Two Stage | 0.76 ± 0.01 | 0.77 ± 0.02 | 0.86 ± 0.01 | 0.86 ± 0.00 | 0.72 ± 0.05 | 0.72 ± 0.06 | 0.77 ± 0.04 | 0.79 ± 0.04 |
| iVAE | Linear | 0.81 ± 0.01 | 0.84 ± 0.00 | 0.85 ± 0.00 | 0.85 ± 0.00 | 0.74 ± 0.06 | 0.76 ± 0.05 | 0.78 ± 0.05 | 0.80 ± 0.04 |
| | Spline | 0.77 ± 0.01 | 0.84 ± 0.00 | 0.86 ± 0.00 | 0.87 ± 0.00 | 0.73 ± 0.06 | 0.78 ± 0.04 | 0.79 ± 0.04 | 0.81 ± 0.04 |
| | RFF | 0.75 ± 0.01 | 0.81 ± 0.01 | 0.85 ± 0.00 | 0.86 ± 0.00 | 0.69 ± 0.06 | 0.75 ± 0.05 | 0.78 ± 0.04 | 0.80 ± 0.04 |
| | Laplacian | 0.74 ± 0.01 | 0.81 ± 0.00 | 0.85 ± 0.00 | 0.86 ± 0.00 | 0.69 ± 0.07 | 0.76 ± 0.05 | 0.78 ± 0.04 | 0.80 ± 0.04 |
| | Two Stage | 0.78 ± 0.01 | 0.82 ± 0.01 | 0.86 ± 0.00 | 0.87 ± 0.00 | **0.77** ± **0.06** | 0.75 ± 0.05 | 0.75 ± 0.05 | 0.78 ± 0.05 |
| CBM [28] | | 0.58 ± 0.01 | 0.65 ± 0.01 | 0.73 ± 0.00 | 0.82 ± 0.00 | 0.57 ± 0.02 | 0.53 ± 0.03 | 0.68 ± 0.04 | 0.78 ± 0.05 |
| CEM [57] | | 0.61 ± 0.01 | 0.64 ± 0.01 | 0.73 ± 0.00 | 0.82 ± 0.00 | 0.64 ± 0.04 | 0.67 ± 0.03 | 0.72 ± 0.05 | 0.78 ± 0.05 |
| HardCBM [17] | | 0.60 ± 0.01 | 0.65 ± 0.01 | 0.74 ± 0.00 | 0.83 ± 0.00 | 0.57 ± 0.05 | 0.53 ± 0.02 | 0.63 ± 0.03 | 0.77 ± 0.05 |

Table 12: NIS and OIS scores when using binary concepts created based on the ground truth latent variables in the Action/Temporal sparsity datasets and the Temporal Causal3DIdent datasets. The $(n)$ indicates the number of train and test points used in each column. We report the mean and standard deviation over 10 random seeds in each column and we indicate the best method by **bold**.

| Model | Method | OIS-score ↓ $(n)$ | | | | NIS-score ↓ $(n)$ | | | |
|---|---|---|---|---|---|---|---|---|---|
| | | 20 | 100 | 1000 | 10000 | 20 | 100 | 1000 | 10000 |
| **Action Sparsity Dataset** | | | | | | | | | |
| DMS-VAE | Linear | **0.63 ± 0.01** | 0.40 ± 0.00 | 0.15 ± 0.00 | 0.12 ± 0.00 | 0.50 ± 0.00 | **0.50 ± 0.00** | **0.50 ± 0.00** | **0.50 ± 0.00** |
| | Spline | 0.63 ± 0.01 | 0.40 ± 0.00 | 0.15 ± 0.00 | 0.12 ± 0.00 | 0.46 ± 0.02 | 0.50 ± 0.00 | 0.50 ± 0.00 | 0.50 ± 0.00 |
| | RFF | 0.64 ± 0.01 | **0.39 ± 0.00** | 0.14 ± 0.00 | 0.12 ± 0.00 | 0.47 ± 0.03 | 0.50 ± 0.00 | 0.50 ± 0.00 | 0.50 ± 0.00 |
| | Laplacian | 0.64 ± 0.01 | 0.40 ± 0.00 | 0.14 ± 0.00 | 0.11 ± 0.00 | 0.47 ± 0.02 | 0.50 ± 0.00 | 0.50 ± 0.00 | 0.50 ± 0.00 |
| | Two Stage | 0.84 ± 0.02 | 0.42 ± 0.01 | 0.15 ± 0.00 | 0.13 ± 0.00 | 0.47 ± 0.03 | 0.57 ± 0.02 | 0.64 ± 0.00 | 0.65 ± 0.00 |
| iVAE | Linear | 0.63 ± 0.01 | 0.40 ± 0.00 | 0.29 ± 0.00 | 0.26 ± 0.00 | 0.50 ± 0.00 | 0.50 ± 0.00 | 0.50 ± 0.00 | 0.50 ± 0.00 |
| | Spline | 0.63 ± 0.01 | 0.40 ± 0.00 | 0.28 ± 0.00 | 0.26 ± 0.00 | 0.50 ± 0.00 | 0.50 ± 0.00 | 0.50 ± 0.00 | 0.50 ± 0.00 |
| | RFF | 0.64 ± 0.01 | 0.39 ± 0.00 | 0.28 ± 0.00 | 0.26 ± 0.00 | **0.45 ± 0.02** | 0.50 ± 0.00 | 0.50 ± 0.00 | 0.50 ± 0.00 |
| | Laplacian | 0.64 ± 0.01 | 0.40 ± 0.00 | 0.27 ± 0.00 | 0.25 ± 0.00 | 0.47 ± 0.02 | 0.50 ± 0.00 | 0.50 ± 0.00 | 0.50 ± 0.00 |
| | Two Stage | 0.92 ± 0.02 | 0.47 ± 0.01 | 0.29 ± 0.00 | 0.27 ± 0.00 | 0.49 ± 0.02 | 0.63 ± 0.01 | 0.78 ± 0.01 | 0.81 ± 0.00 |
| CBM [28] | | 0.92 ± 0.02 | 0.43 ± 0.01 | **0.11 ± 0.00** | 0.07 ± 0.00 | 0.56 ± 0.03 | 0.63 ± 0.01 | 0.63 ± 0.00 | 0.59 ± 0.00 |
| CEM [57] | | 0.90 ± 0.03 | 0.47 ± 0.01 | 0.40 ± 0.01 | 0.57 ± 0.01 | 0.59 ± 0.02 | 0.64 ± 0.01 | 0.71 ± 0.01 | 0.81 ± 0.01 |
| HardCBM [17] | | 0.92 ± 0.02 | 0.46 ± 0.01 | 0.12 ± 0.00 | **0.06 ± 0.00** | 0.57 ± 0.02 | 0.63 ± 0.01 | 0.64 ± 0.00 | 0.59 ± 0.00 |
| **Temporal Sparsity Dataset** | | | | | | | | | |
| DMS-VAE | Linear | **0.62 ± 0.01** | 0.39 ± 0.00 | 0.26 ± 0.00 | 0.23 ± 0.00 | 0.50 ± 0.00 | 0.50 ± 0.00 | **0.50 ± 0.00** | **0.50 ± 0.00** |
| | Spline | 0.62 ± 0.01 | 0.39 ± 0.00 | 0.26 ± 0.00 | 0.23 ± 0.00 | **0.47 ± 0.03** | 0.50 ± 0.00 | 0.50 ± 0.00 | 0.50 ± 0.00 |
| | RFF | 0.63 ± 0.01 | **0.38 ± 0.00** | 0.25 ± 0.00 | 0.23 ± 0.00 | 0.50 ± 0.00 | **0.47 ± 0.01** | 0.50 ± 0.00 | 0.50 ± 0.00 |
| | Laplacian | 0.64 ± 0.01 | 0.39 ± 0.00 | 0.26 ± 0.00 | 0.22 ± 0.00 | 0.49 ± 0.03 | 0.50 ± 0.00 | 0.50 ± 0.00 | 0.50 ± 0.00 |
| | Two Stage | 0.88 ± 0.02 | 0.45 ± 0.02 | 0.26 ± 0.00 | 0.23 ± 0.00 | 0.52 ± 0.03 | 0.62 ± 0.01 | 0.76 ± 0.01 | 0.80 ± 0.00 |
| TCVAE | Linear | 0.62 ± 0.01 | 0.39 ± 0.00 | 0.27 ± 0.00 | 0.25 ± 0.00 | 0.50 ± 0.00 | 0.50 ± 0.00 | 0.50 ± 0.00 | 0.50 ± 0.00 |
| | Spline | 0.62 ± 0.01 | 0.39 ± 0.00 | 0.27 ± 0.00 | 0.25 ± 0.00 | 0.50 ± 0.00 | 0.50 ± 0.00 | 0.50 ± 0.00 | 0.50 ± 0.00 |
| | RFF | 0.63 ± 0.01 | 0.38 ± 0.00 | 0.27 ± 0.00 | 0.25 ± 0.00 | 0.50 ± 0.00 | 0.50 ± 0.00 | 0.50 ± 0.00 | 0.50 ± 0.00 |
| | Laplacian | 0.64 ± 0.01 | 0.39 ± 0.00 | 0.27 ± 0.00 | 0.24 ± 0.00 | 0.50 ± 0.00 | 0.50 ± 0.00 | 0.50 ± 0.00 | 0.50 ± 0.00 |
| | Two Stage | 0.91 ± 0.02 | 0.47 ± 0.01 | 0.28 ± 0.00 | 0.24 ± 0.00 | 0.49 ± 0.02 | 0.63 ± 0.01 | 0.79 ± 0.01 | 0.82 ± 0.01 |
| CBM [28] | | 0.96 ± 0.01 | 0.44 ± 0.02 | **0.11 ± 0.00** | 0.05 ± 0.00 | 0.55 ± 0.02 | 0.63 ± 0.01 | 0.61 ± 0.00 | 0.57 ± 0.00 |
| CEM [57] | | 0.95 ± 0.02 | 0.46 ± 0.01 | 0.38 ± 0.01 | 0.54 ± 0.01 | 0.53 ± 0.02 | 0.65 ± 0.00 | 0.71 ± 0.01 | 0.81 ± 0.01 |
| HardCBM [17] | | 0.93 ± 0.01 | 0.44 ± 0.01 | 0.12 ± 0.00 | **0.05 ± 0.00** | 0.54 ± 0.03 | 0.62 ± 0.01 | 0.62 ± 0.00 | 0.57 ± 0.00 |
| **Temporal Causal3DIdent Dataset** | | | | | | | | | |
| CITRISVAE | Linear | 0.69 ± 0.02 | 0.44 ± 0.01 | 0.19 ± 0.00 | 0.16 ± 0.00 | 0.46 ± 0.02 | **0.50 ± 0.00** | **0.50 ± 0.00** | **0.50 ± 0.00** |
| | Spline | 0.69 ± 0.02 | 0.44 ± 0.01 | 0.13 ± 0.00 | **0.09 ± 0.00** | **0.41 ± 0.03** | 0.50 ± 0.00 | 0.50 ± 0.00 | 0.50 ± 0.00 |
| | RFF | 0.65 ± 0.01 | 0.43 ± 0.01 | 0.16 ± 0.00 | 0.13 ± 0.01 | 0.41 ± 0.03 | 0.50 ± 0.00 | 0.50 ± 0.00 | 0.50 ± 0.00 |
| | Laplacian | **0.64 ± 0.01** | **0.40 ± 0.02** | **0.13 ± 0.00** | 0.09 ± 0.00 | 0.44 ± 0.03 | 0.50 ± 0.00 | 0.50 ± 0.00 | 0.50 ± 0.00 |
| | Two Stage | 0.88 ± 0.03 | 0.46 ± 0.02 | 0.16 ± 0.01 | 0.14 ± 0.01 | 0.46 ± 0.03 | 0.52 ± 0.01 | 0.56 ± 0.01 | 0.59 ± 0.00 |
| iVAE | Linear | 0.69 ± 0.02 | 0.44 ± 0.01 | 0.17 ± 0.00 | 0.14 ± 0.00 | 0.47 ± 0.03 | 0.50 ± 0.00 | 0.50 ± 0.00 | 0.50 ± 0.00 |
| | Spline | 0.69 ± 0.02 | 0.43 ± 0.04 | 0.17 ± 0.00 | 0.13 ± 0.00 | 0.42 ± 0.03 | 0.50 ± 0.00 | 0.50 ± 0.00 | 0.50 ± 0.00 |
| | RFF | 0.65 ± 0.01 | 0.42 ± 0.02 | 0.16 ± 0.00 | 0.13 ± 0.00 | 0.47 ± 0.03 | 0.50 ± 0.00 | 0.50 ± 0.00 | 0.50 ± 0.00 |
| | Laplacian | 0.64 ± 0.01 | 0.43 ± 0.02 | 0.16 ± 0.00 | 0.13 ± 0.00 | 0.45 ± 0.02 | 0.50 ± 0.00 | 0.50 ± 0.00 | 0.50 ± 0.00 |
| | Two Stage | 0.89 ± 0.02 | 0.46 ± 0.03 | 0.17 ± 0.00 | 0.13 ± 0.00 | 0.46 ± 0.04 | 0.52 ± 0.01 | 0.55 ± 0.01 | 0.58 ± 0.01 |
| CBM [28] | | 1.00 ± 0.03 | 0.48 ± 0.02 | 0.25 ± 0.00 | 0.18 ± 0.00 | 0.50 ± 0.05 | 0.55 ± 0.01 | 0.58 ± 0.00 | 0.57 ± 0.00 |
| CEM [57] | | 0.97 ± 0.05 | 0.51 ± 0.02 | 0.36 ± 0.01 | 0.49 ± 0.01 | 0.50 ± 0.03 | 0.59 ± 0.01 | 0.62 ± 0.01 | 0.68 ± 0.01 |
| HardCBM [17] | | 0.96 ± 0.03 | 0.47 ± 0.02 | 0.24 ± 0.00 | 0.16 ± 0.00 | 0.51 ± 0.02 | 0.55 ± 0.01 | 0.58 ± 0.00 | 0.57 ± 0.00 |

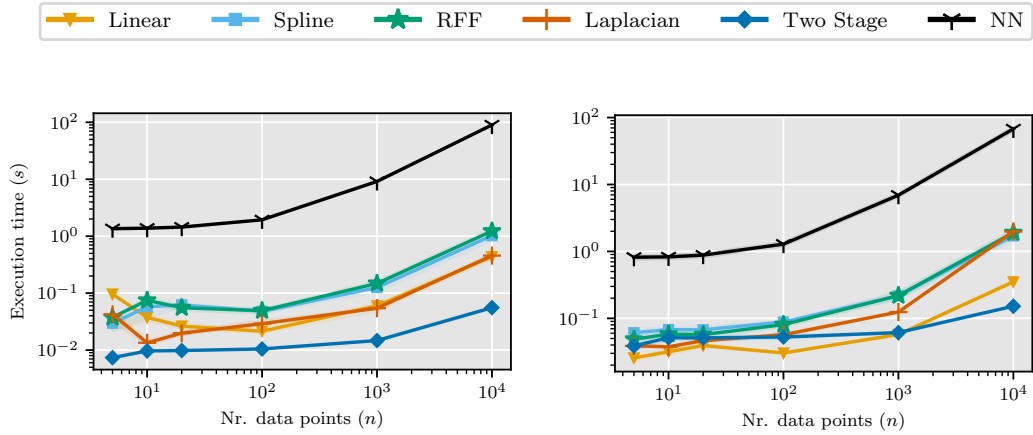

Figure 22: Execution times of the baseline and multiple versions of our estimator on the causal variables and encodings learned based on the Action/Temporal Sparsity Dataset and the Temporal Causal3DIdent dataset

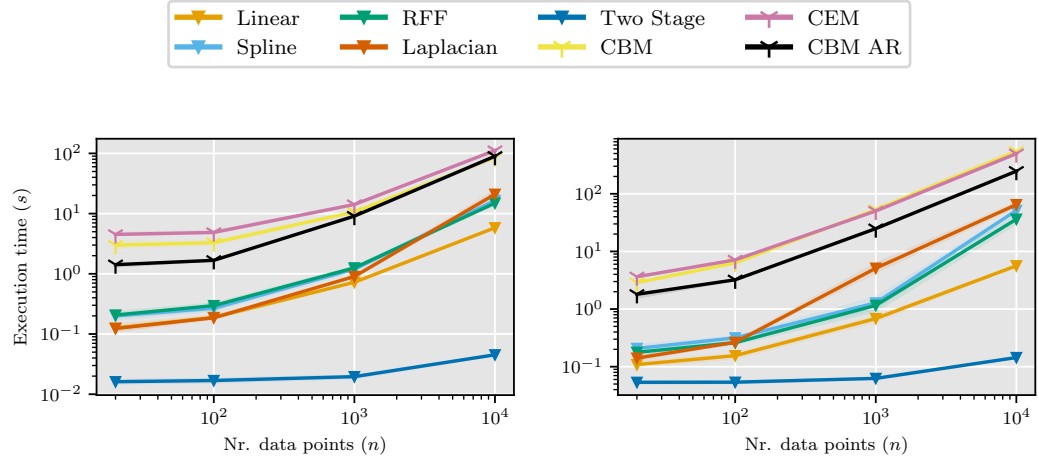

Figure 23: Execution times of the concept-based models and multiple versions of our estimator on the classification downstream tasks based on the Action/Temporal Sparsity Dataset and the Temporal Causal3DIdent dataset

