# OpenReview forum: "Sample-efficient Learning of Concepts with Theoretical Guarantees: from Data to Concepts without Interventions"
_NeurIPS.cc/2025/Conference — NeurIPS 2025 poster_

### Official Review · Reviewer_8oyR · 2025-06-29

**Clarity:** 2
**Significance:** 2
**Originality:** 3
**Rating:** 5
**Confidence:** 2

**Summary:**

This work utilizes causal representation learning methods to first learn causal variables related to a given task, and then maps those causal variables into interpretable concepts for use in a CBM. By first learning causal variables, the authors propose that with few concept labels they are able to train a CBM with less impurity and higher accuracy than prior work (CBM, CEM, HardCBM).

**Questions:**

- Could the authors explain Figure 20 in the appendix as well as the general computation cost in more detail? From what I understand, the proposed approach is a two-stage approach, yet is more computationally efficient than CBM which should naively learn concepts from annotated data. Similarly, if I utilize a more powerful backbone for CBM, can I outperform the proposed approach without needing to perform the additional complexity?
- What do the authors mean by 'few concept labels' throughout the paper? Are fewer concepts used? I don't see this claim substantiated in the main results, as it seems each method is tested over a different number of training points.
- Could the authors explain lines 295-297: "Although CITRIS-VAE provides the correct permutation of the groups, we ignore it and perform a random permutation on the variables. We train an MLP for each of the 32 dimensions that predicts all causal variables. Based on the R2 scores of these regressions, we learn the group assignments." From what I understand the authors are saying the CRL method perfectly learns the true factors of variation within the dataset, is this the case?

**Ethical Concerns:**

["NO or VERY MINOR ethics concerns only"]

**Final Justification:**

Thanks for addressing my questions! While I am not very familiar with the CRL literature, I believe this paper offers an interesting connection between concept-based models and CRL approaches and I am happy to raise my score, even if it is unclear whether the CRL methods which the approach is built upon will be able to scale beyond toy problems in the future.

**Limitations:**

Yes

**Quality:**

3

**Strengths And Weaknesses:**

Strengths
- Concept leakage is a very important problem in CBMs, and learning disentangled concept representations is a very interesting idea for ensuring concepts mean what they say.
- Results are strong on both synthetic datasets evaluated.

Weaknesses
- Both datasets are synthetically created and fairly toy problems. Do the causal representation learning methods just not scale? Why can the authors not evaluate on a classic dataset such as CUB-200? If the theoretical analysis is grounded in the fact that the CRL approach can learn to disentangle the factors of variation within the dataset, then how is learning the causal variables not enough already? Current CBMs offer a solution for when the learning task is too complicated for simpler methods to work effectively, but if this approach is limited to small-scale datasets then how could it be useful in practice?
- From what I understand, the approach has many hyperparameters compared to typical concept bottleneck models. Is performance resistant to the selection of these hyperparameters? How were the parameters in Table 3 of the Appendix selected?

---

> ### Author Rebuttal · Authors · 2025-07-30
>
> Thank you for your careful review, we will address each point in the following.
>
> > Both datasets are synthetically created and fairly toy problems. Do the causal representation learning methods just not scale?
>
> The datasets we considered are from the CRL literature. For CRL methods it is fairly typical that they consider 10-20 causal variables to identify, since indeed they have issues scaling.
>
> > Why can the authors not evaluate on a classic dataset such as CUB-200?
>
> We cannot evaluate on such a dataset, since the concepts are binary, and the very few discrete CRL methods are currently limited to very specific assumptions, e.g. the assumption that different variables always affect different parts of an observation [Kong et al. 2024], which generally don’t hold in this dataset.
>
> [Kong et al. 2024] Kong, Lingjing, et al. "Learning discrete concepts in latent hierarchical models." Advances in Neural Information Processing Systems 37 (2024): 36938-36975.
>
> > If the theoretical analysis is grounded in the fact that the CRL approach can learn to disentangle the factors of variation within the dataset, then how is learning the causal variables not enough already?
>
> CRL methods are unsupervised, and can therefore only identify the factors up to an equivalence class of permutation and element-wise transformation. So even if the CRL method learns to to disentangle the factors of variation,  it will remain unclear what each learned variable means semantically. To actually learn the concepts it is crucial to also learn the alignment map with concept labels (see second stage of Figure 1).
>
> > Current CBMs offer a solution for when the learning task is too complicated for simpler methods to work effectively, but if this approach is limited to small-scale datasets then how could it be useful in practice?
>
> The limitation is not the size of the data, but rather the number of latent dimensions/concepts because of the current CRL methods. Our main concerns are to address the two main problems with CMBs: a) reliability of the learned concepts, which should avoid spurious correlations and provide theoretical guarantees on the accuracy; and b) reducing the number of required concept labels by leveraging unsupervised CRL methods.
>
> > Number of hyperparameters in Table 3 and robustness to their selection:
> For each CRL method, we select the same hyperparameters as the original CRL papers, also since we are evaluating on their original benchmarks. In particular Table 3 reports all of the hyperparameters from the DMSVAE paper.
>
> > Could the authors explain Figure 20 in the appendix as well as the general computation cost in more detail? From what I understand, the proposed approach is a two-stage approach, yet is more computationally efficient than CBM which should naively learn concepts from annotated data.
>
> In Figure 20 we did not consider the CRL phase, since we assume that we get a pre-trained CRL encoder. If this phase was considered the total time would be 8+ hours $\approx$ 29k seconds, since we can use hundreds of thousands of unlabelled data for the unsupervised CRL methods to have a good disentangled representation. On the other hand, the second step of our approach is generally more computationally efficient than the CBM approaches, as we only need to perform a regularized regression, instead of training a whole network.
>
> > Similarly, if I utilize a more powerful backbone for CBM, can I outperform the proposed approach without needing to perform the additional complexity?
>
> A more powerful backbone, e.g. different network architecture, for CBM would not help, because it would not address spurious correlations and it cannot significantly reduce the number of required concept labels, which are used for the whole pipeline in standard CBMs.
>
> > What do the authors mean by 'few concept labels' throughout the paper? Are fewer concepts used? I don't see this claim substantiated in the main results, as it seems each method is tested over a different number of training points.
>
> Standard CBMs use fully supervised learning to learn the concepts $C_i$. In addition to labels for the desired prediction target $Y$, CBMs therefore require labels for all concepts to be provided with the training data. In contrast, we use unsupervised CRL to identify the concepts up to an equivalence class, and then we only need $n$ concept labels to learn the alignment map $\alpha$ (see Figure 1), which requires much fewer labels for the concepts $C_i$ to be provided with the training data. Table 1 shows how the methods perform for different values of the $n$ concept labels.
>
> > Could the authors explain lines 295-297: "Although CITRIS-VAE provides the correct permutation of the groups, we ignore it and perform a random permutation on the variables. We train an MLP for each of the 32 dimensions that predicts all causal variables. Based on the R2 scores of these regressions, we learn the group assignments." From what I understand the authors are saying the CRL method perfectly learns the true factors of variation within the dataset, is this the case?
>
> No, this is not the case. As other CRL methods, CITRIS-VAE is an unsupervised representation learning method and it cannot recover interpretable factors, since the learned representations in the latent space are not semantically meaningful (i.e., they are embeddings of the ground truth variables). So we still need to learn an alignment function between these representations and the concepts by using concept labels, and how to do this efficiently in the samples and with theoretical guarantees is part of our contribution.
>
> As opposed to other CRL methods, CITRIS-VAE also recovers the correct permutation, which simplifies learning this alignment, but at the cost of having knowledge of which variable is intervened upon at each timestep. Since this is in general not true for other CRL methods, in our evaluation we have ignored this permutation and compared CITRIS-VAE with the other methods on the same setting.

---

> > ### Comment · Reviewer_8oyR · 2025-08-02
> >
> > Thanks for addressing my questions! While I am not very familiar with the CRL literature, I believe this paper offers an interesting connection between concept-based models and CRL approaches and I am happy to raise my score, even if it is unclear whether the CRL methods which the approach is built upon will be able to scale beyond toy problems in the future.

---

### Official Review · Reviewer_G2in · 2025-07-01

**Clarity:** 2
**Significance:** 2
**Originality:** 2
**Rating:** 4
**Confidence:** 2

**Summary:**

This paper proposes using Lasso regression to map from a pretrained representation $M$, stemming from some exisitng VAE-based model that has been trained to discover disentangled causal factors, into a predefined set of concepts $C$, via a sparse mapping $\alpha$. Sparsity is very important in this task, since each $M_i$ only becomes semantically interpretable if it maps to a single $C_j$. This is specifically formalized using Group Lasso, in order to allow for multidimensional factors in $M$ to be active together (for instance, 3D position as a factor).
The results on several synthetic datasets, where the underlying factors are known by design, show the advantage of the proposed approach with respect to regular concept bottlenecks.

**Questions:**

My main request for the authors would be a clear delineation of where the claimed novelty lies, including a title that clearly states it (or at least an abstract).
In addition, I would also like to see the response of the authors to the raised weakness points.

**Ethical Concerns:**

["NO or VERY MINOR ethics concerns only"]

**Final Justification:**

The authors have clarified most of the doubts I had. Since several reviewers are positive with this paper, I don't want to stand on the way, since I'm not too familiar with several of its main elements.

**Limitations:**

Some of the main limitations are mentioned in the conclusions section, specifically related to the fact that an actual alignment between the real causal factors and human-understandable concepts may not exist. It is also mentioned that the method is not meant to work with correlated factors, although it is claimed it works better than expected in this setting. The one limitation that is not mentioned is the fact that human-annotated concepts are generally hard to obtain. In addition, all datasets used are synthetic, which raises questions about the usability of the approach on real data.

**Quality:**

3

**Strengths And Weaknesses:**

The main strength of this submission lies in the numerical results, which show the clear advantage of using disentangled features for downstream tasks.

On the weaknesses side:
1. I found it tricky to understand what the contribution is. This may be due to my low familiarity with some of the elements in the paper. In the beginning, based on the title and abstract, I thought that the main contribution would be around CRL, but the it turns out that the CRL component is taken as a pretrained representation. This means that the novel aspect should lie with the learning of $\alpha$, for which two variants of Group Lasso are used. I apologize if I misunderstood this aspect, but it would be helpful to state more clearly which element is novel. Is it the application of Group Lasso to this specific task that is novel? Are some of the provided proofs novel?
2. This issue is reflected in the title itself, where the three lines suggest three different elements where the main contribution is (sample efficiency? Theoretical guarantees? No need for interventions?). By reading the rest of the paper, I could not make up my mind about this.
3. As noted by the authors, CITRIS-VAE already recovers interpretable factors. What is then the contribution of the proposed approach with respect to that?
4. The numerical comparisons are against several CBMs. However, there are no comparisons against the raw factors discovered by the various VAEs used. Does the proposed matching between $M$ and $C$ lead to differences in the performances?
5. I guess the use of Group Lasso is due to the existence of structured, multidimensional causal factors. Is this so? I could not find a clear motivation in the paper.
6. The authors claim that their method performs “better than the theory predicted”. I would like to seem some insights into how this is the case, since I could not find any in the discussion section.

---

> ### Author Rebuttal · Authors · 2025-07-30
>
> We thank the reviewer for their careful reading of our paper. To answer the points raised:
>
> ## Contribution and Title
>
> The main contribution of the paper is the pipeline presented in Figure 1(Left), where we leverage a pretrained CRL method (potentially trained on large numbers of unlabelled data) to propose a Concept Bottleneck Model approach that has theoretical guarantees, e.g. no concept leakage (a guarantee that we get by using the CRL method as a first step) and error bounds on the coefficients of the alignment function that extracts the learned concepts given a small number of labelled samples.
>
> The task itself is novel, since there are no other methods that (i) provide **theoretical guarantees** on the learned concepts in terms of error bounds, (ii) separate the CBM approach in two steps (of which one is unlabelled) and hence can use only a very small number of labelled data as opposed to standard approaches, hence being **sample efficient** (iii) provide **theoretical results** on how many labels would be needed for a certain accuracy. At **no point in this pipeline we require an intervention** on the concepts to learn the alignment with the concepts, since we have an already disentangled representation from CRL.
>
> The use of the GroupLasso is the basis for our theoretical results on the error bounds and sample complexity, but in principle other methods could be used. All of the proofs are novel, since this is a novel setting also in statistical learning theory, i.e., learning a permutation and an element-wise transformation from the learned representations to the concepts. Since we use Group lasso, we leverage some of the results in high-dimensional and non-parametric statistics as a starting point.
>
> ## CITRIS-VAE does not recover interpretable factors
>
> As other CRL methods, CITRIS-VAE is an unsupervised representation learning method and it cannot recover interpretable factors, since the learned representations in the latent space are not semantically meaningful (i.e., they are embeddings of the ground truth variables). So we still need to learn an alignment function between these representations and the concepts by using concept labels, and part of our contribution is how to do this efficiently in terms of concept labeled samples and with theoretical guarantees. As opposed to other CRL methods, CITRIS-VAE also recovers the correct permutation, which simplifies learning this alignment, but at the cost of having knowledge of which variable is intervened upon at each timestep in the dataset for the pre-training phase. Since knowing which interventions are performed or having interventions at all in the data is in general not true for other CRL methods, in our evaluation we have ignored this permutation and compared CITRIS-VAE with the other methods on the same setting.
>
> ## Additional comparisons against raw factors would not make sense
>
> We are mostly interested in the quality of the learned concepts and the raw factors do not have the right permutation or any semantic meaning, so it wouldn’t make sense to compare them directly to how accurately they predict the concept labels, except through learning a mapping from them to the labels with supervision, which is exactly the point of our paper.
>
> The CRL literature also uses labels in the evaluation, but in that case the mappings are learned ad hoc through NNs and with hundreds of thousands of labels. We instead provide a principled approach with theoretical guarantees for this task.
>
> ## Why Group Lasso: non-linear estimation
>
> In the main paper, we first use the GroupLasso so we can use $p$-dimensional feature maps $\phi(M_j)$ (e.g. polynomials or splines) for each causal variable $M_j$ with the linear estimator, allowing us to model some non-linear relationships in that setting as well. All the outputs of the feature maps for a specific $M_j$ represent a group that the Group Lasso will consider together, e.g. by considering all $p$ corresponding coefficients together. The kernelized estimator uses the same idea, but it generalizes it to non-parametric settings.
>
> For the same reason, The Group Lasso is also useful for the more general case in which the $M_j$ are multi-dimensional, which we consider in the appendix.
>
> ## Results better than the theory predicted
>
> Our theoretical results assume an upper bound on the correlation between the $M_j$, but in our experiments we observe that this upper bound can be significantly relaxed. We believe the reason is that the weighted matching in Equation $6$ is very robust to estimation errors in the $\widehat{\beta}_i^j$. For instance, it will find the correct matching if, for all $i$, $||\widehat{\beta}_i^{\pi(i)}|| > ||\widehat{\beta}_i^j||$ for $j \not= \pi(i)$. This is why we write in Section 7 that we believe there may be room to strengthen the theory. We will add this discussion to the paper.
>
> ## Limitations: human-annotated concepts are hard to obtain
>
> This is true, but we believe this issue actually points to a strength of our framework, because we require much fewer concept annotations than existing concept-bottleneck methods by leveraging unsupervised CRL as part of our framework. The sample complexity we provide also informs practitioners how many labels are at least needed, which can help in reducing the cost of labelling, as not too many labels are gathered.

---

> > ### Comment · Reviewer_G2in · 2025-08-01
> >
> > I would like to thank the authors for their response to my comments. They have clarified my understanding of the contribution and I would be inclined towards raising my score.
> >
> > There's one element that's still a bit fuzzy to me (I again apologize if this is due to my lack of familiarity with some aspects of the paper): I understand that CITRIS-VAE requires interventions in order to provide the correct permutation. CBMs (and the proposed method) use concept annotations to induce the correct permutation. Maybe this is obvious, but I would like to see an explanation on why the data required to perform the interventions is fundamentally different from the one needed for the concept annotations. Had this been explicit in the paper, I would have more clearly understood the motivation for reshuffling the CITRIS-VAE representation and not comparing against the raw factors (before reshuffling).

---

> > > ### Author Response · Authors · 2025-08-02
> > >
> > > Thanks for your quick response and your engagement.
> > >
> > > As mentioned, we don’t need interventional data for our method. Concept labels are just annotations of datapoints, e.g. images, so can also be defined on any type of data, including observational data, i.e., data without interventions, which is what we use in the paper.
> > >
> > > Instead, CITRIS-VAE requires interventional data, where the value of each causal variable is intervened or “forced” to be different from the value it would usually be. Intervening can break correlations between the causal variables and create uncommon combinations of values in the intervened data. CITRIS-VAE then leverages the knowledge of which variable is perturbed to isolate its representation in the latent space.
> > >
> > > The difference between the two types of data can be even more easily explained in a simpler setting than the TemporalCausal3D setting we considered: a Pong game with two paddles (see Figure 6 in [Lippe et al. 2022]). Here observational data are just sequences of images of the game that we observe.
> > >
> > > Interventional data are instead data in which we perturb the causal variables, e.g. the positions of the paddles, the ball or the score. For example, we could intervene to create a pair of images in which at timestep $t$ the score is $0:0$, but at timestep $t+1$ we force it to be $6:0$ by intervening on the score variable. We could also intervene to create images in which the paddle is in a position that cannot naturally occur in the game, e.g. in the center of the image, because we intervened on its position.
> > >
> > > As a side note, the type of data we used is orthogonal to the fact that we cannot compare the output of an unsupervised method (e.g., CRL methods with their raw factors) and a supervised method (ours) on the predicting the concept labels on which the supervised method is trained. Maybe it wasn’t clear in the previous response, but shuffling the CITRIS-VAE output is also mostly done to have a unified comparison with the other CRL methods that don’t recover the permutation (which is almost all of them).

---

### Official Review · Reviewer_KPGk · 2025-07-02

**Clarity:** 2
**Significance:** 3
**Originality:** 3
**Rating:** 4
**Confidence:** 2

**Summary:**

The paper proposes a two-stage framework for *sample-efficient concept learning* that couples recent Causal Representation Learning (CRL) encoders with an *alignment stage* that maps latent causal variables to user-defined concepts.  Stage 1 leverages any off-the-shelf CRL method to obtain latent variables that – under suitable identifiability conditions – recover the ground-truth causal factors up to permutation and element-wise transformations.  Stage 2 estimates this unknown permutation and the per-variable transforms with only a small number of concept labels.  Two estimators are introduced:

* **Linear estimator** based on a Group-Lasso regression that comes with finite-sample, high-probability guarantees on both parameter error and exact permutation recovery (Thm 4.2) and an explicit sample-complexity bound.
* **Kernelized estimator** that removes linearity restrictions and remains consistent under general invertible per-variable diffeomorphisms (Thm 5.2).

Experiments on synthetic toy data, Action/Temporal-Sparsity time-series benchmarks, and the image dataset *Temporal-Causal3DIdent* show accurate permutation recovery and competitive downstream classification accuracy with substantially fewer concept labels than Concept Bottleneck Models (CBMs) and their variants.

**Questions:**

1. Can you provide quantitative evidence (or theory) for how permutation recovery degrades when CRL variables contain mixtures of causal factors?
2. Given that empirical $\rho$ exceeds Assump. 4.1, do you observe systematic failures when correlations approach 0.9–0.99?  Could a debiased estimator or decorrelation pre-processing tighten the finite-sample guarantees?
3. Have you considered extending the alignment to *sets* of latent variables (blocks) mapping to a single concept?
4. Would the method scale to CUB or CheXpert concepts if CRL encoders were available?
5. Do you foresee extending Theorem 4.2 to the logistic case, or providing a finite-sample excess-risk bound for binary concepts?

**Ethical Concerns:**

["NO or VERY MINOR ethics concerns only"]

**Final Justification:**

I am inclined toward a borderline accept.

**Limitations:**

Yes

**Quality:**

3

**Strengths And Weaknesses:**

#### Strengths

The work addresses an important bottleneck of CBMs – heavy dependence on concept supervision – by showing how unsupervised CRL can substitute for most concept labels.  Theoretical analysis is unusually thorough: the linear estimator enjoys *finite-sample* guarantees with an explicit tuning rule for the regulariser, while the kernelised version attains asymptotic consistency.  The presentation of assumptions, proof sketches, and ablation studies on the effect of regularisation, dimension, correlation, and data size is clear and technically solid.  Empirically, the method achieves lower Oracle Impurity Scores and higher label accuracy than strong CBM baselines when concept labels are scarce.  The code-free alignment step is light-weight and could be dropped into existing CRL pipelines, potentially broadening the impact of CBMs in domains where labels are expensive.

#### Weaknesses

1. **Over-idealised core assumption.** All theoretical guarantees hinge on the *pre-trained CRL achieving almost perfect disentanglement*, i.e. recovering each latent $G_i$ up to permutation + one-dimensional invertible transform.  In practice, available CRL methods on complex images or temporal data often yield *partially* disentangled variables and are sensitive to permutation/grouping instability; once this fails, the subsequent Group-Lasso or kernel step may learn mis-aligned coefficients, yet the paper provides no error bounds or robustness analysis beyond a brief empirical remark that “it still works”.
2. **Correlation assumptions mis-match practice.** Finite-sample theory requires the *weak-correlation* condition (Assump. 4.1) for Group-Lasso. The authors admit that the empirical $\rho$ in experiments “far exceeds” the theoretical regime, but no new convergence analysis is given, leaving the most central finite-sample bound without guarantees in realistic high-collinearity regimes.
3. **Equating human concepts with causal variables is too strong.** The framework *assumes* that target “concepts” coincide with the data-generating latent causes (or monotone transforms).  In many domains (medical imaging, NLP) human-level concepts are abstractions over multiple causal factors.  While future work on “causal abstraction” is mentioned, the present proofs, estimator design and all experiments rely on the stronger assumption.
4. **Narrow experimental coverage.** Three out of four datasets are synthetic and the sole real-data benchmark (Temporal-Causal3DIdent) is a rendered scene dataset.  There is no evaluation on natural images, text, or genuine human-annotated concept benchmarks, so interpretability remains at the metric-level (OIS/NIS) rather than user utility.
5. **Discrete concepts lack theory.** Although the paper states that the method “also handles binary concepts” via a logistic Group-Lasso variant, there is *no* consistency or sample-complexity analysis for discrete or categorical variables; multi-class concepts such as “object shape” in Causal3DIdent are ignored in practice, leaving an important gap for real-world deployments.

---

> ### Author Rebuttal · Authors · 2025-07-30
>
> We thank the reviewer for an insightful evaluation of our paper.
>
> We acknowledge the listed ``weaknesses", but we view them as current limitations, which can potentially be addressed in future work. In theoretical work, there is often a gap between what can be mathematically proven and what happens in practice. Ideally this gap is closed over time, but even if it isn’t then theory is still helpful: it points out which aspects play an important role, it provides a sanity check that the approach works as advertised at least in simple cases, and it makes very precise and general claims that do not depend on the specific data set under consideration.
>
> ## Over-idealised core assumption
>
> First, a clarification: our framework, including the theory, already covers **partially disentangled** variables in the sense of “block-identifiability”, if we know which factors are grouped together, as shown in the Appendix.
>
> As mentioned in the Conclusions, extending our theoretical results to imperfect disentanglement or larger correlations between concepts is interesting future work. Note that in the benchmarks that we consider, which come from CRL literature, existing CRL methods already provide a good disentanglement and work well enough for our approach to beat existing CBM methods, as also evidenced by the low OIS scores.
>
> > Can you provide quantitative evidence (or theory) for how permutation recovery degrades when CRL variables contain mixtures of causal factors?
>
> We performed a new experiment on the synthetic data, where we sample $d/2$ pairs for the $M_j$ variables and during the alignment step, we feed mixed versions of these causal variables to the estimator, $M_j’= (1-a)M_i + a M_j$. We see that the permutation error goes to 1 whenever $a>0.5$, but we remain almost errorless when we consider the permutation correct if the matching identifies the pairs correctly. The columns indicate the mixing parameter values, the experiments were run over 10 seeds and we provide the mean and the standard deviation.
>
> ### Mean Permutation Error Table
> |          Feature map          |     .00     |    0.1      |    0.22     |     0.33    |     0.44    |    0.56     |    0.67     |    0.78     |    0.89     |    1.0      |
> |:-----------------------------:|:-----------:|:-----------:|:-----------:|:-----------:|:-----------:|:-----------:|:-----------:|:-----------:|:-----------:|:-----------:|
> | Linear                        | 0.00 ± 0.00 | 0.00 ± 0.00 | 0.00 ± 0.00 | 0.00 ± 0.00 | 0.00 ± 0.00 | 1.00 ± 0.00 | 1.00 ± 0.00 | 1.00 ± 0.00 | 1.00 ± 0.00 | 1.00 ± 0.00 |
> | RFF(p=8)                      | 0.00 ± 0.00 | 0.00 ± 0.00 | 0.00 ± 0.00 | 0.00 ± 0.00 | 0.04 ± 0.01 | 0.99 ± 0.01 | 1.00 ± 0.00 | 1.00 ± 0.00 | 1.00 ± 0.00 | 1.00 ± 0.00 |
> | Splines (n_knots=4, degree=3) | 0.00 ± 0.00 | 0.00 ± 0.00 | 0.00 ± 0.00 | 0.00 ± 0.00 | 0.00 ± 0.00 | 1.00 ± 0.00 | 1.00 ± 0.00 | 1.00 ± 0.00 | 1.00 ± 0.00 | 1.00 ± 0.00 |
> | Kernel (RBF)                  | 0.00 ± 0.00 | 0.00 ± 0.00 | 0.00 ± 0.00 | 0.00 ± 0.00 | 0.00 ± 0.00 | 1.00 ± 0.00 | 1.00 ± 0.00 | 1.00 ± 0.00 | 1.00 ± 0.00 | 1.00 ± 0.00 |
>
> ### Paired Mean Permutation Error table:
> |          Feature map          |    0.00     |    0.11     |    0.22     |    0.33     |    0.44     |    0.56     |    0.67     |    0.78     |    0.89     |    1.00     |
> |:-----------------------------:|:-----------:|:-----------:|:-----------:|:-----------:|:-----------:|:-----------:|:-----------:|:-----------:|:-----------:|:-----------:|
> | Linear                        | 0.00 ± 0.00 | 0.00 ± 0.00 | 0.00 ± 0.00 | 0.00 ± 0.00 | 0.00 ± 0.00 | 0.00 ± 0.00 | 0.00 ± 0.00 | 0.00 ± 0.00 | 0.00 ± 0.00 | 0.00 ± 0.00 |
> | RFF(p=8)                      | 0.00 ± 0.00 | 0.00 ± 0.00 | 0.00 ± 0.00 | 0.00 ± 0.00 | 0.03 ± 0.01 | 0.03 ± 0.01 | 0.00 ± 0.00 | 0.00 ± 0.00 | 0.00 ± 0.00 | 0.00 ± 0.00 |
> | Splines (n_knots=4, degree=3) | 0.00 ± 0.00 | 0.00 ± 0.00 | 0.00 ± 0.00 | 0.00 ± 0.00 | 0.00 ± 0.00 | 0.00 ± 0.00 | 0.00 ± 0.00 | 0.00 ± 0.00 | 0.00 ± 0.00 | 0.00 ± 0.00 |
> | Kernel (RBF)                  | 0.00 ± 0.00 | 0.00 ± 0.00 | 0.00 ± 0.00 | 0.00 ± 0.00 | 0.00 ± 0.00 | 0.00 ± 0.00 | 0.00 ± 0.00 | 0.00 ± 0.00 | 0.00 ± 0.00 | 0.00 ± 0.00 |
>
>
> ## Correlation assumptions mis-match practice
>
> It is clear that strongly correlated concepts cannot be distinguished with a limited amount of data using any method, so some assumptions are unavoidable. But there indeed remains a gap here between what we can prove and what we would like to prove. We suspect that this empiric robustness to correlations comes from the matching procedure, but we have not been able to prove it yet.
>
> > Do you observe systematic failures when correlations approach 0.9–0.99?
>
> Yes, we do, but if the correlations between the concepts are that high, e.g. 0.99, then the concepts are (almost) deterministically related. If this is due to the underlying causal variables, even most CRL methods would not be able to disentangle them, since they often assume faithfulness (which implies no deterministic relations). If this is due to the imperfect disentanglement, then we might need to assume that the true factors have a substantially higher correlation with the correct concept than the spurious ones, and this might increase the necessary sample size for the concept labels.
>
> ## Equating human concepts with causal variables is too strong
>
> We agree with this point. As mentioned in the Conclusions, we intend to actively pursue a causal abstraction approach, but this would be too much in a single paper.
>
> ## Narrow experimental coverage
>
> We have focussed here on data sets from the CRL literature with a clear ground truth. Moving to more natural data would indeed be an important threshold to cross in follow-up work.
>
> > Would the method scale to CUB or CheXpert concepts if CRL encoders were available?
>
> This would first require developing the theory for binary/discrete concepts, and using CRL methods for discrete causal variables, which are currently quite limited to very specific assumptions, e.g. the assumption that different variables always affect different parts of an observation [Kong et al. 2024].
>
> [Kong et al. 2024] Kong, Lingjing, et al. "Learning discrete concepts in latent hierarchical models." Advances in Neural Information Processing Systems 37 (2024): 36938-36975.
>
> ## Discrete concepts lack theory
>
> Besides the points regarding the lack of general purpose discrete CRL methods as described above, in general we believe there is no fundamental obstacle to extending our theoretical results to the logistic Group Lasso. Indeed, the results in [Meier, vd Geer, Bühlmann, 2008] would allow us to prove that the permutation can be recovered from a logistic regression on binary labels with the group lasso regularization. They also comment that their results can be extended to generalized linear models, which indicates we can apply the result to the feature maps we consider.
>
> Meier, van der Geer, Bühlmann, The Group Lasso for Logistic Regression, Journal of the Royal Statistical Society Series B: Statistical Methodology 70.1 (2008): 53-71.
>
> > “Do you foresee extending Theorem 4.2 to the logistic case, or providing a finite-sample excess-risk bound for binary concepts?”
>
> Yes, as we said in the previous results. Consistency and variable selection results exist for logistic regression with the group lasso regularization, which would allow us to prove a similar result as in the Appendix, but for the logistic loss, which is indeed the intention in future work. However, there were some non-trivial steps involved in proving this that it was beyond the scope of one article.
>
> ## Could a debiased estimator or decorrelation pre-processing tighten the finite-sample guarantees?
>
> Debiasing might indeed be useful to avoid underestimating the norm of the parameter vector. For the theory, it is known that the resulting estimates are asymptotically normally distributed, which might provide improved asymptotic guarantees, but there is some doubt if the method is reliable for finite samples [Wüthrich, Zhu, 2023].
>
> Decorrelation would not resolve this issue, as identification of the correct permutation relies on the sparse structure of the true parameter in the linear model. Adding a decorrelation step would affect the true parameter as well and in particular it would destroy this sparse structure of the true parameter and permutation identification remains infeasible. On top of the loss of sparsity. Decorrelating would involve inverting a correlation matix, which will be close to non-invertible if the correlation is high. So this procedure would face numerical instabilities in this regime.
>
> [Wüthrich, Zhu, 2023] K. Wüthrich, Y. Zhu (2023). Omitted Variable Bias of Lasso-Based Inference Methods: A Finite Sample Analysis. The Review of Economics and Statistics, 105 (4): 982–997.
>
> ## Extending the alignment to sets of latent variables (blocks) mapping to a single concept
>
> Yes, the most general version of our theory, which we report in the appendix for readability, can already handle blocks of causal variables matching to one concept.

---

### Official Review · Reviewer_rz7E · 2025-07-02

**Clarity:** 3
**Significance:** 3
**Originality:** 3
**Rating:** 5
**Confidence:** 3

**Summary:**

This work studies the problem of extracting interpretable concepts from data. This problem is broken into two steps. The first step is to extract latent causal generative factors from data, which is abstracted away as causal representation learning and is not addressed in this work. The paper focuses on the second step, which is to recover interpretable concepts from these latent factors. For this latter step, two approaches are proposed with theoretical guarantees. The first one is to use group regularized linear regression and the work presents sample complexity guarantees (w.h.p). The second one is to kernelize this procedure using basic ideas from RKHS theory and the paper obtains asymptotic guarantees borrowing results from high dimensional statistics. Experiments on synthetic and image benchmarks show the validity of their results.

**Questions:**

None

**Ethical Concerns:**

["NO or VERY MINOR ethics concerns only"]

**Final Justification:**

I read the other reviews and the authors' responses. The authors clarified my questions satisfactorily.

**Limitations:**

Limitations have been discussed.

**Quality:**

3

**Strengths And Weaknesses:**

## Strengths:

- While the field of causal representation learning is rapidly advancing, sample complexity guarantees are far less studied, so this work is a relevant and important direction in progressing the field.

- The main claims seem to be supported well with proofs, and the experimental details are well hashed out, and the paper also probed various aspects of the experimental settings. I also appreciate the detail in the proofs which make them easy to follow, as well as the extensive plots and tables.

## Weaknesses:

- The assumption that the ground truth causal variables correspond to the concepts that we would like to learn about (even the minor modifications in appendix A that blocks of causal variables correspond to a concept), seems very strong. It's not clear this will be the case and indeed, the field of causal representation learning aims to recover these causal factors regardless of their interpretability.

- Similarly, assuming that the mixing function can be recovered by CRL techniques (assumption 3.2) is abstracting away the difficulty of the problem at hand. It's this precise step that requires various assumptions such as availability of interventional data, therefore this limits the technical contributions of the results of this paper.

- The experimental results validate the theory to some extent, however the main drawback is that they're still somewhat simple and limited -- they mostly work with toy data (including misspecification) and semi-synthetic setups like causal3DIdent. However, this is in line with most prior works in this direction.

---

> ### Author Rebuttal · Authors · 2025-07-30
>
> We thank the reviewer for their thoughtful and encouraging feedback and discuss the weaknesses mentioned in the review.
>
> > The assumption that the ground truth causal variables correspond to the concepts that we would like to learn about (even the minor modifications in appendix A that blocks of causal variables correspond to a concept), seems very strong. It's not clear this will be the case and indeed, the field of causal representation learning aims to recover these causal factors regardless of their interpretability.
>
> We agree that this is one of our main limitations and, as mentioned in the Conclusions, in future work we would like to generalize it to the setting in which the concepts are abstractions of the ground truth variables, which could cover the case in which we have a very fine-grained ground truth variable (e.g. the position of an object) and a high-level interpretable concept (e.g. “top left”).
>
> In general, we need this assumption mostly to have theoretical guarantees on the fact that the learned representation will be disentangled and we do not use any of the causal semantics in the learned representations. So in principle, if in the future new “non-causal” methods that provide the same theoretical guarantees for dependent variables are developed, we could also use them in this pipeline.
>
> > Similarly, assuming that the mixing function can be recovered by CRL techniques (assumption 3.2) is abstracting away the difficulty of the problem at hand. It's this precise step that requires various assumptions such as availability of interventional data, therefore this limits the technical contributions of the results of this paper.
>
> This is indeed the main technical challenge in CRL methods, but in our case the technical challenge is the combination of techniques from many different fields. In particular, our main goal is to bridge the results from CRL, XAI and high-dimensional statistics to provide theoretical guarantees to concept-based models.  Especially the fields of CRL and XAI have often developed similar ideas and facing similar or complementary challenges, but seem to have few interactions. For CRL, this would provide a notable downstream task with real-world applicability. For XAI, we hope to provide a principled approach to CBMs that would not suffer from spurious correlations between concepts and depend on a large number of labelled data.
>
> Moreover, our empirical results seem to confirm that our methods could work also in case of large correlations between the learned representations, e.g. if the perfect disentanglement is not achieved by the CRL methods. This suggests that our theory might be currently too conservative, so in future work we plan to extend it and test it also on the settings in which Assumption 3.2 doesn’t hold.
>
> A small side note regarding the availability of interventional data, in principle our pipeline works also with CRL methods that do not need interventions, but for example require parametric assumptions, e.g. iVAE in settings for conditional exponential priors and assuming an auxiliary variable $u$ for which $z_i$ and $z_j$ are conditionally independent, or settings with polynomial mixing functions or independent support (see also Table 1 in [Ahuja et al. 2023]), or multi-view settings [Yao et al. 2024].
>
> > The experimental results validate the theory to some extent, however the main drawback is that they're still somewhat simple and limited -- they mostly work with toy data (including misspecification) and semi-synthetic setups like causal3DIdent. However, this is in line with most prior works in this direction.
>
> This is a fair point and is indeed in line with most prior works in CRL. As mentioned in the Conclusions, it would be interesting to use real-world datasets from CBM, but since in this dataset the concepts are discrete, we would need to substantially change our setup, adapt our theory to this setting and evaluate with discrete CRL methods, which are currently quite limited to very specific assumptions, e.g. the assumption that different variables always affect different parts of an observation [Kong et al. 2024].
>
> [Kong et al. 2024] Kong, Lingjing, et al. "Learning discrete concepts in latent hierarchical models." Advances in Neural Information Processing Systems 37 (2024): 36938-36975.

---

> > ### Comment · Reviewer_rz7E · 2025-08-04
> >
> > I thank the authors for their rebuttal and stand by my decision to accept.

---

### Note · Authors · 2025-08-13

We would like to thank all the reviewers for all their extensive reviews and engaging with our responses. We summarise the main points in our rebuttal.

## Additional results on mixtures of variables ##

We clarified that our framework already covers **partially disentangled variables** (or blocks of causal variables mapping to a single concept).

We provided additional results for mixtures of causal factors, which show that our methods perform well empirically in this case.

## Difference with CRL methods ##

We would like to reinstate that CRL methods are **unsupervised** methods and their factors are embeddings of concepts in a potentially arbitrary order, so we cannot directly compare to them. Our contribution is how to learn an alignment function between these representations and each of the actual concepts by using **a small number** concept labels with theoretical guarantees. Moreover, our pipeline is agnostic to the CRL method used and can also work with methods that do not require interventional data.

## Assumptions ##

While some of the assumptions that we use for the theoretical guarantees might not hold in realistic settings, our method can still work well also beyond the theory, as shown in the experiments.

The assumption that the concepts correspond to the causal variables is indeed something we would like to relax in future work by considering causal abstractions. In general, we use this assumption to have theoretical guarantees on the disentanglement of the variables and we do not use any causal semantics, so in principle we could also use other methods with similar guarantees.

## Experimental coverage ##

The datasets we are using are synthetic or image datasets generated from a CRL benchmark. It would be interesting to use real-world datasets from CBMs, but since in this dataset the concepts are discrete, we would need to substantially change our setup, adapt our theory to this setting and evaluate with discrete CRL methods, which are currently quite limited to very specific assumptions, e.g. the assumption that different variables always affect different parts of an observation [Kong et al. 2024].

[Kong et al. 2024] Kong, Lingjing, et al. "Learning discrete concepts in latent hierarchical models." NeurIPS 2024.

## Discrete concepts theory ##

While we think this extension is interesting and pointed at some ways to implement it in our framework, we believe this is beyond the scope of the paper.

---

### Decision · Program_Chairs · 2025-09-17

**Decision:**

Accept (poster)

**Comment:**

This paper explores the sample complexity of learning concepts in a limited setting where a perfect pre-trained decoder is available (i.e. perfect disentanglement), and the problem reduces to extracting concepts from the decoder. They consider both linear (finite-sample) and nonlinear (asymptotic) settings, and develop theoretical results in both settings. Experiments validate the theory.

Initially, reviews were somewhat mixed, but after discussion, all concerns were addressed and all reviewers support accepting the paper. Despite its limitations, the paper tackles an important problem, which will hopefully stimulate further extensions down the line.